# Latent space-based network analysis for brain–behavior linking in neuroimaging

**Selena Wang** [1] ✉, **Xinzhi Zhang**[2], **Yunhe Liu**[3], **Wanwan Xu**[2], **Xinyuan Tian**[2] & **Yize Zhao** [2] ✉

We propose a latent space-based statistical network analysis (LatentSNA) method that implements network science in a generative Bayesian framework, preserves neurologically meaningful brain topology and improves statistical power for imaging biomarker detection. LatentSNA (1) addresses the lack of power and inflated type II errors in current analytic approaches when detecting imaging biomarkers, (2) allows unbiased estimation of the influence of biomarkers on behavioral variants, (3) quantifies uncertainty and evaluates the likelihood of estimated biomarker effects against chance and (4) improves brain–behavior prediction in new samples as well as the clinical utility of neuroimaging findings. LatentSNA is broadly applicable across multiple imaging modalities and outcome measures in developing, aging and transdiagnostic cohorts, totaling 8,003 to 11,861 participants. LatentSNA achieves substantial accuracy gains (averaging 110–150%) and replicability improvements (averaging 153%) over existing approaches in moderate to large datasets. As a result, LatentSNA elucidates how network topology is implicated in brain–behavior relationships.

Neuroimaging encompasses techniques that provide in vivo depiction of the anatomy and function of the central nervous system to study the human brain in a noninvasive manner. Some imaging techniques focus on the structure of the brain (for example, computerized axial tomography and diffusion tensor imaging (DTI)), while others allow us to characterize brain activity or function, for example, functional magnetic resonance imaging (fMRI) and positron emission tomography (PET). A major hurdle for modeling neuroimaging data is the highly correlated and connected nature of measurements throughout the brain, not dissimilar to networks[1], which contributes to low statistical power for identifying brain–behavior links. Given the networked nature of the brain, a marriage between network science, a complexity-driven discipline focused on the shared architecture of networks emerging across physical, biological and social domains[2], and neuroimaging analysis is needed.

Neuroimaging connectivity models recognize and select meaningful patterns from neuroimages that explain individual differences in behavior, cognition and other outcomes. For example, case–control comparisons measure differences in connectivity between healthy individuals and patients to identify markers of dysfunction[3]. Univariate and marginal association analyses calculate associations between connectivity and outcomes to identify links[4]. By vectorizing unique pairwise edges from symmetric functional connectomes, connectome-based predictive modeling (CPM)[5] achieves functional imaging biomarker detection using a multivariate regression model controlling overfitting with cross-validation. Machine learning algorithms such as Ridge[6], least absolute shrinkage and selection operator (Lasso)[6], support vector machines (SVM)[7], random forest (RF)[8] and convolutional neural networks (CNNs)[9] are integrated to improve the predictability of the connectivity model for individual outcomes.

A critical challenge remains: connectivity edges are treated as independent observations, whereas evidence supports the dependent organization of brain networks as informative neurobiological indicators[10]. Graphical models, consisting of both undirected Gaussian graphical models[11] and directed acyclic graphs[12], describe the conditional dependence among random variables and directly address the violation of the independence assumption. A key task of graphical models, when applied to neuroimaging data, is to estimate and

[1]Department of Biostatistics and Health Data Science, Indiana University School of Medicine, Indianapolis, IN, USA. [2]Department of Biostatistics, Yale University, New Haven, CT, USA. [3]Department of Statistics, Texas A&M University, College Station, TX, USA. ✉e-mail: selewang@iu.edu; yize.zhao@yale.edu

create brain connectivity networks based on whether signals from two brain regions are conditionally independent of each other[13]. Although individual behaviors and outcomes can be incorporated in graphical models, they are often used to influence the estimation of brain connectivity networks[14]. By contrast, LatentSNA aims to understand the structure and property of brain networks (not their estimation) and how its structure is related to individual behaviors and outcomes. For further exploration of the differences between these two methodologies, we refer to these discussions[2,15].

What differentiates a network science-driven analytic approach is that it draws on insights regarding the universality of the communicative structures of real-world networks[1]. Characteristics such as the small-world property and sparsity are universal properties found in social networks[15], political networks[16], the World Wide Web[17] and human connectomes[18]. Shared network architectures, as a result of being governed by universal principles[15], allow us to use a common set of mathematical and statistical instruments for network modeling. Network science is characterized by mathematical investigations about the universal principles of network generation: what mathematical principles define the generations of network with power law degree distributions[1]. This discipline may have overlap with neuroimaging connectivity analysis, although they are not the same. For example, CPM analyzes neuroimaging connectivity data (is a neuroimaging connectivity analysis method), but it does not incorporate the networked (dependent) characteristics of the brain when modeling brain connectivity edges: it assumes one connectivity edge to be an independent observation from another.

LatentSNA, an inference-focused generative Bayesian framework capturing universal network topologies and leveraging latent space estimation techniques, is designed to analyze human connectomes and identify meaningful neuroimaging biomarkers of individual outcomes (Fig. 1). It comprises an integrated workflow containing three modules: networked connectome modeling (preserving transitivity and modularity), psychometric behavior profiling and two-way brain–behavior linking. We achieve robust neuroimaging biomarker detection with markedly improved statistical power, demonstrating generalizability of the method across seven neuroimaging landmark studies: Alzheimer's Disease Neuroimaging Initiative (ADNI) Grand Opportunities, ADNI Phase 2 (ADNI-GO/2) and ADNI Phase 3 (ref. [19]), Anti-Amyloid Treatment in Asymptomatic Alzheimer's Disease (A4)[20], the Human Connectome Project in Aging (HCP-A)[21], Adolescent Brain Cognitive Development Study Baseline (ABCD-B) and second release (ABCD-2)[22] and transdiagnostic data collected at Yale[23]. These studies involve eight different imaging modalities and 20 outcome measures with a total of 8,003 to 11,861 participants. LatentSNA consistently improves model fit performance over nine established methods, including three deep learning techniques (SVM, RF and CNN), two network-based brain analysis methods including penalized graph classification (GC[24]) and tensor network factorization analysis (TNFA[25]) and four popular brain–behavior linking approaches such as CPM, ridge CPM, Lasso and canonical correlation analysis (CCA). It enhances the predictability (an average of 110% improvement over TNFA and an average of 150% improvement over CPM) and replicability (averaging 153% improvement over CPM) of various imaging techniques, including fMRI, $T_1$-weighted structural MRI (sMRI), DTI and PET. Moreover, it is generalizable to different outcome measures, including but not limited to cognition, emotion, assessment of mental disorders, focal tau PET ([$^{18}$F]flortaucipir) standardized uptake value ratio (SUVR) metrics and different participant demographics across developing, aging and transdiagnostic populations.

As a result, our proposed method can substantially improve the interpretability of current neuroimaging connectivity studies, for example, providing a view of how brain network topology is implicated in brain–behavior relationships, exemplified by the ABCD study. Large-scale disruptions in the functional communicative patterns of brain connectomes across multiple interconnected functional systems are found to explain differences in internalizing symptoms among children[26]. Starlike topological architectures, known for their efficiency in information dissemination, resiliency with local transmission failure and affiliation with congestion[27], are identified as the fingerprints of internalizing psychopathology and its deterioration in children. Overall, LatentSNA demonstrates high-quality fit to various imaging data, generates scientific insights and enriches discussions surrounding existing neuroscience questions.

## Results

### Conceptual framework

Motivated by the need to enhance the power for identifying neuroimaging biomarkers, we propose LatentSNA as a generative statistical network analysis (SNA) model to identify significant links between brain networks and behavioral traits (Fig. 1). Existing SNA models often analyze brain–behavior links as one-sided regression models. These models either analyze (reduced-dimension) brain connectivity as predictors in a regression with behavior as the response[28,29] or they analyze connectivity as the response in a matrix-response regression to quantify behavioral covariate effects[30]. However, both types of models lack the ability to capture the mutual variations between behavioral profiles and brain variations, that is, brain development influences children's behavior and abnormal behaviors potentially reinforce brain abnormalities due to brain plasticity[31]. By contrast, LatentSNA allows connectivity differences to inform behavior–outcome variations and vice versa: both brain connectivity (Fig. 1b) and individual outcomes (Fig. 1a) are the targeted modeling interests. LatentSNA is ideal for detecting complicated and potentially noisy and weak signals hidden in high-dimensional functional connectivity data, for example, high heterogeneity and strong motion artifacts in children's fMRI data[32]. LatentSNA reinforces potentially weak signals in connectivity with a two-way cross-sectional brain–behavior linking module (Fig. 1c) that allows true connectivity signals and true internalizing signals to mutually inform each other, thus strengthening connectivity signals. Additionally, LatentSNA partials out random noise variations from true signal variations to further reinforce potentially weak connectivity signals.

Second, focused on inferring the relationships between brain networks and behaviors, LatentSNA is, philosophically, an inference model (also called explanatory model), not a prediction model[33,34]. LatentSNA provides uncertainty quantification for biomarker detection and robust statistical inference under the Bayesian framework (Fig. 1a–c). Inference models are built to describe how potential predictors and explanatory variables explain individual differences in responses, while prediction models ignore this process and focus on accurately predicting future responses. Inference models rely on statistical theories such as the central limit theorem and large sample properties to derive unbiased estimates of the significant effect coefficients with controlled type I error, while prediction models often introduce biases to improve prediction. Inference models are more optimal for detecting imaging biomarkers, as they allow us to quantify the uncertainty associated with the identification of imaging biomarkers, which is not possible with prediction models. With a large enough sample size, our model can, in an unbiased manner, identify true mutual relationships between the connectivity of each region and individual outcomes with high enough power and controlled type I error. Meanwhile, machine learning methods such as Lasso[35] do not offer unbiased quantification of the relationships and suffer from low power and inflated type II errors.

Third, LatentSNA builds on the statistical network modeling literature and preserves the topological structure of the brain network. Higher-order dependencies in real-world networks are defined as dependencies among three (triad) or more nodes[36]. Common examples of higher-order dependencies in real-world networks include homophily, balance and clusterability[37]. Homophily is often associated with the

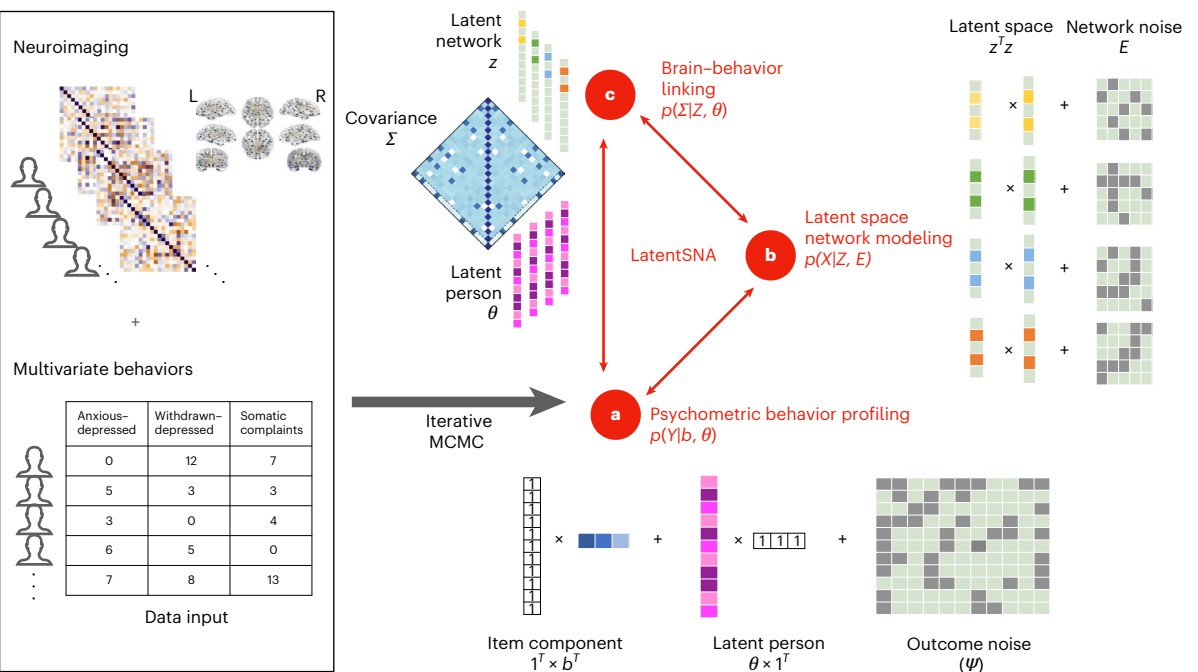

**Fig. 1 | Schematic diagram of LatentSNA.** The LatentSNA Bayesian diagram demonstrates a holistic model for multivariate outcomes $Y$ and brain networks $X$. Neuroimaging and multivariate behavior data are input into the LatentSNA model, which subsequently goes through an iterative MCMC algorithm that estimates the model parameters theorizing the data generation process of three interconnected components. **a–c**, These three interconnected components consist of psychometric behavior profiling (**a**), latent space network modeling (**b**) and brain–behavior linking (**c**). **a**, LatentSNA allows multivariate modeling of a latent behavior variable (for example, internalizing psychopathology) with multiple variables (for example, anxious–depressed, withdrawn–depressed and somatic complaints) to improve precision. The observed psychopathology is generated following a modified version of a psychometric Rasch model[61], in which outcomes are decomposed into item and person components. **b**, LatentSNA uses the symmetric bilinear interaction effect to capture network topology (transitivity, balance and clusterability)[38]. **c**, LatentSNA infers relationships between the brain and behavior, for example, internalizing psychopathology and functional connectivity. We propose a joint latent variable model, in which we allow the latent connectivity variables $Z$ and the latent behavior variables $\theta$ to covary with a shared covariance matrix, $\Sigma$. L, left; R, right.

transitive property of a network, explaining how new connections are established based on existing connections, also known as transitivity. Balance suggests a state of harmony, in which positive connections are found among nodes with similar attributes and negative connections are found among nodes with divergent attributes. Clusterability represents a more relaxed criteria for harmony than balance[38]. With balanced cycles among triads, the entire network can be divided into cohesive groups with $x_{u,v} > 0$ if nodes $u$ and $v$ are in the same group and $x_{u,v} < 0$ if they are in opposite groups[38]. Therefore, the presence of higher-order dependencies such as balance contributes to relational patterns and topology across the whole network, including higher-order dependencies. By modeling higher-order dependencies, the proposed LatentSNA captures relational patterns across the entire network.

Bilinear effects account for transitive, balanced and clusterable network structures[39]. Vector product-based latent space models, which include bilinear effect models, capture higher-order dependencies such as homophily, balance and clusterability[39]. Furthermore, such models show satisfactory model fit for networks with varying degrees of transitivity and clusterability[40]. Given that brain functional networks possess small-world properties[18], likely exhibiting both transitivity and clusterability, it is optimal for us to use bilinear effects to model higher-order dependence structures. Consequently, LatentSNA captures how network topology is implicated in brain–behavior relationships.

Finally, LatentSNA offers powerful predictions of both connectivity and behavioral variants. We provide a predictive mechanism for behavior based on connectivity, which simultaneously serves as a predictive mechanism for connectivity based on behavioral variants. Accurate prediction is achieved by incorporating latent variables to separate signal from noise, using joint modeling frameworks and allowing information communication between behavior and connectivity during model estimation. Additionally, preserving the topology of brain networks and capturing complex dependence structures is not possible with simple linear additive models.

## Assessment of LatentSNA using generated data

We compared LatentSNA to CPM, Lasso and CCA, a multivariate method exploring possible dependencies between datasets. The comparison was conducted with varying sample sizes, network sizes, signal-to-noise ratios in brain connectivity and different levels of relationships (signal proportions) between connectivity and behavior (Fig. 2). Based on both power and specificity, LatentSNA shows the highest success rate for recovering true relationships and true null relationships, making it the most sensitive and accurate method for identifying imaging biomarkers. The relatively low power observed via CPM reflects the general challenges associated with identifying imaging biomarkers when fMRI data are noisy and relationships between connectivity and behavior are sparse. To reduce prediction error, Lasso introduces a penalty term in the loss function, inducing downward bias in the coefficient estimates, and, unsurprisingly, reports the lowest power. The high specificity of Lasso is likely a byproduct of the downward bias in parameter estimation. By comparison, CCA exhibits higher power than Lasso and CPM when there are more relationships between connectivity and behavior. Using CCA, we find linear combinations of variables on both sides that maximize the dependence between the two, making CCA more powerful when the dependence is strong. Meanwhile, CCA reports low specificity when the signal proportions are large, suggesting that CCA tends to overidentify effects with high type I error when the relationships between connectivity and behavior are numerous.

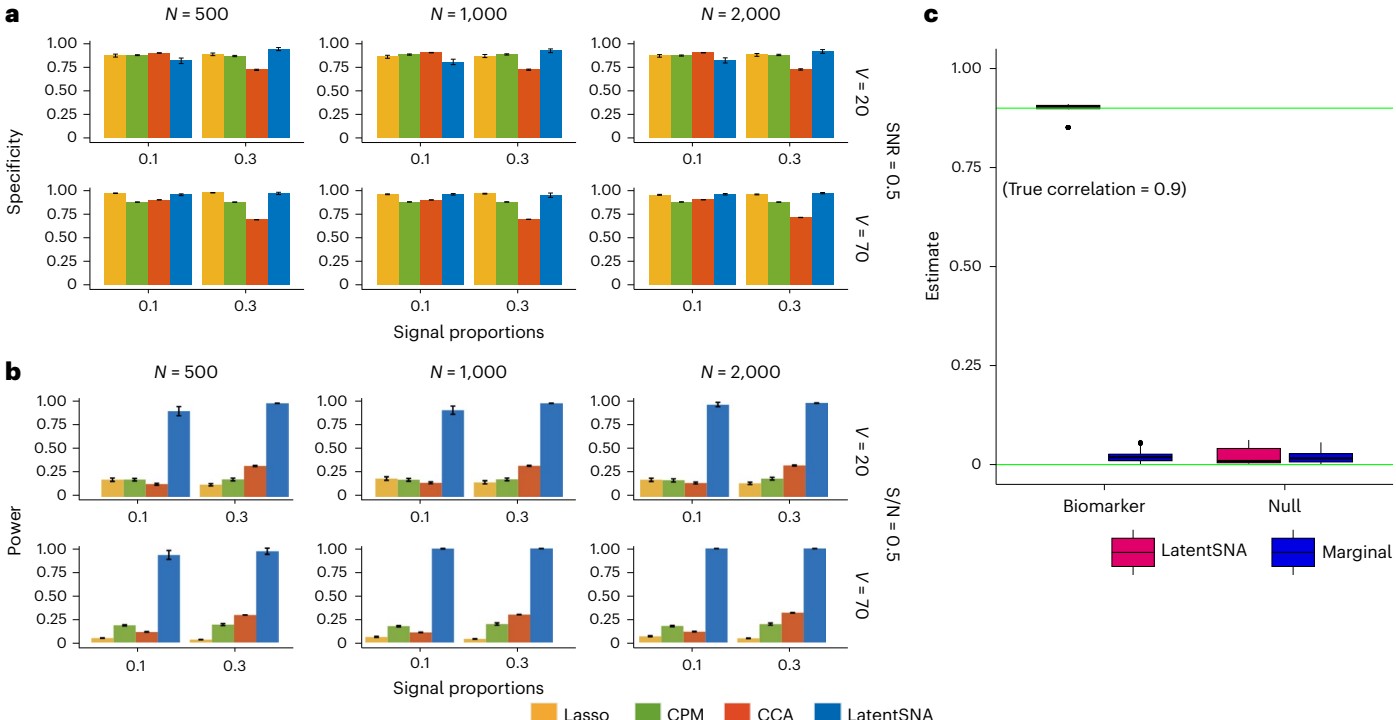

**Fig. 2 | LatentSNA improves statistical power for biomarker detection.**
**a**,**b**, Bar plots comparing the specificity (**a**) and power (**b**) of LatentSNA with CPM, Lasso and CCA in different data situations across 100 replications; replication is generated data. From left to right, the sample size increases from 500 to 1,000 to 2,000. From top to bottom, we include small ($V = 20$) and large ($V = 70$) networks as well as signal-to-noise ratios (SNR) of 0.5. We show 25–75% quantiles as error bars to reflect the uncertainty. **c**, The recovery of biomarkers versus null effects using LatentSNA versus marginal correlation tests. Box plots show the centra and 25% and 75% quantiles of the estimated effects using LatentSNA versus marginal association analysis.

To assess whether the improved power for detection translates to better prediction accuracy of behavior, we report the estimated correlation between the predicted and observed behavior in randomly sampled test data (Supplementary Fig. 3). LatentSNA demonstrates the highest prediction accuracy for behavior across various data scenarios, with accuracy increasing as the relationship between brain connectivity and behavior strengthens and as the sample size grows. Additionally, we provide the prediction accuracy of connectivity using LatentSNA (Supplementary Figs. 1–3). With LatentSNA's dual-predictive capability, we can robustly predict the connectivity of each testing sample based on behavior information. By contrast, the comparison averaging method uses the sample average connectivity as a prediction for a new participant's connectivity. In both prediction tasks, LatentSNA reliably predicts connectivity networks and behavior in new samples, particularly when connectivity and behavior are strongly related.

**Assessment of LatentSNA using real-world data**
To demonstrate the generalizability of our method across studies, we have focused on two aspects. First, our method fits well to a range of datasets with strong out-of-sample prediction accuracy. Second, our method can show robust and replicable results with consistent effect estimation across random samplings within the same study. For both predictability and replicability, we applied LatentSNA to seven different datasets, involving eight different imaging modalities and 20 outcome measures, with information from a total of 8,003–11,861 participants included. We demonstrate our method via an investigation of the ABCD study, which offers neurological insights and a view of how network topology is implicated in brain–behavior relationships during neurodevelopment.

**Improved model performance is observed across imaging modalities, outcome measures and population demographics.** To evaluate the accuracy of predicting individual outcomes in independent samples,

we apply LatentSNA to multiple landmark neuroimaging studies (Fig. 3). We demonstrate broad applicability of the model in predicting various types of outcome measures, including cognition, emotion, assessments of mental disorders and focal tau PET SUVR metrics, using imaging modalities such as structural imaging measuring fiber density, number of fibers and fiber length as well as resting and task state fMRIs. The proposed method consistently shows improvement in model fit across different data scenarios compared to seven available connectivity models: penalized GC, TNFA, CPM, ridge CPM, SVM, RF and CNN. Across varied imaging modalities, outcome measures and populations, our method consistently outperforms existing alternatives, demonstrating, for example, an average improvement of 110% over TNFA and 150% over CPM (Supplementary Figs. 1–3). These results validate LatentSNA as an interesting adaptation of statistical network analysis concepts and methods for linking real-world networks to brain and behavior.

The lack of robustness and replicability of current fMRI studies is a well-known challenge[41,42]. We investigated the robustness and replicability performance of our proposed method by comparing covariance effect estimation across random samplings of the test data, that is, replicability with the same data. We calculated the absolute correlation of the estimated effects between replications (Fig. 4). Our model-estimated effects (covariance and/or correlations between brain and behaviors) were consistent across independent replications when we randomly split the data into 90% training and 10% test samples. The CPM, on the other hand, shows lower replicability and robustness. LatentSNA shows consistently higher replicability (with correlations above 0.75) across datasets, while CPM shows substantially lower and more variable reproducibility (correlations ranging from 0.25 to 0.75).

**LatentSNA accurately predicts internalizing psychopathology and connectivity in independent samples.** We apply LatentSNA to multivariate internalizing profiles and functional connectivity during the

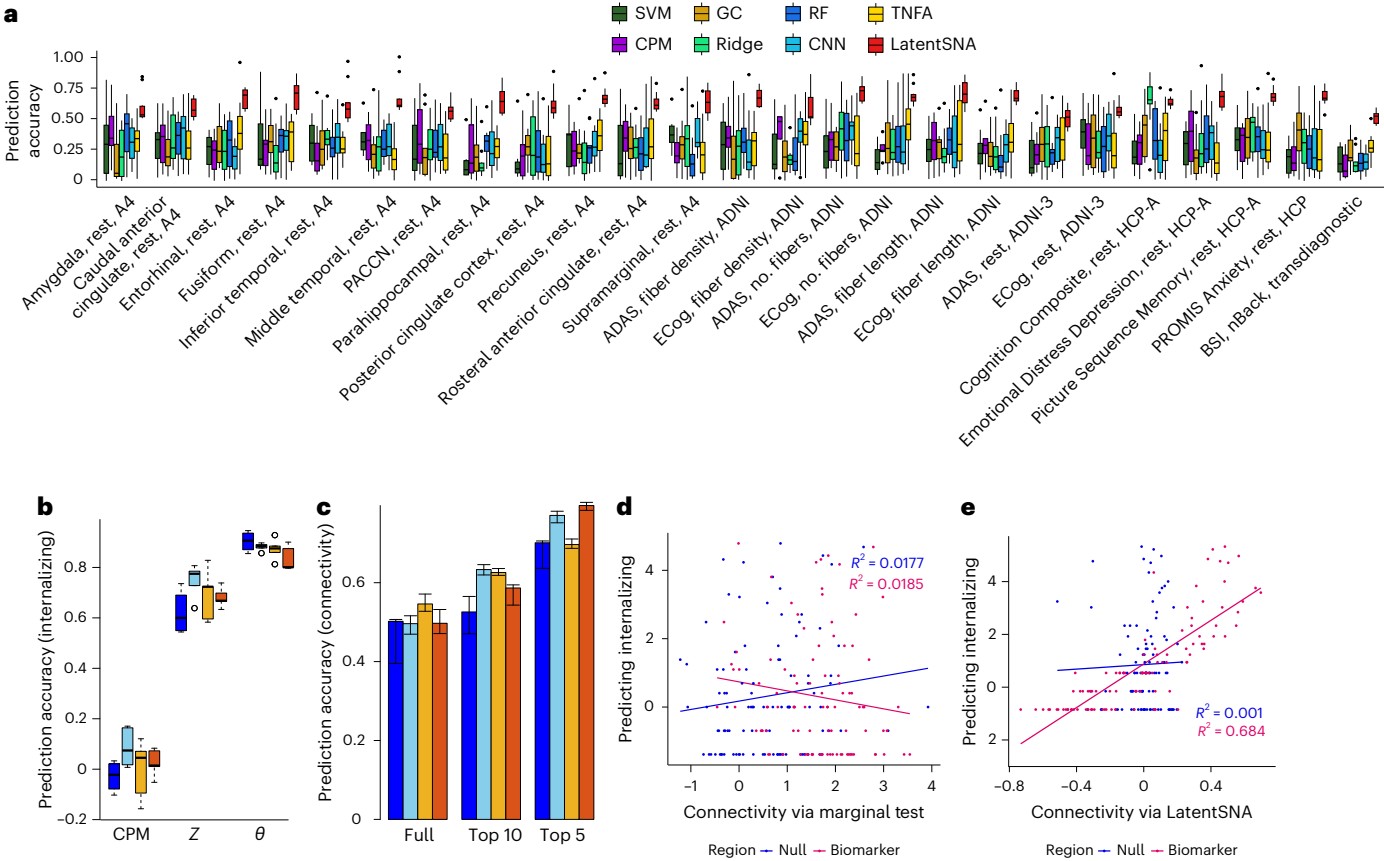

**Fig. 3 | LatentSNA shows substantially improved predictive performance over existing approaches. a**, Model performance observed across imaging modalities, outcome measures and population demographics. We include data from ADNI Grand Opportunities, ADNI-GO/2, ADNI Phase 3, A4, HCP-A and transdiagnostic data collected at Yale. The model is fitted to each imaging modality and outcome measure with sample size outlined in Supplementary Table 1. Box plots show the centra and 25% and 75% quantiles of prediction accuracy using LatentSNA versus other methods. ADAS, Alzheimer's Disease Assessment Scale; ECog, Everyday Cognition Scale; BSI, Brief Symptom Inventory; PACCN, Preclinical Alzheimer's Cognitive Composite; nBack, emotional N-back task. **b**, Box plots show the centra and 25% and 75% quantiles of correlations between observed and predicted internalizing psychopathology

for RS (blue), MID (light blue), SST (yellow) and EN-Back (red) for 100 test participants across ten runs. From left to right, predictive correlations based on CPM, LatentSNA ($Z$) and LatentSNA ($\theta$) are reported. **c**, Bar plots display correlations between observed and predicted connectivity for 100 test participants during resting and task conditions (same color scheme as before). Error bars represent the range between the 25th and 75th quantiles of prediction accuracy. From left to right, predictive correlations based on full networks, networks of the top ten internalizing regions and networks of the top five internalizing regions are reported. **d**, Scatterplot predicts internalizing values in independent samples using connectivity via the marginal correlation test. **e**, Scatterplot predicts internalizing values in independent samples using connectivity via LatentSNA.

emotional n-back task (EN-back), the stop signal task (SST) and the monetary incentive delay (MID) task conditions as well as the resting state (RS) for 5,000 to 7,000 children in the baseline ABCD study. Our aim is to uncover functional fingerprints under different cognitive states for childhood internalization, replicate the results and investigate alterations in the fingerprints between different task and resting conditions. We show the prediction accuracy results for the ABCD baseline study (Fig. 3b). We randomly split the dataset into training and test sets with ten random splits, each with a test sample size of 100 to maintain consistency across task and resting conditions. LatentSNA ($\theta$) shows median correlation above 0.9 between observed and predicted internalizing information in all four cognitive states, and LatentSNA ($Z$) shows correlations between 0.6 and 0.8. On the other hand, CPM only provides correlations around or below 0.1. This strongly supports the advantage of LatentSNA in dissecting reliable predictive information from functional connectivity under each cognitive state. Through those constructed joint learning mechanisms by LatentSNA, we can effectively predict internalizing profiles for new participants based on the available functional connectivity data.

To assess the prediction accuracy of functional connectivity, we fitted LatentSNA to training data and calculated the correlation between the observed and the estimated average connectivity (Fig. 3c). For 100

test participants, LatentSNA reports a median correlation of 0.502 for the recovery of whole-brain connectivity, 0.557 for connectivity among the top ten risk-internalizing regions and 0.707 for connectivity among the top five risk-internalizing regions. This result shows that LatentSNA provides sufficiently accurate prediction of the connectivity measurements, posing a unique opportunity to uncover brain connectivity for new participants incorporating their internalizing measures.

We compared model-identified biomarkers with null effects based on how much they can explain individual differences in internalizing psychopathology in new samples (Fig. 3d,e). We show scatterplots predicting internalizing values in independent samples using the sum of significant connectivity edges via marginal univariate tests (Fig. 3d) and connectivity via LatentSNA (Fig. 3e). The marginal test cannot differentiate the imaging biomarker from the null effect, as both predictive models show close to zero predictability for internalizing values. By contrast, LatentSNA is able to differentiate the imaging biomarker from the null effect based on predictive $R^2$ values. This result demonstrates that the estimated connectivity in LatentSNA differentiates biomarkers from null effects. The information flow between connectivity and internalizing, data integration, allows model-estimated brain connectivity to be informed by internalizing in a data-driven manner.

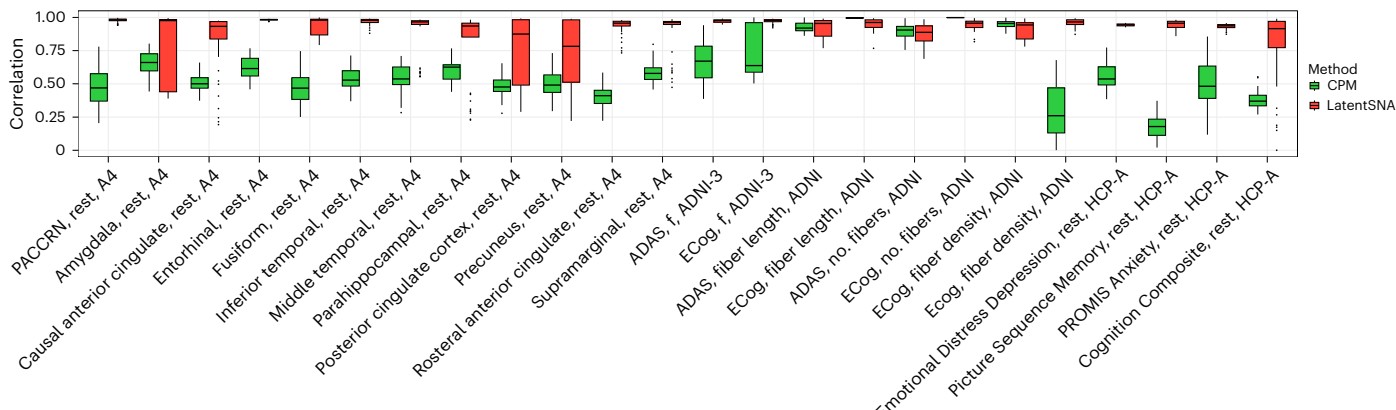

**Fig. 4 | Reproducibility analysis of CPM and LatentSNA methods across different behaviors, fMRI conditions and datasets.** Box plots show the centra and 25% and 75% quantiles of reproducibility using LatentSNA versus other methods. Each box plot summarizes reproducibility performance through correlations across ten independent replications. We use the absolute correlation between the estimated effects (regression coefficient estimates in CPM (green) and covariance estimates in LatentSNA (red)) across replications to represent the replicability and robustness of both methods. Higher correlation values indicate better reproducibility. f, resting state functional connectivity.

LatentSNA learns whether a relationship exists between the functional connectivity of a brain region and internalizing psychopathology and correctly uses or ignores that relationship depending on whether it exists. In this manner, estimated latent connectivity variables contain varying degrees of internalizing information, and a connectivity region contains more internalizing information when it is significantly linked with internalizing psychopathology and less when it is not linked with internalizing psychopathology.

Within the LatentSNA framework, we modeled internalizing psychopathology as an abstract latent construct (or variable) underlying three observed dimensions of internalizing psychopathology: the anxious–depressed, withdrawn–depressed and somatic complaint dimensions[43]. These three dimensions represent three conceptually distinct but complementary manifestations of internalizing psychopathology. Thus, by directly incorporating these dimensions into LatentSNA, we allowed more information to be included than if the internalizing psychopathology were simply modeled as the sum scores of the three dimensions, as is common in the current literature. To assess whether the incorporation of dimensions has improved modeling, we also modeled internalizing as the sum scores using LatentSNA and compared the fit. We see improvements in predicting internalizing from 0.825 to 0.885 for MID, from 0.886 to 0.893 for SST, from 0.846 to 0.901 for EN-Back and from 0.791 to 0.846 for RS when multiple dimensions are directly incorporated. This result suggests that incorporating multiple dimensions of psychopathology is superior to modeling internalizing sum scores.

**Large-scale disruptions of multiple functional systems are consistently found with internalizing psychopathology across cognitive conditions.** We report the number of significant regions identified in each of the ten functional systems[44,45]; we show the corresponding 95% credible intervals, the uncertainty quantification obtained from Markov Chain Monte Carlo (MCMC) under Bayesian inference of covariance estimates for each of the 268 brain regions (Fig. 5a,b). Across three task conditions, we found consistent involvement of 131 of the 268 brain regions and seven of ten functional systems in internalizing psychopathology, supporting internalizing psychopathology as a complex and involving large-scale affective interference of multiple coordinating functional systems. While existing psychopathology literature indicates the involvement of functional systems such as the default mode network, the prefrontal cortex, the amygdala and other structures[46], rarely do the studies have large enough power to test the disruption across the whole brain and support large-scale involvement.

Using LatentSNA, we were able to identify and replicate this involvement with other task conditions.

LatentSNA reveals a shared set of functional architectures attributable to individual variations in internalizing psychopathology when participants are tasked to perform different emotional and cognitive tasks. This finding corresponds to a recent ABCD study showing similar predictive brain features for various cognitive, personality and mental health scores[47]. During the MID task, functional connectivity shows the strongest relationship with psychopathology with the highest average covariance estimates (Supplementary Fig. 6). We show consistent discrimination of the functional systems and their contributions to developing internalizing psychopathology across tasks (Fig. 5a). While the motor system, the medial–frontal system, the basal ganglia system, the limbic system, the default mode network and the visual I systems are consistently found to be implicated in internalizing psychopathology, there is also a consistent lack of implications of the fronto-parietal, visual II and visual association systems.

The functional architectures of internalizing psychopathology are different for an intrinsic brain versus an active brain. While current literature supports the existence of an intrinsic functional brain during rest with a set of small changes common across tasks[48], little is known about differences in the functional architectures of internalizing psychopathology under different cognitive states. Our results show evidence for a difference in affective interference between RS and task states due to internalizing psychopathology. Different functional connectivity architectures are found to be implicated in internalizing psychopathology between rest and task states.

During RS, three functional systems emerge as the top risk ones to explain individual variations in internalizing psychopathology: the cerebellum, visual I and visual association systems. The cerebellum plays an important role in social and emotion processing[49], and abnormalities are found in the cerebellum during rest for individuals with depression[50] and schizophrenia[51]. Our results suggest that, during rest, the cerebellum is a major functional system contributing to internalizing psychopathology, and its relationship to internalizing is specific to an intrinsic brain, not when the brain is active. Individual differences in the spontaneous functional activities of the RS visual network, including visual I and visual associations, are also related to individual differences in internalizing psychopathology across individuals.

**The core–periphery functional network feature is more pronounced with LatentSNA.** LatentSNA differentiates signal from noise in functional connectivity networks via latent variables. Different

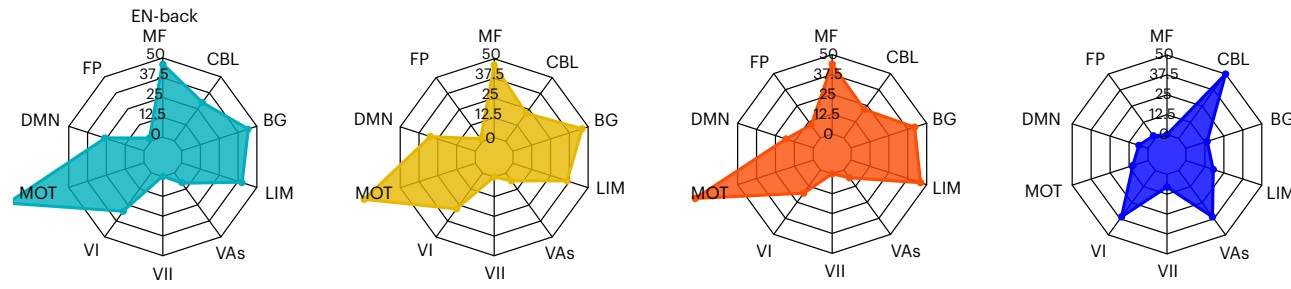

**a** Numbers of significant brain regions

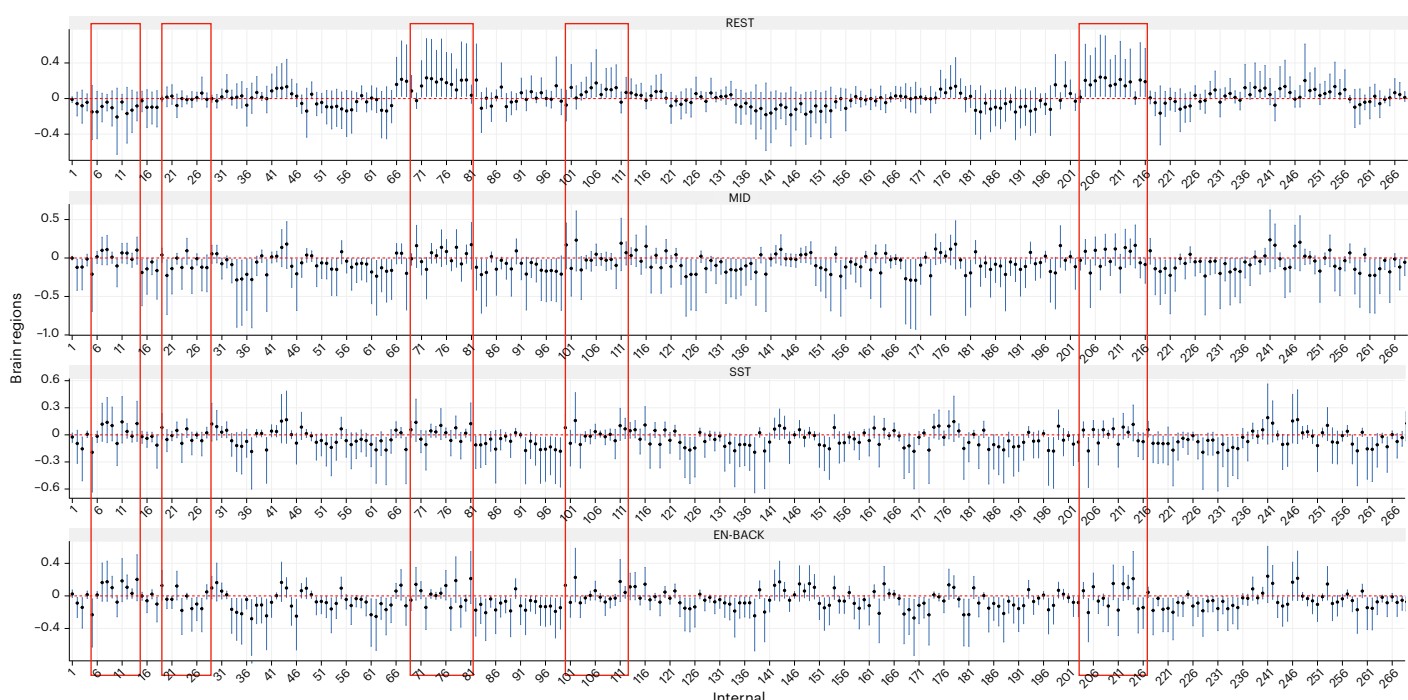

**b** 95% credible intervals of covariances between connectivity and internalizing

**Fig. 5 | Large-scale disruptions of multiple functional systems are consistently found with internalizing psychopathology across cognitive conditions.** Sample size is outlined in Supplementary Table 1. **a**, Radar plot showing the number of identified brain regions associated with ten functional systems for EN-Back, SST, MID and RS. **b**, The 95% credible intervals of covariances between the connectome and internalizing behaviors for each brain region. The centra is the estimated posterior mean of the covariance. The connectivity edges that show substantial differences between task states and resting state are highlighted in red boxes. MF, medial–frontal; FP, fronto-parietal; DMN, default mode; MOT, motor; VI, visual I; VII, visual II; VAs, visual association; LIM, limbic; BG, basal ganglia; CBL, cerebellum.

from random noise, latent variables capture patterns of meaningful variations in functional signals across individuals. In LatentSNA, each brain region is allowed to exhibit different levels of variations in functional signals across individuals and different levels of association with internalizing psychopathology. We captured true signal variations in reduced dimensions that are much smaller than the dimensions of the network, and we projected these reduced dimensions back to the network dimensions. In this manner, we obtained the latent connectivity network capturing true variations of functional signals distinct from noise. We reported observed versus estimated latent connectivity for an average participant in the MID condition (Fig. 6b). Latent connectivity shows a different topological structure than the observed network.

We show densities of the node strength and closeness based on the latent network and the observed network for an average participant in the MID condition (Fig. 6c). The distributions of node strength, for both the latent network and the observed network, are approximately symmetric based on the d'Agostino skewness test[52]. The observed network shows a platykurtic distribution with significantly negative kurtosis ($P < 10^{-5}$, Anscombe–Glynn kurtosis test[53]), while the latent network

fails to reject the null. Negative kurtosis suggests that the node strength has a flat distribution with thin tails. By comparison, the latent network has more node strength in the tails with more extremely active and extremely dormant regions. Closeness, for both the latent network and the observed network, is positively skewed with highly positive kurtosis. Compared with the observed network, the latent network shows larger skewness and kurtosis. Centrality measures show that the latent network more strongly discriminates core and peripheral regions, reflecting a more pronounced core–periphery differentiation optimal for communication, parallel to those of an efficient information distribution system[54].

By preserving the topological structure of the brain, it is not surprising that our identified imaging biomarkers are biologically meaningful and show strong associations with anatomical structures of the brain (Fig. 6a). Based on LatentSNA, the strongest biomarker signals (with covariance estimates above 0.4 and top six biomarker regions) come from the cuneus, the middle frontal gyrus, the middle temporal gyrus, the superior temporal gyrus and Heschl's gyrus. These regions play key roles in brain functions and are the central actors of the overall brain network connectivity.

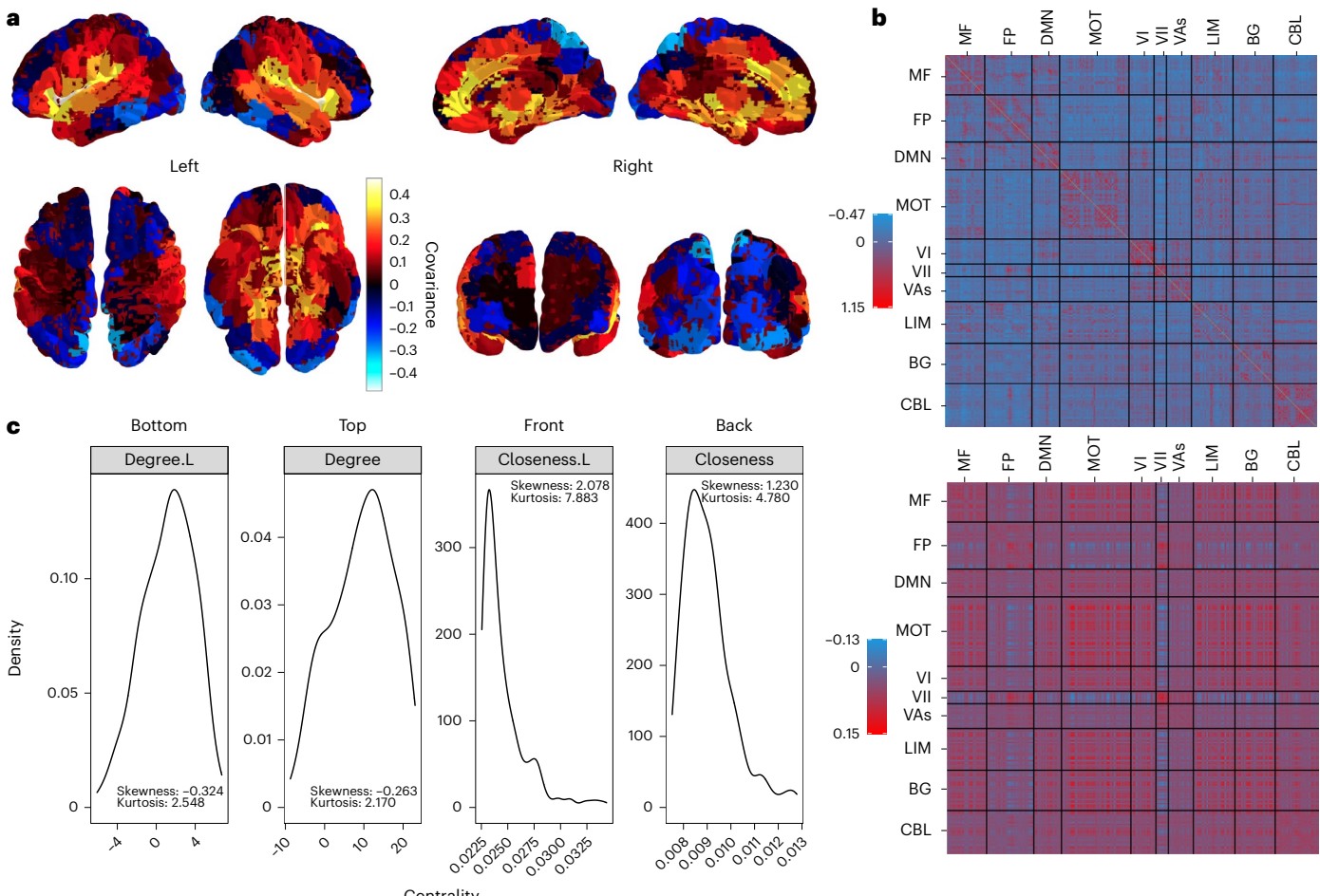

**Fig. 6 | LatentSNA identifies biologically meaningful imaging biomarkers with strong anatomical associations. a**, Brain surface plots colored by the intensity of biomarker effects. We plot estimated covarying relationships between the connectivity and internalizing of each brain region. **b**, Heatmaps of observed connectivity (top) and latent connectivity (bottom) for an average participant during the MID condition. Default Mode (DMN), Medial Frontal (MF), Fronto-

parietal (FP), Motor (MOT), Visual I (VI), Visual II (VII), Visual Association (VAs), Limbic (LIM), Basal Ganglia (BG), and Cerebellum (CBL). **c**, Densities of the centrality of brain regions measured by degrees and closeness based on the latent (L) network (left) and the observed network (right) for an average participant during the MID condition.

**Functional architectures of internalizing psychopathology are driven by the core actors of the connectivity network.** We report node strength for regions of the motor system (Extended Data Fig. 1a) and other systems (Supplementary Fig. 7). We also show node strength, closeness and betweenness for all regions (Supplementary Fig. 7). The results show that malfunctions associated with internalizing psycho- pathology are driven by the core actors of the connectivity network. The location and connectivity edges of top imaging biomarker are compared against those of the null effect (Extended Data Fig. 1b,c). Core regions with high levels of connectivity across the whole brain contribute to individual differences in internalizing psychopathology. Compared to null effects, imaging biomarkers are the central actors of the functional network with high node strength and high closeness: they are able to transmit a large quantity of information effectively. Development of internalizing psychopathology relies on regions that transmit large quantities of information (high strength) efficiently (high closeness). Low-strength and high-closeness regions are not identified as biomarkers: they tend to be the peripheral actors of the network with only localized connectivity edges.

**Internalizing psychopathology in children is attributable to starlike functional networks.** As the brain is divisible into many coordinat- ing functional systems with distinct connectivity architectures and topology, we report the latent internalizing networks with significant

internalizing biomarkers and their connectivity edges in each func- tional system in Extended Data Fig. 2. Starlike structures emerge across functional systems. These starlike structures consist of a few core actors (stars) with many links and many peripheral actors with a few links. The star nodes are almost completely connected with each other, forming a central clique, and almost all peripheral nodes are connected with the star nodes. The starlike structure corresponds to the rich club structure often found with brain networks[54]. In both structures, there are preferential connectivities among core regions. Different from the rich club structure, in the starlike structure, peripheral regions and central regions are efficiently linked with short path distances, and the peripheral regions are rarely linked to each other with low probability of connections. The starlike configurations contribute to the core–peripheral structure in the latent functional network and the skewness of the centrality distributions.

The starlike structure is consistent with the current literature on our lack of efficiency when multitasking. The starlike structure is cheap to assemble with a small number of edges and efficient searchability[55]. In an ideal one-star network, all peripheral actors are linked with the star, and there is no peripheral-to-peripheral edge. The number of steps to reach an actor in the network is always two, regardless of the network size, making the one-star network the most optimal for communication when only one information search is performed at one time. However, search on the polarized starlike networks quickly becomes expensive

when multiple searches occur simultaneously due to congestions at the star nodes.

Our star structure theory provides validity evidence for the current multitasking literature, which supports the idea that the brain is prone to congestions when multiple mental tasks are to be performed[56]. The highly connected star regions and their central cliques such as the fronto-parietal control and dorsal attention systems are crucial for completing goal-oriented tasks, but the capacities of these star regions are not limitless. When we multitask, the star regions are likely to be bombarded by competing streams of information with multiple sources of relevant and irrelevant signals, which could lead to congestion. On the other hand, with the star structure theory, the brain is efficiently organized and robust to transmission failure. The star topology reduces the impact of a transmission failure by independently connecting each peripheral region to the star clique. Peripheral regions communicate with each other via transmission to and from the star. Loss of links between peripheral regions has no impact on network communication. When there is a failure of transmission between a peripheral region and the star, the peripheral region is isolated, yet communication in the networked brain is unaffected, making it robust to failures.

Due to the efficiency of brain network communication and its general robustness to transmission failure, degeneration of the functional brain network is damaging when the star regions are compromised. Our results provide evidence for this hypothesis. Internalizing psychopathology in children is attributable to star regions and core cliques of the functional organization (Extended Data Fig. 2). Coherent starlike internalizing functional architectures are concentrated in the motor, limbic, medial–frontal, basal ganglia, default node network and visual I functional systems. By comparison, the internalizing functional networks identified through CPM do not exhibit a coherent pattern nor do they follow central–peripheral differentiation. Our results show that individual differences in the coordinating functional activities of a few star regions can explain substantial individual differences in psychopathology. Thus, the malefactions of star regions could have major impacts on the development of psychopathology and its further deterioration.

## Discussion

In this study, we developed a network science-driven analytic method that addresses the lack of power and inflated type II errors in neuroimaging biomarker detection. The proposed method represents an effort to extend SNA[1] to jointly model brain connectomes and outcome measurements, enhancing the ability to detect region-specific imaging biomarkers. While the current SNA methods mentioned above primarily focus on modeling single networks, brain connectivity networks can be viewed as multiplex networks with multiple layers of brain connections observed across a shared set of brain regions. To model these multiplex structures, a shared set of latent variables across layers can be used, assuming a joint relational structure across sets of connectivity[57]. Alternatively, we can distinguish between shared and individual components across layers[58]. In contrast to these approaches, LatentSNA captures individual differences in brain connectivity networks across layers and identifies specific brain regions where the covariation between layers of brain networks and outcome variables is substantial.

LatentSNA contributes to current neuroimaging connectivity methods by offering a high-power whole-brain approach for identifying brain–behavior links. A critical challenge of current neuroimaging connectivity methods is that connectivity edges are treated as independent observations, resulting in low statistical power and inflated type II errors. Univariate and marginal association analyses independently calculate associations between each connectivity edge and outcomes to identify significant links[4]. CPM[5] identifies imaging biomarker detection by vectorizing unique pairwise edges from symmetric functional connectomes for behavior prediction.

LatentSNA makes a contribution to existing neuroimaging regression methods such as network response regression[59] and scalar-on-network regression[60]. LatentSNA offers several advantages over network response and scalar-on-network regressions by positing a shared data generation process for connectivity and outcomes. First, unlike regression models that typically assume one-directional relationships between brain and behaviors or outcomes, estimating either the impact of brain on behavior or vice versa, LatentSNA acknowledges the mutual relationship between them. Changes in the brain often correlate with changes in behavior, but neuroplasticity suggests that disordered behaviors and dysfunctional environments can also influence brain function over time. Second, in scalar-on-network and network response regressions, using brain connectivity (or behavior outcomes) as predictors assumes that these variables are fully observed. This assumption becomes problematic when data include partially missing observations for brain connectivity and individual outcomes. Regression methods struggle to handle situations in which data are incomplete for both brain connectivity and outcomes. Lastly, traditional regression methods lack robustness in estimating parameters related to brain connectivity or behavior when they do not simultaneously model the reciprocal influence between them. By contrast, LatentSNA integrates both brain and behavior within a unified modeling framework, allowing mutual information exchange during model estimation.

The LatentSNA model has limitations that prompt important future extensions. First, with LatentSNA, researchers can obtain satisfactorily accurate predictions of both connectivity and behavioral variants in cross-section settings. Accurate prediction is achieved by incorporating latent variables to separate signal from noise, using joint modeling frameworks and allowing information communication between behavior and connectivity during model estimation. With the increased availability of longitudinal datasets such as ADNI and ABCD, it is of importance to extend current LatentSNA to longitudinal data. Longitudinal extensions would allow us to explore the temporal dynamics of fMRI across developmental or aging stages.

Second, LatentSNA offers substantially improved interpretability of neuroimaging studies, as it provides inferences about specific neuroimaging connectivity features that contribute to behavior outcomes. Future research is needed to investigate the clinical relevance of LatentSNA by exploring the specific contributions of different neuroimaging modalities in behavior predictions and investigating how these features can translate to clinical applications that ultimately improve the practical value of LatentSNA. In particular, a more clinically heterogeneous cohort is needed to understand functional substrates of psychopathologies. The ABCD study offers an opportunity to explore brain–behavior relationships in a large population of children. Yet, at these ages (9–10 at baseline and 11–12 at ABCD-2), relatively few children exhibit depression-related symptoms. Minimal participants in the ABCD study are diagnosed with depression, which limits the psychopathology findings. A more clinically heterogeneous child cohort is thus needed to explore psychopathologies in children.

Future work should also consider the group structure among the regions and how regions collectively contribute to internalizing psychopathology: past work has documented the importance of group structures of the functional brain via functional systems in cognition and disease. Beyond neuroscience, LatentSNA allows the detection of dependence between complex networks and nodal attributes, with potential applications in many other domains of science. Many complex systems such as social relationships, worldwide webs and transportation grids are impacted by higher-level attributes, and LatentSNA is a statistical technique that can open up many fields with rigorous and powerful analysis.

## Online content

Any methods, additional references, Nature Portfolio reporting summaries, source data, extended data, supplementary information,

acknowledgements, peer review information; details of author contributions and competing interests; and statements of data and code availability are available at https://doi.org/10.1038/s41592-025-02896-9.

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

## Methods

In Supplementary Table 1, we provide an overview of the study cohorts and datasets included in our analysis, consisting of the following studies: ADNI Grand Opportunities and ADNI-GO/2, ADNI Phase 3, A4, HCP-A, ABCD-B and its 2-year follow-up (ABCD-2) and the transdiagnostic data collected at Yale. We fitted the model to each combination of imaging modality and outcome measure. Our focus includes cognition outcomes, commonly used to assess the performance of new methods; and emotion outcomes, closely aligned with internalizing outcomes such as depression and anxiety; as well as disorder and focal tau PET SUVR outcomes, which directly reflect biological changes in the brain.

### Adolescent Brain Cognitive Development Study
We used brain imaging data from both the first and second releases of the largest long-term study of brain development and child health in the US, gathered from 11,875 children aged between 9 and 10 years old[22]. Here, we describe the data processing of the first-release data, and the second release was processed using the same procedures.

**Functional magnetic resonance imaging.** To investigate links, blood oxygen-level-dependent (BOLD) functional activation was recorded for children during RS and while they performed three emotional and cognitive tasks. The fMRI data underwent initial preprocessing using BioImage Suite[62]. Standard preprocessing procedures, including slice time and motion correction, and registration to the MNI template, were described in detail by Greene et al.[63] and Horien et al.[64]. Eligible scans exhibited no more than a mean frame-to-frame displacement of 0.10 mm. Brain images were parceled into 268 regions of interest (ROIs) or nodes using the Shen atlas, encompassing the cortex, subcortex and cerebellum[65]. Within each node, voxel-level time courses were aggregated. Functional connectivity was then constructed for each child in the study during both RS and each task state. Functional connectivity matrices were created, with each row and column representing all nodes, and each entry $(i, j)$ in the matrix denoting the Pearson correlation coefficient between the $i$th and $j$th nodes, scaled to be normally distributed via Fisher's $z$ transformation.

To investigate whether a shared set of neural substrates exists for internalizing psychopathology across different emotional and cognitive tasks and to determine whether these substrates differ from those observed during rest, we separately applied LatentSNA to RS functional connectivity and functional connectivity during each task state. Our analysis included 7,606 adolescents with RS functional connectivity data, capturing intrinsic brain functional activity. Additionally, we investigated the functional connectivity of 4,871 adolescents performing the EN-back task, 5,096 adolescents performing the SST and 5,298 adolescents performing the MID task.

**Internalizing psychopathology.** In the ABCD study, internalizing psychopathology is assessed through self-reported surveys using the Child Behavior Checklist (Stavropoulos et al.[43]), which comprises 119 items aggregated into eight empirical subscales. Three subscales of the Child Behavior Checklist, namely anxious–depressed (13 items), withdrawn–depressed (eight items) and somatic complaints (11 items), contribute to the assessment of internalizing psychopathology. We applied the proposed LatentSNA to both multivariate and univariate representations, interpreting the results based on the model with superior fit, namely the multivariate internalizing measures.

### Alzheimer's Disease Neuroimaging Initiative
Data used in the preparation of this article were obtained from the ADNI database (http://adni.loni.usc.edu).

**Structural magnetic resonance imaging and diffusion tensor imaging.** We downloaded $T_1$-weighted sMRI and DTI data from the ADNI-GO/2 database from 174 participants. We applied an overcomplete local

principal-component analysis[66] to process DTI data following standard steps including denoising, motion correction and distortion correction. We performed probabilistic white matter fiber tractography using fiber assignment by continuous tracking[67]. We registered sMRI scans to the lower-resolution $b_0$ volume of the DTI data using the FLIRT toolbox in the FMRIB Software Library[68], and we then defined cortical ROIs in FreeSurfer space using the Lausanne 2008 parcellation with 68 cortical ROIs[69]. We obtained the number of the fibers connecting each pair of ROIs as well as the surface area of the regions. Fiber density-based structural connectivity was calculated by dividing the number of fibers between two ROIs with their average surface areas[70]. Three types of structural brain networks were constructed as the number of fibers between a pair of brain regions, the length of the fibers as well as the fiber density of tracts connecting pairs of ROIs.

**Functional magnetic resonance imaging.** We used RS functional neuroimaging data from the third release of the ADNI study. We processed the images using the Connectome Mapper 3 pipeline[71] built in Nipype[72]. RS fMRI images were processed with despiking and slice timing correction following the method of Cox[73]; the images were also motion corrected and distortion corrected using FSL. RS fMRI images were registered to the $b_0$ sMRI using the FLIRT toolbox[74]. The BOLD time signals of each ROI were bandpass filtered and then detrended using a linear regression. We constructed functional brain networks for each participant as the Pearson correlation between the BOLD time signals for pairs of ROIs.

**Disorder–cognition outcomes.** For outcomes, we included the ADAS, Cognitive Subscale[75], a rating of dysfunction by AD. We included the ADAS score as the sum of 13 diagnostic questions collected at baseline. In addition, we included the sum score of the Everyday Cognition Scale[76], a questionnaire measuring the patient's cognitive function. We applied the proposed LatentSNA to both the ADAS and the Everyday Cognition Scale to assess the model's generalizability to alternative outcomes for the aging population.

### Anti-Amyloid Treatment in Asymptomatic Alzheimer's Disease
The A4 study is a secondary prevention trial targeted toward older people with amyloid accumulation and at high risk for AD dementia[20].

**Structural and functional magnetic resonance imaging.** For the fMRI data, we used the same processing procedure as that for the ABCD. MPRAGE scans were skull stripped using optiBET[77] and nonlinearly aligned to the MNI-152 template using BioImage Suite.

**Focal tau PET SUVR metrics.** We used PETSurfer within FreeSurfer for an integrated MRI–PET analysis[78]. We derived focal tau PET ([18F]flortaucipir) SUVR metrics from the A4 images using 90–110-min (4 × 5-min frames) post-injection images, preprocessed and analyzed using PET-Surfer in FreeSurfer (version 6.0+). We summed and motion corrected the 5-min tau PET frames. We then aligned the composite PET images to corresponding MRI images, parcellated using the Desikan–Killiany Atlas[79] and partial-volume corrected using FreeSurfer. We gathered the average tracer absorption values for each region defined by the atlas and computed SUVRs using the whole cerebellar cortex as the reference region.

**Cognition outcome.** To assess cognition changes, we included the Preclinical Alzheimer's Cognitive Composite (PACC, Donohue et al.[80]) collected as part of the A4 project. PACC is a composite cognitive score combining tests that assess episodic memory, executive function and general cognition, and it is the primary outcome measure for A4 targeting the preclinical AD population. PACC is found to be sensitive to the earliest disease-related changes[81].

## Human Connectome Project in Aging

The Lifespan HCP-A aims to characterize how brain organization and connectivity change during typical aging, compared to an 'abnormal' aging process[21].

**Functional magnetic resonance imaging.** For the fMRI data, we used the same processing procedure as that for the ABCD.

**Emotion–cognition outcomes.** For the HCP-A project, we focused on cognition and emotion measures. To assess the cognitive capability of the healthy aging population, we included the composite scores for the Picture Sequence Memory Test as well as the Cognition Composite score including Fluid Composite and Crystallized Composite, derived from all National Institutes of Health (NIH) Toolbox Cognition tasks[82]. For the emotion outcomes, we chose Emotional Distress Depression and PROMIS Anxiety to maintain relative consistency with the internalizing outcome. Emotional Distress Depression is captured by the Sadness Survey from NIH Toolbox Emotion Battery[83], which measures negative mood and perceptions. PROMIS Anxiety is captured by the Fear Affect Survey, a self-report measure assessing fear and anxious misery from NIH Toolbox Emotion Battery.

## Transdiagnostic project

The Transdiagnostic project aims to recruit clinically naturalistic and demographically diverse participants to more effectively study the links between imaging and behaviors[23]; the project was conducted at Yale between February 2018 and March 2021. Participants in the Transdiagnostic project tended to show a wide range of symptom severity and commonly had multiple psychiatric diagnoses. All imaging information was collected at the Yale Magnetic Resonance Research Center.

**Functional magnetic resonance imaging.** Preprocessing of fMRI data from the Transdiagnostic project is the same as the processing of fMRI data from the ABCD study.

**Disorder outcomes.** We included the global severity index of the Brief Symptom Inventory[84,85], a rating scale aiming to identify clinically relevant psychological symptoms in adolescents and adults. The global indices measure the level of symptomatology, its intensity and number of occurrences.

## LatentSNA

Our method makes use of techniques of Bayesian statistical inference, in which we propose a generative network model to theorize how neuroimaging connectivity and individual behaviors and outcomes intertwine with each other under random statistical processes with noise. We fitted the neuroimaging connectivity data and accompanying outcome measures and estimated covariances between the connectivity of each brain region with outcome measures across participants.

LatentSNA is motivated by the need to improve the power for detecting meaningful biomarkers of individual behaviors and outcomes using noisy imaging connectivity networks. To achieve this aim, we propose LatentSNA with a few distinctive features. First, LatentSNA is a joint model integrating imaging connectivity and behavior variants. Consider a symmetric connectivity tensor, $\mathcal{X} \in \mathbb{R}^{V \times V \times N}$, where $V$ is the number of nodes for the brain atlas and $N$ is the number of participants. Simultaneously, we have information about the behavior of the participants, denoted by the $N \times P$ matrix $Y$, where each row includes the response value for participant $i$ with $p$ outcome measurements. The proposed LatentSNA is distinct from a network response regression, where the network is the response and the effect of behavior on the network is estimated as the regression coefficient of covariates. Similarly, the model differs from a connectivity-based predictive model with behavior as the response and the network as the predictor[28]. Instead, we proposed a joint data generation process that allows

connectivity alternations to inform behavior variations and vice versa: both brain connectivity and behavior are the targeted modeling interests.

Second, LatentSNA has roots in statistical network methods and preserves the topological structure of the network. When modeling brain connectivity (one of the three components of the model), we made use of the symmetric bilinear interaction effect to capture third-order dependence patterns (transitivity, balance and clusterability) often present in symmetric networks[38,86]. While additive effects only capture variations across the rows and the columns of the network (variation in node degrees), bilinear interaction effects capture triangular structures of the network and relatedness among multiple brain regions. This is important because these higher-order dependencies exist in brain connectivity. For example, functional systems capture the coactivation of three or more brain regions that creates behavior, cognition and psychopathology. Bilinear effects capture how the distributed patterns of interactions create function and account for the complexity of integrated multimodal brain systems not possible with additive effects. For each participant, we introduced unidimensional region-specific latent variables $z_{u,i}$ to represent connectivity information for participant $i$ and region $u$ and use $z_{u,i}z_{v,i}$ as the driver of connection between brain regions $u$ and $v$ for participant $i$. Each node $u$ is part of a dependent network with strength of connection to node $v$ via the bilinear effect of the two nodes. Specifically, the connectivity between nodes $u$ and $v$, $u < v$, $u$, $v = 1, 2,..., V$ is modeled by

$$x_{u,v,i} = w_i^T \beta + a_i + z_{u,i}z_{v,i} + e_{u,v,i}, \qquad e_{u,v,i} \overset{iid}{\sim} N(0, \sigma^2), \qquad (1)$$

where $a_i$ is the fixed connectivity intercept for participant $i$, $e_{u,v,i}$ is the error term, $\sigma^2$ is the error variance and iid stands for independent and identically distributed. We adjusted for $Q$ covariates, for example, age and gender, denoted by $w_i$ with the first element to be 1 corresponding to the intercept with their effects on the connectivity matrix characterized by $\beta$. Given that each connectivity value is standardized across persons, node-level additive effects are not necessary. The mean of the connectivity values for each node across persons is zero. In matrix form, we used $Z$ to denote the $N \times V$ matrix of latent variable values, $z_i$ to denote the $V \times 1$ vector of latent variable values for participant $i$ and $E_i$ to denote the $V \times V$ matrix of errors. The approximation of the posterior distributions of the unknown quantities is facilitated by setting an MVN$(\mu_\beta, \Sigma_\beta)$, $\mu_\beta = (0, 0,..., 0, 0)^T$, $\Sigma_\beta = I_Q$ prior distribution for $\beta$, a gamma(½, ½) prior distribution for $\sigma_e^{-2}$ and an N(0, 1) prior distribution for $a_i$ (where MVN stands for multivariate normal and N for normal). The prior for the covariance of the latent network dimensions is described in the joint component.

The third distinguishing feature of LatentSNA is that it focuses on the inference of relationships between connectivity and behaviors. For each participant $i$, the probability of pairwise brain connectivity also depends on the participant's behavior $y_i$, and this influence is achieved via joint multivariate normal distribution of the connectivity and behavior parameters. Suppose that we have $\theta_i$, the unidimensional random latent variable representing the behavior information for participant $i$. The connectivity and individual behaviors and outcomes are integrated in the following way:

$$(z_{1,i}, z_{2,i}, ..., z_{V,i}, \theta_i)^T \overset{iid}{\sim} MVN \left( \begin{pmatrix} O_V \\ O_D \end{pmatrix}, \Sigma_{V+D} \right), \qquad \Sigma = \begin{pmatrix} \Lambda_z & \Lambda_{z\theta}^T \\ \Lambda_{z\theta} & \Lambda_\theta \end{pmatrix}, \quad (2)$$

where $\Lambda_{z\theta}$ is the $V \times D$ matrix modeling the relationship between connectivity and behaviors, $D = 1$. When there are nonzero elements in the $\Lambda_{z\theta}$ matrix, the connectivity and the attributes regulate and inform each other, which leads to better estimation for both connectivity and behaviors. Approximation of the posterior distribution of $\Sigma^{-1}$ is facilitated by setting a prior distribution of Wishart($I_{V+D}$, $V + D + 2$). To infer whether the connectivity of a brain region is related to behaviors, we tested whether

the corresponding covariance parameter equals zero, controlling for reflection indeterminacy. We delved deeper into the issue of reflection indeterminacy when discussing estimation. Via the joint distribution, we assume that there is a latent dependence structure between the network and the behavior, $\Sigma_{V+D}$. This dependence structure is region specific, with behavior having significant links with some brain regions and not others. This dependence structure captures the true (in a statistical sense) covariation between connectivity and behaviors across individuals, separate from variations due to random noise. If a covariance parameter is significantly different from zero, we can conclude that the associated brain region is significantly linked with behaviors, and its differences across individuals can explain individual differences in behaviors.

Last but not least, using latent behavior variables, LatentSNA allows multivariate modeling of individual behaviors and outcomes with more information to improve its estimation precision than univariate modeling. In this manner, observed individual outcomes are generated following a modified version of a psychometric Rasch model. The original Rasch model[61] proposes a data generation process for random test responses in which each test question has a unique difficulty parameter and each person is ranked based on the number of correct responses. We modified this model in a few ways. The original Rasch model does poorly at accommodating data types that are not binary. We included a more flexible linking mechanism for the latent responses and the observed data, allowing for both discrete and continuous data distributions. The original Rasch model also does not account for covariate effects such gender and race, and, to improve, we included a covariate term that allows the probability of responses to vary depending on participant demographics. Most importantly, we introduced a dependence between the latent behavior variables and connectivity, which allows the latent space of behaviors to be informed by brain connectivity. The degree of dependence is learned via data, and it organically influences how much the behavior information is integrated. As the behavior component of the joint model, participant $i$'s response on variable $p$ is modeled by

$$y_{i,p} = h_i^T \gamma + b_p + \theta_i + \epsilon_{i,p}, \qquad \epsilon_{i,p} \overset{iid}{\sim} N(0, \tau^2), \tag{3}$$

where $b_p$ is the fixed intercept for variable $p$. We adjusted for $Q'$ covariates, for example, age and gender, denoted by $h_i$ with the first element to be 1 corresponding to the intercept with their effects on the connectivity matrix characterized by $\gamma$. In matrix notation, we used $b$ to denote the $P \times 1$ vector of the intercepts, $\theta$ to denote the $N \times D$ matrix of latent variables and $\Psi$ to denote the $N \times P$ matrix of psychopathology errors. As is common in Rasch models, the parameters for the question items are fixed and the person variables are random. Approximation of the posterior distribution of the intercept parameters is facilitated by setting a standard normal prior distribution. We set a prior distribution of gamma(½, ½) for $\tau^{-2}$.

**Estimation**

Fitting the model involves iterative samples of the full conditional distributions of each parameter defined in the model until we find stable and converged Markov chains to approximate various quantities of the targeted posterior distributions via the Gibbs sampler. To achieve the global optimum for parameter estimation, we start with ten random initializations for parameter values and choose the most optimal results based on out-of-sample prediction accuracy. We iterated the following steps:

- simulate $\beta$, $a$ from their full conditional distributions,
- simulate $\sigma^2$ given $\beta$, $a$, $\tau^2$, $\gamma$, $b$, $Z$, $\theta$, $\Sigma$, $X$, $Y$,
- simulate $\gamma$, $b$ from their full conditional distributions,
- simulate $\tau^2$ given $\beta$, $a$, $\sigma^2$, $\gamma$, $b$, $Z$, $\theta$, $\Sigma$, $X$, $Y$,
- simulate $\{Z \text{ and } \theta\}$ from their full conditional distributions and
- simulate $\Sigma$ from its full conditional distribution.

To allow the information in connectivity and individual behaviors and outcomes to flow between each other and mutually inform parameter estimation, we sampled $\{Z \text{ and } \theta\}$ from their joint full conditional distribution given both the connectivity and behaviors. For participant $i$, the joint full conditional distribution of $z_i$ and $\theta_i$ is the product of the three parts (connectivity, behaviors and joint):

$$p\left(\begin{pmatrix} z_i \\ \theta_i \end{pmatrix} | t_i, \tilde{f}_{u,i}, \Sigma, \sigma_\epsilon^2\right)$$

$$\propto p(t_i|\theta_i, \sigma_\epsilon^2) p(\tilde{f}_{u,i}|z_{u,i}) p\left(\begin{pmatrix} z_i \\ \theta_i \end{pmatrix} | \Sigma\right) \propto \exp\left(-\frac{1}{2}\sigma_\epsilon^{-2} \sum_{p=1}^{P} (t_{i,p} - \theta_i)^2\right) \tag{4}$$

$$\exp\left(-\frac{1}{2}\sum_{v=1,v\neq u}^{V} (\tilde{f}_{u,v,i} - cz_{u,i}^T z_{v,i})^2\right) \& \exp\left(-\frac{1}{2}\begin{pmatrix} z_i \\ \theta_i \end{pmatrix}^T \Sigma^{-1} \begin{pmatrix} z_i \\ \theta_i \end{pmatrix}\right),$$

where $T = Y - 1b^T - H\gamma 1_p^T$ and $F_i$ is $X_i - a_i - w_i\beta = z_i z_i^T + E_i$. We can transform $F_i$ in such a way that the transformed error term is a standard normal distribution using $\tilde{F}_i = cF_i$, where $c = \sigma_\epsilon^{-1}$. Therefore, $\tilde{F}_i = cz_i z_i^T + \tilde{E}_i$, where $\tilde{e}_{u,v,i}$ follows a standard normal distribution. The joint part of the distribution $p(\binom{z_{u,i}}{\theta_i}|\Sigma')$ can be written as $\exp(-\frac{1}{2}(z_{u,i}Q_z' z_{u,i} + z_{u,i}Q_{\theta z}' \theta_i + \theta_i Q_{z\theta}' z_{u,i} + \theta_i^T Q_\theta' \theta_i))$, where $\Sigma^{-1} = \begin{pmatrix} Q_z & Q_{\theta z} \\ Q_{z\theta} & Q_\theta \end{pmatrix}$ (each component is a function of $\Lambda$s) and $\Sigma'$ is part of $\Sigma$ only involving the specific brain region. Extracting relevant terms from $p(\binom{z_i}{\theta_i}|t_i, \tilde{f}_{u,i}, \Sigma, \sigma_\epsilon^2)$, we can see that the full conditional distribution of $z_{u,i}$ is

$$p\left(z_{u,i}|\tilde{f}_{u,i}, \Sigma, \theta_i\right)$$

$$\propto \exp\left(-\frac{1}{2}z_{u,i}\left(\sum_{v=1,v\neq u}^{V} c^2 z_{v,i}z_{v,i} + Q'\right)z_{u,i}\right.$$

$$\left. + z_{u,i}^T\left(\sum_{v=1,v\neq u}^{V} c\tilde{f}_{u,v,i}z_{v,i} - \frac{1}{2}Q_{\theta z}'\theta_i - \frac{1}{2}Q_{zy}'^T \theta_i\right)\right), \tag{5}$$

a multivariate normal distribution, with variance $\left(\sum_{v=1,v\neq u}^{V} c^2 z_{v,i}z_{v,i} + Q_z'\right)^{-1}$ and mean $\left(\sum_{v=1,v\neq u}^{V} c^2 z_{v,i}z_{v,i} + Q_z'\right)^{-1}\left(\sum_{v=1,v\neq u}^{V} c\tilde{f}_{u,v,i}z_{v,i} - \frac{1}{2}Q_{\theta z}'\theta_i - \frac{1}{2}Q_{zy}'^T\theta_i\right)$. The latent variable value for psychopathology is informed by brain connectivity and should be sampled from

$$p\left(\theta_i|t_i, \Sigma, z_{u,i}, A, \sigma_\epsilon^2\right)$$

$$\propto \exp\left(-\frac{1}{2}\theta_i^T(\sigma_\epsilon^{-2}\sum_{p=1}^{P} \alpha_p \alpha_p^T + Q_\theta)\theta_i\right.$$

$$\left. + \theta_i^T\left(\sum_{p=1}^{P} \sigma_\epsilon^{-2}t_{i,p}\alpha_p - \frac{1}{2}Q_{\theta z}^T z_i - \frac{1}{2}Q_{z\theta}z_i\right)\right), \tag{6}$$

a multivariate normal distribution, with variance $\left(\sum_{p=1}^{P} \sigma_\epsilon^{-2}\alpha_p \alpha_p^T + Q_\theta\right)^{-1}$ and mean $\left(\sum_{p=1}^{P} \sigma_\epsilon^{-2}\alpha_p \alpha_p^T + Q_\theta\right)^{-1}\left(\sum_{p=1}^{P} t_{i,p}\sigma_\epsilon^{-2}\alpha_p - \frac{1}{2}Q_{\theta z}^T z_i - \frac{1}{2}Q_{z\theta}z_i\right)$. Crucially, we sampled the covariance matrix $\Sigma$ from an inverse Wishart (IW) $(I_{V+D} + F'^T F', N + V + D + 2)$ with $F'$ as an $N \times (V+1)$ matrix with the $i$th row as $(z_i^T, \theta_i^T)$.

The introduction of the bilinear effect $z_{u,i}z_{v,i}$ induces partial reflection indeterminacy. For each set of latent variable values, $\hat{z}_{u,i}$ and $\hat{z}_{v,i}$, the positions given by $-\hat{z}_{u,i}$ and $-\hat{z}_{v,i}$ give the same set of product and consequently the same likelihood. During the MCMC chain, the sign of $z_{u,i}$, $u = 1$ can change while maintaining the same connectivity value. Crucially, the connectivity latent variables are also related to individual behaviors and outcomes, whether $z_{u,i}$ is estimated as $\hat{z}_{u,i}$ or $-\hat{z}_{u,i}$ has consequences on the correlation between $z_{u,i}$ and $\theta_i$. Put in a different way, $z_{u,i}$ is softly identified, as the signs of $z_{u,i}$ need to satisfy the correlation between $z_{u,i}$ and $\theta_i$. To estimate such a model, we assume that, after a sufficient burn-in period, the signs of $z_{u,i}$ have reached a sufficiently optimal point, where its correlation with $\theta_i$ has researched a stabilized

estimate resembling the true correlation. After this burn-in period, we fix the signs of $z_{u,i}$ to the same as those of the target, that is, target = estimated $z_{u,i}$ from the first iteration after burn in. Therefore, there is no reflection indeterminacy issue after burn in.

The estimation algorithm for this paper was implemented in R. The code is available via the user-friendly GitHub page at https://github.com/selenashuowang/latentSNA with a tutorial. For each task condition, we performed posterior inference based on the MCMC algorithm under random initialization. No obvious nonconvergence issues were found via trace plots. For each task condition of the ABCD study, we compared the model fit of the multivariate behaviors with that of the univariate behavior outcome. The univariate outcome is the sum of the three internalizing variables as mentioned before.

Identification of imaging biomarkers is based on whether the estimated covariances between connectivity and behavior are significantly different from zero. Therefore, it is of interest to expand on the sensitivity of the prior specification of the covariance parameters.

In LatentSNA, the approximation of the posterior distribution of $\Sigma$ is facilitated by setting a prior distribution of $\mathrm{IW}(I_{V+1}, V+1+2)$ with the identify scale matrix $S_0 = I$ and degree of freedom equal to $m_0 = V+1+2$. The use of an IW distribution as a prior for the variance–covariance parameter matrix is fairly common in Bayesian analysis; see discussions of Leonard and Hsu[87]. The IW prior is a conjugate prior for the covariance matrix of the normal data. In LatentSNA, we are interested in estimating the covariance matrix $\Sigma$ of the joint distribution of the latent connectivity and behavior variables, $D = (z_{1,i}, z_{2,i}, \ldots, z_{V,i}, \theta_i)$. With the IW prior, the posterior distribution of $\Sigma$ can be obtained through Bayes' theorem:

$$p(\Sigma|D) = \frac{p(D|\Sigma)p(\Sigma)}{p(D)}. \tag{7}$$

From it, we can obtain the posterior distribution of $\Sigma$ with the specified prior distribution as

$$\Sigma|D \sim \mathrm{IW}(S_0 + F'^T F', m_0 + 2), \tag{8}$$

where $F'$ is an $N \times (V+1)$ matrix with the $i$th row as $(z_i^T, \theta_i^T)$. Therefore, the posterior mean of $\Sigma$ is a weighted average of the sample covariance matrix $F'^T F'$ and the prior mean $S_0$. When the sample size $N \to \infty$, the posterior mean approaches the sample mean.

In a sensitivity analysis conducted by Zhang[88], the author set the scale matrix as identity and varied the degrees of freedom by increasing $m_0$. With the increase in $m_0$, the posterior means become smaller and the posterior variances also become smaller. Thus, given the large sample size in the data, we expect the posterior mean of $\Sigma$ to approach the sample mean.

## Simulation

The data generation process for the simulation was as follows. For simplicity and consistency, the number of behavior variables was assigned as one in all generated data. We first generated the connectivity latent variables as well as the latent behavior variables from the multivariate normal distribution with the mean zero and the predefined covariance matrix with unit variances. To conduct a comprehensive assessment of the model performance, we created a range of data situations with varying sample sizes, connectivity scale, signal-to-noise ratio and signal proportions. To assess the model's ability to accurately identify true imaging biomarkers for outcomes that have both strong and weak biological signals, we varied the amount of true signals in the data by assigning the signal proportion to 0.1 and 0.3. When the signal proportion equaled 0.1 (0.3), we randomly assigned 10% (30%) of the covariance parameters between connectivity and behavior to be nonzero. To ensure the positive definiteness of $\Sigma$, we assigned both the covariances between connectivity and behavior and the corresponding dimensions in the

latent connectivity covariance matrix as 0.9. We randomly sampled the errors for the connectivity from a normal distribution with mean 0 and variance defined by the signal-to-noise ratio. Errors for the behavior were sampled from the normal distribution with the mean 0 and variance 0.5.

We considered three sample sizes, $N = 500$, $N = 1,000$ and $N = 2,000$ and two conditions for the number of nodes $V = 20$ and $V = 70$, and we specified two levels of the signal-to-noise ratio, 0.5 and 1, controlled by the error variance while keeping the variance of the latent variables constant. The individual-specific intercepts for connectivity and behavior were set to 0. In total, we considered 24 different scenarios combining from different signal proportions, sample sizes, node numbers and signal-to-noise ratios. Under each scenario, we simulated the 100 data.

We compared LatentSNA with CPM, Lasso and CCA. For Lasso, we fitted the model to the training set using the glmnet package[89]. We selected significant edges based on minimizing mean squared error with tenfold cross-validation. For CCA, we fitted the model to the training set using the CCA package[90], and regions with strong loadings were considered to be related to behavior. The cutoff thresholds are determined by the true signal proportions. For example, when the true signal proportion equals 0.1, we considered the top 10% of regions with highest absolute loadings to be significantly linked with behavior.

## Predicting outcomes

For LatentSNA ($\theta$), predicting the behavior outcome of a new participant amounts to additional draws for each new $y_i$ from a distribution with probability determined by the model. For LatentSNA ($Z$), on the other hand, predicting the behavior outcome of a new participant is based on the estimated latent connectivity variable $Z$ from the training data. We evaluated the out-of-sample predictive performance for LatentSNA ($Z$) and LatentSNA ($\theta$) as follows:

- We randomly sampled 100 participants and their behavior outcome as the test data and the other sets of data points as the training data.
- We fitted the training data to LatentSNA and obtained the posterior mean of the model parameters.
- For LatentSNA ($\theta$),
  - Predicting the behavior outcome of a new participant amounts to additional draws for each new $y_i$ from a distribution with probability determined by the model.
  - The full conditional of the new observations $y^{(\mathrm{test})}$ is, for any $y_i \in y^{(\mathrm{test})}$, determined by $\pi(y_i|\theta, b_i, \Psi_i)$.
- For LatentSNA ($Z$),
  - Predicting the behavior outcome of a new participant is based on the estimated latent connectivity variable $\hat{Z}$ from the training data.
  - We first selected significant imaging biomarkers based on 95% posterior credible intervals of the covariance parameters and used latent connectivity variables of significant imaging biomarkers as predictors.
  - Second, we split the estimated latent connectivity variables into the test set $\hat{Z}^{(\mathrm{test})}$ and the training set $\hat{Z}^{(\mathrm{train})}$ following the split of the data.
  - Third, we fitted the training model using $\hat{Z}^{(\mathrm{train})}$ as the predictors and the observed psychopathology outcomes for the training participants as the response.
  - We obtained the estimated regression coefficients $\hat{\beta}$ based on the training model.
  - Lastly, we predicted the psychopathology outcome of a new participant, for any $y_i \in y^{(\mathrm{test})}$, under LatentSNA ($Z$) following $y_i = \hat{\beta} \times \hat{Z}^{(\mathrm{test})}$.

We repeated the process ten times. Figure 3b shows the out-of-sample correlations between the observed and predicted internalizing values on the test data using LatentSNA ($Z$) and LatentSNA ($\theta$). Between LatentSNA ($Z$) and LatentSNA ($\theta$), the former does not

directly, but indirectly, incorporate training internalizing information for prediction, while the latter does. This shows that, by constructing joint learning mechanisms using LatentSNA, we can effectively predict internalizing profiles for new participants based on the available data.

**Comparison methods.** We have added model evaluation results against two network-based brain analysis methods, the penalized GC approach[24] and the TNFA. Additionally, we have incorporated comparisons with three widely used machine learning techniques, SVM[7], RF[8] and CNNs[9], to provide a comprehensive assessment of our methods' performance. The GC approach uses brain connectivity as predictors and adopts both L1 penalty, the absolute value of coefficient magnitudes and a generic group Lasso penalty. We fitted the GC approach using the graphclass R package[91]. The tuning of the penalty factor pair $(\lambda, \rho)$ was conducted on a $3 \times 4$ grid, with $\lambda$ selected from the set $\{10^{-6}, 10^{-5}, 10^{-4}\}$ and $\rho \in \{1, 10, 20, 30\}$. It was observed that a $\lambda$ value exceeding $10^{-3}$ and a $\rho$ value surpassing 40 result in the penalization of all coefficients to zero.

For the TNFA approach, similar to the tensor network principal-component analysis method[25], we embedded the $V \times V$ symmetric adjacency matrices into a low-dimensional matrix; each row contains participants' principal-component scores, and each column contains the basis network; only significant basis networks were included as predictors. We then performed a network predictor regression with the embedded low-dimensional basis networks as predictors of the outcome variables. SVM predicted behavioral outcomes based on a low-dimensional matrix derived from the $V \times V$ symmetric adjacency matrices, akin to the TNFA approach. This process involves embedding the adjacency matrices into a reduced space, where only significant basis networks were retained as predictors. These features were then used to train an SVM model with a linear kernel using the e1071 R package[92]. The model undergoes parameter tuning using a grid search to optimize the cost parameter, and the best model is used to predict behavioral outcomes from test data. The RF method is implemented using the ranger package within the caret framework in R[93], and, similarly to SVM, it uses features derived from a low-dimensional matrix of brain connectivity data. A grid search strategy optimizes key parameters: the number of variables per split (mtry), the node splitting criterion (splitrule) and the minimum node size (min.node.size). For CNN, we fitted the model with the torch package in R[94]. Our CNN architecture consists of sequential dense layers with ReLU activations, specifically designed to handle the features extracted from the low-dimensional connectivity data. The model undergoes training using an Adam optimizer and a cross-entropy loss function across multiple epochs, ensuring optimal learning from the training data. After training, the CNN is used to predict outcomes on the test dataset.

**Predicting connectivity**
We evaluated the out-of-sample performance for predicting connectivity of new participants as follows:

- We randomly sampled 100 participants and their connectivity values as the test data and the other part of data points as the training data.
- We fitted the training data to LatentSNA and obtained the posterior mean of the model parameters.
- Predicting the connectivity of a new participant amounts to additional draws for each missing $x_i \in \mathcal{X}^{(\text{test})}$ from a distribution with probability determined by the model.
- The full conditional of the new observations $\mathcal{X}^{(\text{test})}$ is, for any $x_i \in \mathcal{X}^{(\text{test})}$, determined by $\pi(x_i | z_i, a_i, E_i)$.

We repeated the process ten times. Figure 3c shows the average out-of-sample correlations between the observed and predicted connectivity values in the test data for predicting the whole graph, the top ten internalizing regions and the top five internalizing regions. The results show that LatentSNA provides sufficiently accurate prediction of the connectivity measurements, posing a unique opportunity to uncover brain connectivity for new participants, incorporating their internalizing measures.

**Comparison method.** The Average method and its extensions represent one of the most common methods to capture group-level connectivity and to perform subsequent analysis[95], often with satisfactory prediction accuracy[96]. We first randomly divided the connectivity data into ten equal sizes, using one set of data points as the test data and the other sets of data points as the training data. We then captured the group-level connectivity using the entry-wise sample mean of individual connectivity matrices in the training data. We performed predictions for connectivity in the test set using estimated connectivity from the training data. We show the average out-of-sample correlations between the observed and predicted connectivity values across 100 random samples (Supplementary Fig. 3a). Our results suggest that LatentSNA shows satisfactory prediction accuracy for brain connectivity using individual-level estimates, and it outperforms the Average method when the signal proportion is large. When predicting connectivity using group-level estimates, LatentSNA and the Average method both show satisfactory performance for the whole graph, and LatentSNA outperforms the Average method for regions with strong relational ties with behavior.

**Network statistics**
Node strength, an extension of degree in weighted networks, is the sum of the edge weights associated with each node[97]. Closeness reflects how quickly one node can reach others. We calculated closeness in the weighted graphs using the igraph R package[98], and a uniform magnitude equaling the largest negative edge is added to all edges to ensure that all weights are positive. Among the shortest paths in a network that pass through intermediate nodes, betweenness reflects how many times a node is present in those paths and demonstrates the extent to which a node is part of connections among other nodes[99]. We calculated the betweenness of the connectivity networks with positive weights defined as before using the igraph R package[100]. High betweenness reflects power as it positions the region with an important bridging role allowing the neighboring regions to connect[101], an investment into the communication between distant clusters.

**Reporting summary**
Further information on research design is available in the Nature Portfolio Reporting Summary linked to this article.

## Data availability
The individual-level imaging and behavior data used in the present study are available from four publicly accessible data resources: ABCD (https://abcdstudy.org/), HCP (https://www.humanconnectome.org/), A4 (https://www.a4studydata.org) and ADNI (https://adni.loni.usc.edu). Transdiagnostic data are available via the NDA website (https://nda.nih.gov/).

## Code availability
The estimation algorithm for this paper was implemented in R. The code is available at https://github.com/selenashuowang/latentSNA with a tutorial. The code is released under the MIT License. In this GitHub repository, we have provided instructions for installation (specifying prerequisite packages), explanations of outputs and a sample toy example with evaluations.

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

## Acknowledgements

S.W. was partially supported by the Alzheimer's Disease Data Initiative from the 2025 William H. Gates Sr Fellowship and NIH grant P30 AG072976. Y.Z. was partially supported by NIH grants R01AG068191, RF1AG081413, R01EB034720, P30AG072976 and P30AG021342. We thank the individuals represented in the ADNI (https://adni.loni.usc.edu), A4 (https://www.a4studydata.org), ABCD (https://nda.nih.gov/abcd), HCP (https://db.humanconnectome.org) and transdiagnostic studies for their participation and the research teams for their work in collecting, processing and disseminating these datasets for analysis. Some data collection and sharing for this project was funded by the ADNI (NIH grant U01 AG024904) and DOD ADNI (Department of Defense award number W81XWH-12-2-0012). The A4 study is a secondary prevention trial in preclinical Alzheimer's disease, aiming to slow cognitive decline associated with brain Aβ accumulation in clinically typical older individuals. Some data used in the preparation of this article were obtained from the ABCD study, held in the National Institute of Mental Health Data Archive (https://nda.nih.gov). HCP data were provided by the HCP WU-Minn Consortium (principal investigators D. Van Essen and K. Ugurbil; 1U54MH091657) funded by the 16 NIH institutes and centers that support the NIH Blueprint for Neuroscience Research and the McDonnell Center for Systems Neuroscience at Washington University.

The transdiagnostic study was supported by the National Institute of Mental Health (R01MH123245 and R01MH120080).

## Author contributions

S.W. and Y.Z. designed the study; S.W. and Y.L. performed the simulation; S.W., W.X., X.T. and X.Z. analyzed data; S.W. wrote the paper; and all authors have given comments and edits. All authors contributed to the interpretation of results, discussion and paper revision.

## Competing interests

The authors declare no competing interests.

## Additional information

**Extended data** is available for this paper at https://doi.org/10.1038/s41592-025-02896-9.

**Correspondence and requests for materials** should be addressed to Selena Wang or Yize Zhao.

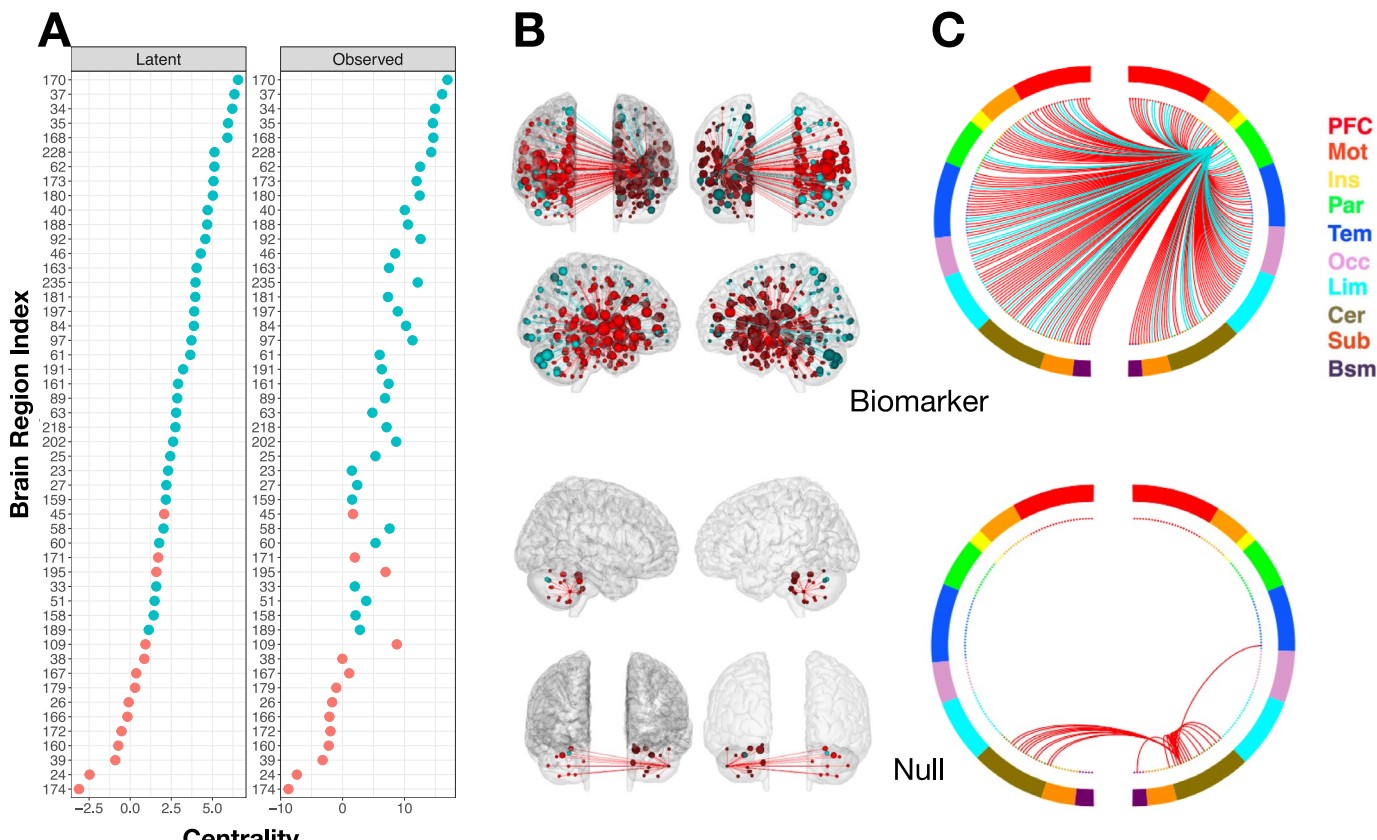

**Extended Data Fig. 1 | Functional architectures of internalizing psychopathology are driven by the core actors of the connectivity network.** (**A**) The strength of each brain region in MOT based on the latent network (left) and the observed network (right) for an average participant during MID condition. Regions identified to play a significant role in explaining individual differences in internalizing behaviors are colored as green, and non-significant regions are colored as red. (**B**) The location and connectivity networks of an imaging biomarker (top) versus a null effect (bottom) with no identifiable contribution to internalizing. The 3D brain plots show the front (top left), back (top right), right (bottom left) and left (bottom right) views. (**C**) The circle plots of the connectivity edges associated with the imaging biomarker (top) and the null effect (bottom). In B and C, red line indicates positive connectivity edges, and blue line indicates negative connectivity edges.

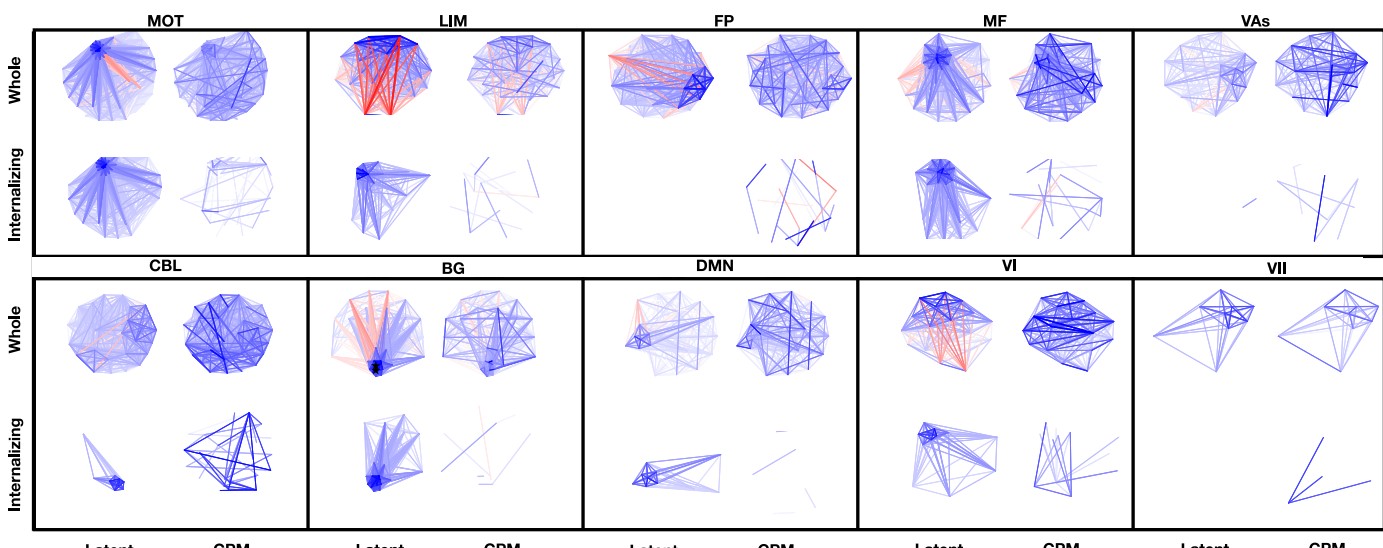

**Extended Data Fig. 2 | Internalizing psychopathology in children are attributable to star-like functional networks.** Latent internalizing networks (left) against CPM networks (right) for an average participant in each functional system during MID task. Node positions of the latent networks are then determined using the fruchterman-reingold force-directed graph layout algorithm. The nodes are fixed in the same positions when plotting the internalizing connectivity edges identified via CPM. We also show the corresponding (whole) latent networks, with both significant and non-significant connectivity edges, estimated via LatentSNA, as well as the average observed networks. MF: Medial-Frontal, FP: Fronto-parietal, DMN: Default Mode, MOT: Motor, VI: Visual I, VII: Visual II, VAs: Visual Association, LIM: Limbic, BG: Basal Ganglia, CBL: Cerebellum. Blue represents the positive connectivity edges, and red represents negative edges.

# Reporting Summary

## Statistics

For all statistical analyses, confirm that the following items are present in the figure legend, table legend, main text, or Methods section.

| n/a | Confirmed | |
|---|---|---|
| ☐ | ☒ | The exact sample size (*n*) for each experimental group/condition, given as a discrete number and unit of measurement |
| ☐ | ☒ | A statement on whether measurements were taken from distinct samples or whether the same sample was measured repeatedly |
| ☐ | ☒ | The statistical test(s) used AND whether they are one- or two-sided *Only common tests should be described solely by name; describe more complex techniques in the Methods section.* |
| ☐ | ☒ | A description of all covariates tested |
| ☐ | ☒ | A description of any assumptions or corrections, such as tests of normality and adjustment for multiple comparisons |
| ☐ | ☒ | A full description of the statistical parameters including central tendency (e.g. means) or other basic estimates (e.g. regression coefficient) AND variation (e.g. standard deviation) or associated estimates of uncertainty (e.g. confidence intervals) |
| ☐ | ☒ | For null hypothesis testing, the test statistic (e.g. *F*, *t*, *r*) with confidence intervals, effect sizes, degrees of freedom and *P* value noted *Give P values as exact values whenever suitable.* |
| ☐ | ☒ | For Bayesian analysis, information on the choice of priors and Markov chain Monte Carlo settings |
| ☐ | ☒ | For hierarchical and complex designs, identification of the appropriate level for tests and full reporting of outcomes |
| ☐ | ☒ | Estimates of effect sizes (e.g. Cohen's *d*, Pearson's *r*), indicating how they were calculated |

*Our web collection on statistics for biologists contains articles on many of the points above.*

## Software and code

Policy information about availability of computer code

| Data collection | Public data from the ABCD study, no new data are collected |
|---|---|
| Data analysis | R 4.2.0 can be downloaded at https://cran.r-project.org/bin/windows/base/old/4.2.0/ The code has been released on the code ocean with digital object identifier (DOI) 10.24433/CO.8871706.v1. Code is also available here: https://github.com/selenashuowang/latentSNA |

For manuscripts utilizing custom algorithms or software that are central to the research but not yet described in published literature, software must be made available to editors and reviewers. We strongly encourage code deposition in a community repository (e.g. GitHub). See the Nature Portfolio guidelines for submitting code & software for further information.

## Data

Policy information about availability of data

All manuscripts must include a data availability statement. This statement should provide the following information, where applicable:
- Accession codes, unique identifiers, or web links for publicly available datasets
- A description of any restrictions on data availability
- For clinical datasets or third party data, please ensure that the statement adheres to our policy

We used brain imaging data from the first release of the ABCD study collected from 11, 875 children aged between 9 to 10 years old. Public access: https://abcdstudy.org/

# Human research participants

Policy information about studies involving human research participants and Sex and Gender in Research.

| Reporting on sex and gender | We use functional brain imaging data from the first release of the Adolescent Brain Cognitive Development (ABCD) study. And We have 6,185 males and 5,681 females. |
|---|---|
| Population characteristics | We used brain imaging data from the first release of the ABCD study collected from 11, 875 children aged between 9 to 10 years old. The blood-oxygen-level-dependent (BOLD) functional activation was recorded for children during resting state (RS) and when they performed three emotional and cognitive tasks.<br>Internalizing psychopathology represents a spectrum of conditions characterized by negative emotion including depression, anxiety and phobias. In the ABCD study, the internalizing psychopathology is collected via self-reported survey using the Child Behavior Checklist (CBCL, Stavropoulos et al. , 2017), which includes 119 items aggregated into 8 empirical sub-scales. Three sub-scales of CBCL, anxious-depressed (13 items), withdrawn-depressed (8 items) and somatic complaints (11 items) are parts of the internalizing psychopathology. The multivariate representation of the internalizing psychopathology with anxious-depressed, withdrawn-depressed and somatic complaints variables likely outperforms the univariate representation (sum of the three variables) due to the loss of information in the latter. |
| Recruitment | We used brain imaging public data from the first release of the ABCD study collected from 11, 875 children aged between 9 to 10 years old. |
| Ethics oversight | Adolescent Brain Cognitive Development(ABCD)  study |

Note that full information on the approval of the study protocol must also be provided in the manuscript.

# Field-specific reporting

Please select the one below that is the best fit for your research. If you are not sure, read the appropriate sections before making your selection.

☒ Life sciences          ☐ Behavioural & social sciences          ☐ Ecological, evolutionary & environmental sciences

For a reference copy of the document with all sections, see nature.com/documents/nr-reporting-summary-flat.pdf

# Life sciences study design

All studies must disclose on these points even when the disclosure is negative.

| Sample size | We use functional brain imaging data from the first  release of the Adolescent Brain Cognitive Development (ABCD) study, collected from 11, 875 children aged between 9 to 10 years old (Casey et al, 2018). The functional MRI (fMRI) data is collected from children when they were resting capturing intrinsic brain activity (Rest), when they were performing the the emotional n-back task (EN-back), the Stop Signal task (SST), and the Monetary Incentive Delay (MID) task. |
|---|---|
| Data exclusions | We included 7, 606 adolescents with RS functional connectivity capturing intrinsic brain functional activity. We separately investigated the functional connectivity of 4, 871 adolescents who are asked to perform the emotional n-back task (EN-back), 5, 096 adolescents who are asked to perform the Stop Signal task (SST) and 5, 298 adolescents who are asked to perform the Monetary Incentive Delay (MID) task. |
| Replication | Results are successfully replicated with different task conditions. |
| Randomization | N/A; there were no experimental groups in the study. |
| Blinding | N/A; there was no group allocation in the study. |

# Reporting for specific materials, systems and methods

We require information from authors about some types of materials, experimental systems and methods used in many studies. Here, indicate whether each material, system or method listed is relevant to your study. If you are not sure if a list item applies to your research, read the appropriate section before selecting a response.

## Materials & experimental systems

| n/a | Involved in the study |
|-----|----------------------|
| ☒ ☐ | Antibodies |
| ☒ ☐ | Eukaryotic cell lines |
| ☒ ☐ | Palaeontology and archaeology |
| ☒ ☐ | Animals and other organisms |
| ☒ ☐ | Clinical data |
| ☒ ☐ | Dual use research of concern |

## Methods

| n/a | Involved in the study |
|-----|----------------------|
| ☒ ☐ | ChIP-seq |
| ☒ ☐ | Flow cytometry |
| ☐ ☒ | MRI-based neuroimaging |

# Magnetic resonance imaging

## Experimental design

**Design type**

resting state and and different cognitive, emotional and behavioral task states

**Design specifications**

We use functional brain imaging data from the first and second releases of the Adolescent Brain Cognitive Development (ABCD) study, collected from 11, 875 children aged between 9 to 10 years old (Casey et al, 2018).

**Behavioral performance measures**

Internalizing psychopathology data represents a spectrum of conditions characterized by negative emotion including depression, anxiety and phobias. In the ABCD study, the internalizing psychopathology is collected via self-reported survey using the Child Behavior Checklist (CBCL, Stavropoulos et al. , 2017), which includes 119 items aggregated into 8 empirical sub-scales. Three sub-scales of CBCL, anxious-depressed (13 items), withdrawn-depressed (8 items) and somatic complaints (11 items) are parts of the internalizing psychopathology. The multivariate representation of the internalizing psychopathology with anxious-depressed, withdrawn-depressed and somatic complaints variables likely outperforms the univariate representation (sum of the three variables) due to the loss of information in the latter.

## Acquisition

**Imaging type(s)**

functional

**Field strength**

3T

**Sequence & imaging parameters**

High spatial and temporal resolution multiband echo-planar imaging (EPI) resting-state fMRI data with fast integrated distortion correction are acquired using three 3T scanner platforms: Siemens Prisma, General Electric (GE) 750, and Phillips. Resting-state fMRI parameters are similar across platforms: a standard multiband EPI sequence, repetition timeTR)/echo time (TE) = 800/30 ms, voxel spacing size = 2.4 x 2.4 x 2.4 mm, slice number = 60, flip angle (FA) = 52, field oiview (FOV) = 216 x 216 mm, multiband acceleration = 6.
ABCD_Website_MRI_Acq link: https://abcdstudy.org/wp-content/uploads/2021/05/ABCD_Website_MRI_Acq.pdf

**Area of acquisition**

Whole brain scans were acquired.

**Diffusion MRI**   ☐ Used   ☒ Not used

## Preprocessing

**Preprocessing software**

BioImage Suite and SPM5.

**Normalization**

We removed the linear trend from all signals in accordance with the methodologydetailed in Shen et al. (2013).

**Normalization template**

MNI

**Noise and artifact removal**

We deleted scans with more than 0.10 mm mean frame-to-frame displacement.

**Volume censoring**

First, we performed motion correction and slice-time correction using SPM5; and via BioImage Suite, the data were registered to a standardized 3mm X 3mm x 3mm common space, where we generated masks representing white matter, gray matter, and cerebrospinal fluid (CSF) and computed the mean time courses for both white matter and CSF. We orthogonalized each gray matter time course with respect to the mean time courses of both white matter and CSF, and we orthogonalized each gray matter timec ourse to the six motion-related signals via SPM5. We then applied a bandpass Butterworth filterwith a frequency range of 0.02Hz to 0.1Hz to the orthogonalized time courses. We used a Gaussiankernel with a full-width at half-maximum (FWHM) of 6mm to enhance spatial coherence and spatialsmoothing. Lastly, we removed the linear trend from all signals in accordance with the methodologydetailed in Shen et al. (2013). We deleted scans with more than 0.10 mm mean frame-to-framedisplacement. Additional details about the standard preprocessing procedures, such as slice timeand motion correction, registration to the MNI template can be found in Greene et al 2018) andHorien et al (2019).

## Statistical modeling & inference

| | |
|---|---|
| Model type and settings | Imaging Biomarker effects described in the method section. It is built on joint bayesian framework |
| Effect(s) tested | Co-variation between functional connectivity and internalizing |

Specify type of analysis: ☒ Whole brain ☐ ROI-based ☐ Both

| | |
|---|---|
| Statistic type for inference<br>(See Eklund et al. 2016) | New method is proposed to test region-specific co-variation in whole functional connectivity and internalizing |
| Correction | Inference under Bayesian posterior sampling |

## Models & analysis

| n/a | Involved in the study |
|---|---|
| ☐ | ☒ Functional and/or effective connectivity |
| ☐ | ☒ Graph analysis |
| ☐ | ☒ Multivariate modeling or predictive analysis |

| | |
|---|---|
| Functional and/or effective connectivity | pearson correlation |
| Graph analysis | Joint modeling framework with both connectivity and behavior as dependent variables, statistical network analysis is used for modeling graphs |
| Multivariate modeling and predictive analysis | Statistical network analysis is used to reduce dimension via latent variable modeling. Predictions of future behaviors and connectivity are performed under the joint modeling framework. |

