## [Peer Review File · Nature Methods]

Neuroimaging connectivity analysis needs network science for brain-behavior linking

Corresponding Author: Dr Selena Wang

A version of this paper was originally rejected for publication by Nature Methods, however that decision was reconsidered after appeal by the authors.

Version 0:

Decision Letter:

5th Jan 2024

Dear Selena,

Let me first apologize for the long delays in the review process, which is all the more regrettable since I don't have positive news. Your Article entitled "Inference-based statistical network analysis uncovers star-like brain functional architectures for internalizing psychopathology in children" has now been seen by two reviewers, whose comments are attached. In the light of their advice we have decided that we cannot offer to publish your manuscript in Nature Methods.

You will see that, while they find your work of some potential interest, the reviewers raise concerns about the advance your methodological approach represents over available methods and about its broad applicability at this stage. I realize that reviewer #2 was supportive of your work, but the concerns of reviewer #1 about advance over established approaches and lack of adequate comparisons were serious enough to overrule reviewer #2. We think that these criticisms are sufficiently important as to prevent publication of your work in Nature Methods.

Although we cannot publish your paper, it may be appropriate for another journal in the Nature Portfolio. If you wish to explore the journals and transfer your manuscript please use our manuscript transfer portal. You will not have to re-supply manuscript metadata and files, unless you wish to make modifications. For more information, please see our [manuscript transfer FAQ](http://www.nature.com/authors/author_resources/transfer_manuscripts.html?WT.mc_id=EMI_NPG_1511_AUTHORTRANSF&WT.ec_id=AUTHOR) page.

I am sorry that we cannot be more positive on this occasion but hope that you find the reviewers' comments helpful when preparing your paper for submission elsewhere.

Best regards,
Nina

Nina Vogt, PhD
Senior Editor
Nature Methods

Reviewer Comments:

Reviewer #1 (Remarks to the Author):

In this paper, Wang et al. introduces a method, called latent variable-based statistical network analysis (LatentSNA), to improve the statistical power for detection imaging biomarkers. The method is applied to model multi-state functional networks with multivariate internalizing profiles using of a few thousand children in the Adolescent Brain Cognitive Development (ABCD) study. The approach was demonstrated to yield markedly improved statistical power in detecting biomarkers and the

latent variables were shown to explain the internalizing profiles better.

Comments:

1. My main concern is with the lack of focus of the paper. What is the main contribution? LatentSNA or the neuroscientific findings associated with internalizing psychopathology?

If the main contribution is LatentSNA, the method has not been shown to be generalizable. The authors claimed: "Our approach can be used to detect other types of biomarkers using positron emission tomography, T1-weighted structural MRI and Diffusion Tensor Imaging, etc. Our approach can also detect imaging biomarkers of other behavioral, cognitive and psychopathology measures." However, judging based on the limited results presented, these claims are at best unsubstantiated.

If the contribution is on the neuroscientific findings, the findings are overly narrow being focusing on internalizing psychopathology. Observations of star-like architectures are not necessarily generalizable to other conditions. The paper does not appear to be a good fit for Nature Methods.

2. The current paper compares LatentSNA only with some of the most basic methods. There are plenty of latent-variable graphical models that are applicable to brain network analysis. Even dynamic causal modeling methods can be considered as based on latent models. A common trait of these methods, like LatentSNA, is that they represent complex signals in a low-dimensional space encoded by latent variables. A significant amount of work is needed to evaluate LatentSNA against the true state of the art.

3. The authors claimed that LatentSNA uses "the symmetric bilinear interaction effect to capture possible higher-order dependencies, among three or more connectivity edges." However, Equation 0.1 seems to only be modeling the interaction between two nodes. The effect of this interaction is also not validated. What if this term is omitted from the model? How much degradation in performance, if any, will it cost. Is it possible to go beyond modeling bilinear interactions? How would modeling even higher-order dependencies affect the performance?

4. The code is not accessible, making it more challenging to assess the validity of the method.

5. Figure 1 is poorly done. It does not clearly explain the core concepts of LatentSNA. Notations are not defined, making it impossible to make any sense out of the figure.

6. On page 9 it is unclear what is meant by "the Averaging method simply takes the sample average connectivity as a prediction for a new subject's connectivity."

Reviewer #2 (Remarks to the Author):

A. The LatentSNA framework proposed in this paper combines brain functional connectivity with internalizing psychopathology. In particular, the proposed model was applied to the ABCD study, which achieved sufficiently accurate prediction of both children's internalizing traits and functional connectivity, and substantially improved the ability to explain the individual internalizing differences compared with current approaches.

B. This original work is of great significance, especially in detecting the neurologically meaningful network topology in the adolescents and children population.

C. Both simulation studies and real applications to the ABCD study were conducted to demonstrate the outperformance of the proposed method.

D. The Bayesian framework and the proposed MCMC algorithm are solid and convincing.

E. The findings provided by latentSNA are consistent with existing literature.

F. Some comments and suggestions are listed as follows:

1. Figure 1 shows the big picture of the LATentSNA. However, it is not informative and is poorly illustrated and explained in the caption contents.

2. More discussions about the differences between LatentSNA (Z) and LatentSNA (THETA) are expected.

3. What kind of cross-validation approach is adopted for the prediction tasks?

4. I would like to see the sensitivity of the prior settings in the MCMC algorithm.

5. The mechanism for generating simulation data is not very clear.

G. Previous related works are mostly cited in this paper.

H. The whole paper is well written and organized.

** For Nature Portfolio general information and news for authors, see <http://npg.nature.com/authors>.

Version 1:

Decision Letter:

20th Mar 2024

Dear Selena,

Thank you for your letter asking us to reconsider our decision on your Article, "Inference-based statistical network analysis uncovers star-like brain functional architectures for internalizing psychopathology in children". After careful consideration we have decided that we are willing to consider a revised version of your manuscript that revised as proposed in your appeal.

Please note that we will be adding an additional reviewer, should we decide to send your manuscript out for peer review again.

- * include a point-by-point response to our referees and to any editorial suggestions
- * please underline/highlight any additions to the text or areas with other significant changes to facilitate review of the revised manuscript
- * address the points listed described below to conform to our open science requirements
- * ensure it complies with our general format requirements as set out in our guide to authors at www.nature.com/naturemethods
- * resubmit all the necessary files electronically by using the link below to access your home page

Link Redacted

We hope to receive your revised paper within 2-3 months. If you cannot send it within this time, please let us know. In this event, we will still be happy to reconsider your paper at a later date so long as nothing similar has been accepted for publication at Nature Methods or published elsewhere.

OPEN SCIENCE REQUIREMENTS

REPORTING SUMMARY AND EDITORIAL POLICY CHECKLISTS

When revising your manuscript, please submit reporting summary and editorial policy checklists.

DATA AVAILABILITY

CODE AVAILABILITY

Please include a "Code Availability" subsection in the Online Methods which details how your custom code is made available. Only in rare cases (where code is not central to the main conclusions of the paper) is the statement "available upon request" allowed (and reasons should be specified).

MATERIALS AVAILABILITY

ORCID

Best regards,
Nina

Nina Vogt, PhD
Senior Editor
Nature Methods

Version 2:

Decision Letter:

11th Nov 2024

Dear Dr Wang,

Thank you for your patience. Your Article, "Neuroimaging connectivity analysis needs network science for brain-behavior linking", has now been seen by three reviewers (the two original ones and a new one). As you will see from their comments below, although the reviewers find your work of considerable potential interest, they continue to raise a number of concerns. We are interested in the possibility of publishing your paper in Nature Methods, but would like to consider your response to

these concerns before we reach a final decision on publication.

We therefore invite you to revise your manuscript to address these concerns. Specifically, please do work on the clarity of the presentation and highlight the added datasets. Please also discuss the overfitting issue brought up by reviewer #1. Finally, I would like to mention that clinical relevance is not our focus at Nature Methods, and we therefore ask not to focus on that.

Link Redacted

We hope to receive your revised paper within four weeks. If you cannot send it within this time, please let us know. In this event, we will still be happy to reconsider your paper at a later date so long as nothing similar has been accepted for publication at Nature Methods or published elsewhere.

OPEN SCIENCE REQUIREMENTS

REPORTING SUMMARY AND EDITORIAL POLICY CHECKLISTS

DATA AVAILABILITY

All novel DNA and RNA sequencing data, protein sequences, genetic polymorphisms, linked genotype and phenotype data, gene expression data, macromolecular structures, and proteomics data must be deposited in a publicly accessible database, and accession codes and associated hyperlinks must be provided in the "Data Availability" section.

CODE AVAILABILITY

Please include a "Code Availability" subsection in the Online Methods which details how your custom code is made available. Only in rare cases (where code is not central to the main conclusions of the paper) is the statement "available upon request" allowed (and reasons should be specified).

For more information on our code sharing policy and requirements, please see: <https://www.nature.com/nature-research/editorial-policies/reporting-standards#availability-of-computer-code>

MATERIALS AVAILABILITY

ORCID

Nature Methods is committed to improving transparency in authorship. As part of our efforts in this direction, we are now requesting that all authors identified as 'corresponding author' on published papers create and link their Open Researcher and Contributor Identifier (ORCID) with their account on the Manuscript Tracking System (MTS), prior to acceptance. This applies to primary research papers only. ORCID helps the scientific community achieve unambiguous attribution of all scholarly contributions. You can create and link your ORCID from the home page of the MTS by clicking on 'Modify my Springer Nature account'. For more information please visit <http://www.springernature.com/orcid>

Best regards,
Nina

Nina Vogt, PhD
Senior Editor
Nature Methods

Reviewers' Comments:

Reviewer #1 (Remarks to the Author):

1. Unfortunately, despite the authors' efforts, the paper remains unfocused. Although the authors have shown some applications of LatentSNA to multiple datasets in predicting outcome measures, a large portion of the paper is similar to the previous version. About half of the paper is dedicated to validating LatentSNA and the other half is dedicated to predicting internalizing psychopathology and connectivity based on the ABCD data.
2. The title is awkward – “connectivity analysis” has always been closely related to “network science”. So it is unclear why now “connectivity analysis needs network science”. Similar problem with the first sentence of the abstract.
3. The authors' definition of “network science” is unhelpful – “network science, a complexity-driven discipline focused on the shared architecture of networks emerging across physical, biological, and social domains.” Staring a few minutes at this sentence, I'm still having difficulty understanding what it means. The general reader will unlikely understand it as well.
4. In general, the entire paper needs to be rewritten for clarity and understandability. This is important for broader impact and readership.
5. Page 3: the mentioned limitations of graphical models are not necessarily true. Point 1: Graphical models, particularly when higher-order connections are considered, do consider network topologies. Point 2: Graphical models have certainly been used for biomarker detection. Point 3: Graphical models have been certainly used for outcome prediction.
6. Page 3, Para 2: “What differentiates a network-science-driven analytic approach is that it draws on insights regarding the universality of the communicative structures of real-world networks [barabasi2013network].” No idea what this means, particularly in the context of the current paper.
7. Page 3, Para 2: This paragraph is intended to motivate the use of statistical network-based analysis (SNA). But judging based on the way it is written, the motivation remains entirely unclear.
8. Page 4, Para 1: Why is “preserving transitivity and modularity” important? Why not consider other network properties?
9. Figure 1a remains unclear and unhelpful.
10. In Section “Markedly improved model performance is observed across imaging modalities, outcome measures and population demographics”, is “model fit” assessed for overfitting? If not, improvements might not be surprising and may be simply due to overfitting.
11. The discussion is short and does not provide meaningful insights.

Based on my comments above, I cannot recommend the manuscript for publication at Nature Methods.

Reviewer #2 (Remarks to the Author):

All my comments on the Figures, statistical method, validation strategies, and simulation settings are well addressed.

Reviewer #3 (Remarks to the Author):

In this paper, Wang et al. introduce LatentSNA, a novel latent variable-based statistical network analysis designed to enhance predictive accuracy of behavior outcomes and interpretability as providing inferences about specific neuroimaging connectivity features that contribute to behavior outcomes. I appreciate the novelty of the proposed methodology and its proof of generalizability using different image modalities for different type of outcome prediction through various datasets. However, further clarifications and rescaling are needed for the manuscript to be ready for publication.

1. Significance: The methodological significance of LatentSNA is well explained, particularly its increased statistical power, lower Type II error rates in prediction, and feature selection capabilities. Biologically, the authors emphasize that a strength of LatentSNA lies in its ability to identify neuroimaging biomarkers; however, it is not clearly stated what type of biomarkers are targeted. Additionally, the manuscript lacks detail on how these biomarkers might enhance clinical utility. Given that the ADNI and ABCD datasets are longitudinal, it is especially important to clarify what is meant by “prediction” in this context—whether LatentSNA aims to predict future behavioral outcomes based on baseline connectivity or merely capture associations in a cross-sectional manner.
2. It is not well illustrated the process of the proposed LatentSNA in Figure 1. Figure 1 provides different components in the LatentSNA, however, it is hard to make the connection between these components and understand how it works. A clearer, more sequential depiction of how LatentSNA functions would greatly improve comprehension.
3. Experiment results: It would be nice to see more results on proving the generalizability of the method rather than just heavily focusing on the ABCD dataset for the following reasons:

1) Interpretability and Clinical Relevance:

The interpretability of LatentSNA is presented as a major benefit, yet it would be helpful to more explicitly discuss the specific contributions of different neuroimaging modalities in behavior predictions. Moreover, clarifying how these features can translate to clinical applications would make the method's practical value more tangible.

2) Limitations of the ABCD Dataset for Psychopathology Findings:

The authors focus on understanding different internalizing psychopathologies using the ABCD baseline and ABCD-2 datasets, particularly examining depression-related items from the Child Behavior Checklist (CBCL), such as anxious-depressed (13 items) and withdrawn-depressed (8 items). However, at these ages (9-10 at baseline and 11-12 at ABCD-2), relatively few children exhibit depression-related symptoms. Moreover, there are minimal participants diagnosed with depression through the KSADS in the ABCD dataset, which limits the robustness of the psychopathology findings.

4. Minor: Figure 6, there is no color legend for (B,C).

Reviewer #3 (Remarks on code availability):

A more detailed documentation, including clear instructions for installation (specifying prerequisite packages), explanations of outputs, and a sample toy example with evaluations, would be greatly appreciated.

Version 3:

Decision Letter:

Our ref: NMETH-A54306C

24th Jun 2025

Dear Selena,

Thank you for submitting your revised manuscript "Neuroimaging connectivity analysis needs network science for brain-behavior linking" (NMETH-A54306C). It has now been seen by the original referees and their comments are below. The reviewers find that the paper has improved in revision, and therefore we'll be happy in principle to publish it in Nature Methods, pending minor revisions to satisfy the referees' final requests and to comply with our editorial and formatting guidelines.

TRANSPARENT PEER REVIEW

ORCID

Best regards,
Nina

Nina Vogt, PhD
Senior Editor
Nature Methods

Reviewer #1 (Remarks to the Author):

The manuscript is substantially improved compared to the initial submission. I have no further comments.

Reviewer #3 (Remarks to the Author):

The authors have addressed all my comments.

Reviewer #3 (Remarks on code availability):

The authors have provided a usable resource of the code with detailed documentation this time.

Version 4:

Decision Letter:

6th Oct 2025

Dear Selena,

I am pleased to inform you that your Article, "Neuroimaging connectivity analysis needs network science for brain-behavior linking", has now been accepted for publication in Nature Methods. The received and accepted dates will be October 30th, 2023 and October 6th, 2025. This note is intended to let you know what to expect from us over the next month or so, and to let you know where to address any further questions.

Over the next few weeks, your paper will be copyedited to ensure that it conforms to Nature Methods style. Once your paper is typeset, you will receive an email with a link to choose the appropriate publishing options for your paper and our Author Services team will be in touch regarding any additional information that may be required. It is extremely important that you let us know now whether you will be difficult to contact over the next month. If this is the case, we ask that you send us the contact information (email, phone and fax) of someone who will be able to check the proofs and deal with any last-minute problems.

Authors may need to take specific actions to achieve compliance with funder and institutional open access mandates.

If your research is supported by a funder that requires immediate open access (e.g. according to [Plan S principles](https://www.springernature.com/gp/open-science/plan-s-compliance) or the [NIH public access policy](https://www.springernature.com/gp/open-science/us-federal-agency-compliance)) then you should select the gold OA route, and we will direct you to the compliant route where possible. Because authors warrant under our subscription licensing terms that they haven't committed to licensing any version of their article under a licence inconsistent with the terms of our agreement – including the applicable embargo period – publication under the subscription model isn't suitable for authors whose funders require no embargo.

If you are active on Twitter/X or Bluesky, please e-mail me your and your coauthors' handles so that we may tag you when the

paper is published.

Best regards,
Nina

Nina Vogt, PhD
Senior Editor
Nature Methods

** Visit the Springer Nature Editorial and Publishing website at http://editorial-jobs.springernature.com?utm_source=ejP_NMeth_email&utm_medium=ejP_NMeth_email&utm_campaign=ejp_Nmeth or www.springernature.com/editorial-and-publishing-jobs for more information about our career opportunities. If you have any questions please click [here](mailto:editorial.publishing.jobs@springernature.com). **

Open Access This Peer Review File is licensed under a Creative Commons Attribution 4.0 International License, which permits use, sharing, adaptation, distribution and reproduction in any medium or format, as long as you give appropriate credit to the original author(s) and the source, provide a link to the Creative Commons license, and indicate if changes were made. In cases where reviewers are anonymous, credit should be given to 'Anonymous Referee' and the source.

Authors' Response to Reviewers' Comments:
“Inference-based statistical network analysis uncovers star-like brain functional architectures for internalizing psychopathology in children”
(NMETH-A54306)

We sincerely thank the Editor and reviewers for their comments and suggestions for our paper. We have thoroughly addressed all the comments and provided detailed point-by-point responses in this letter.

Reviewer 1

In this paper, Wang et al. introduces a method, called latent variable-based statistical network analysis (LatentSNA), to improve the statistical power for detection imaging biomarkers. The method is applied to model multi-state functional networks with multivariate internalizing profiles using of a few thousand children in the Adolescent Brain Cognitive Development (ABCD) study. The approach was demonstrated to yield markedly improved statistical power in detecting biomarkers and the latent variables were shown to explain the internalizing profiles better.

We really appreciate Reviewer 1 for the comments and for taking the time to review our work. We have made changes according to your suggestions and our response to each comment is as follows.

1. My main concern is with the lack of focus of the paper. What is the main contribution? LatentSNA or the neuroscientific findings associated with internalizing psychopathology?

If the main contribution is LatentSNA, the method has not been shown to be generalizable. The authors claimed: “Our approach can be used to detect other types of biomarkers using positron emission tomography, T1-weighted structural MRI and Diffusion Tensor Imaging, etc. Our approach can also detect imaging biomarkers of other behavioral, cognitive and psychopathology measures.” However, judging based on the limited results presented, these claims are at best unsubstantiated.

If the contribution is on the neuroscientific findings, the findings are overly narrow being focusing on internalizing psychopathology. Observations of star-like architectures are not necessarily generalizable to other conditions. The paper does not appear to be a good fit for Nature Methods.

Response: Thanks for the comment. We would like to confirm that we contribute to the current literature on brain connectivity-to-behavior modeling by proposing a novel generative Bayesian method that jointly models the correspondence between latent structures from brain connectivity and behavior profiles. The proposed method addresses the lack of power and inflated Type II errors in current analytic approaches in this field when detecting brain network neuromarkers, allows unbiased estimation of neuromarkers' relationship with behavior variants, quantifies the uncertainty and evaluates the likelihood of the estimated

neuromarkers’ effects against chance and ultimately improves brain-behavior prediction in novel samples and the clinical utilities of neuroimaging findings.

In response to the concern about generalizability, we have extensively evaluated our method under six new neuroimaging landmark studies with diverse populations. We outline the study cohorts and datasets in Table 1 and demonstrate the method’s generalizability to additional imaging biomarker types in Table 2 and to different outcome measures in Table 3. Collectively, we have fitted the method to 7 different datasets involving 8 different imaging modalities and 20 outcome measures with a total of 8,003 - 11,861 subjects’ information included. The results demonstrate that the proposed method is broadly applicable to a variety of imaging techniques including functional MRI (fMRI), T1-weighted structural MRI, Diffusion Tensor Imaging (DTI) and positron emission tomography (PET); and it is generalizable to different types of outcome measures, which include but are not limited to cognition, emotion, assessment of mental disorders and focal tau PET SUVR metrics. We use disorder outcomes to represent scales measuring psychopathological and psychological symptoms though many disorder symptoms are malfunctions in one and more cognitive and emotional dimensions. The proposed method shows consistent improvement in the model fit across data situations against available alternatives with satisfactory prediction accuracy for a variety of outcomes using functional and structural imaging modalities in developing, aging and transdiagnostic populations.

In Table 1, we outline study cohorts and datasets included in the study: Alzheimer’s Disease Neuroimaging Initiative Grand Opportunities and ADNI Phase 2 (ADNI-GO/2), ADNI Phase 3, Anti-Amyloid Treatment in Asymptomatic Alzheimer’s Disease (A4), Human Connectome Project Aging (HCP-A), Adolescent Brain Cognitive Development Baseline (ABCD-B) and the 2-year follow-up (ABCD-2) and Transdiagnostic data collected at Yale. We fit the model to each imaging modality and outcome measure outlined in Table 2 and Table 3. We focus on cognition outcomes that are often used to assess the performance of novel methods, emotion outcomes that are closely aligned with the internalizing outcomes in the original manuscript such as depression and anxiety, disorder and focal tau PET SUVR outcomes that directly reflect biological changes in the brain.

Table 1: Study cohorts and data

Data	Outcome				Neuroimaging			Sample Size
	Cognition	Disorder	Emotion	Tau	Structural ^a	fMRI/R ^b	fMRI/T ^c	
ADNI-GO/2	✓	✓			✓			410
ADNI-3	✓	✓				✓		174
A4	✓			✓		✓		394
HCP-Aging	✓		✓			✓		529
ABCD-B			✓			✓	✓	4,871-7,606
ABCD-2			✓			✓	✓	1,435-2,558
Transdiagnostic		✓					✓	190
Total	✓	✓	✓	✓	✓	✓	✓	8,003 - 11,861

^aThree types of structural imaging information are used including the fiber density, the number of fibers and fiber length. ^bfMRI collected when subjects are asked to rest. ^cfMRI collected when subjects are asked to perform cognitive and emotional tasks. We include emotional n-back task (EN-back), the Stop Signal task (SST) and the Monetary Incentive Delay (MID) task conditions.

Description of Data

We now provide details about the neuroimaging data and outcome measures for each study. For the ABCD study, the neuroimaging and internalizing outcomes have been discussed in the original submission, and are thus excluded here.

ADNI

Data used in the preparation of this article were obtained from the Alzheimer’s Disease Neuroimaging Initiative (ADNI) database (adni.loni.usc.edu). The ADNI was launched in 2003 as a public-private partnership, led by Principal Investigator Michael W. Weiner, MD. The primary goal of ADNI has been to test whether serial magnetic resonance imaging (MRI), positron emission tomography (PET), other biological markers, and clinical and neuropsychological assessment can be combined to measure the progression of mild cognitive impairment (MCI) and early Alzheimer’s disease (AD).

sMRI and DTI. We downloaded the T1-weighted structural MRI (sMRI) and Diffusion Tensor Imaging (DTI) from the ADNI-GO/2 databases from 174 subjects. We applied an overcomplete local principal components analysis (Manjón et al., 2013) to process DTI data following standard steps including denoising, motion-correction and distortion-correction. We performed the probabilistic white matter fiber tractography using the fiber assignment by continuous tracking (Moore & Sciacca, 2019). We registered sMRI scans to the lower resolution b0 volume of the DTI data using the FLIRT toolbox in the FMRIB Software Library (Jenkinson et al., 2012), and we then defined the cortical ROIs in the FreeSurfer space using the Lausanne 2008 parcellation with 68 cortical regions of interest (Cammoun et al., 2012). We obtained the number of the fibers (NOF) connecting each pair of ROIs as well as the regions’ surface area (SA). The fiber density-based structural connectivity was calculated by dividing NOF between two ROIs with their average surface areas (Xu et al., 2022; Yan et al., 2020). Three types of structural brain networks were constructed as the number of fibers between a pair of brain regions, the length of the fibers as well as the fiber density of tracts connecting pairs of ROIs.

fMRI. We used resting state functional neuroimaging data from the third release of the ADNI study. We processed the images using the ConnectomeMapper3 pipeline (Touber et al., 2022) built in Nipype (Gorgolewski et al., 2011). The resting-state fMRI images were processed with despiking and slice timing correction following Cox (1996); the images also were motion-corrected and distortion-corrected using FSL. The rs-fMRI images were registered to the b0 sMRI using the FLIRT toolbox (Jenkinson et al., 2002; Jenkinson & Smith, 2001). The BOLD time signals of each ROI were band-pass filtered and then detrended using a linear regression. We constructed functional brain networks for each subject as the Pearson correlation between the BOLD time signals for pairs of ROIs.

Disorder/Cognition Outcomes. For outcomes, we included the Alzheimer’s Disease Assessment Scale, cognitive subscale (ADAS, Kueper et al., 2018), a rating of dysfunction by AD. We included the ADAS score as the sum of 13 diagnostic questions collected at baseline. In addition, we included the sum score of the Everyday Cognition Scale (Ecog, Farias et al., 2008), a questionnaire measuring the patient’s cognitive function, which is often used to aid the diagnosis of dementia and other neurodegenerative aging-related disorders. We applied the proposed LatentSNA to both ADAS and Ecog to assess the model’s generalizability to alternative outcomes for the aging population.

A4

Anti-Amyloid Treatment in Asymptomatic Alzheimer’s Disease (A4) study is a secondary prevention trial targeted towards older people with amyloid accumulation and are at high risk for AD dementia (Sperling et al., 2014).

Table 2: Prediction accuracy in independent samples with different types of imaging biomarkers

Biomarker Type	Outcome	Method				
		CPM	rCPM	GC	TNFA	LatentSNA
Fiber Density	ADAS	0.223	0.256	0.238	0.265	0.646
	ECog	0.234	0.176	0.212	0.362	0.572
# Fibers	ADAS	0.226	0.381	0.692	0.297	0.692
	ECog	0.256	0.258	0.326	0.448	0.663
Fiber Length	ADAS	0.421	0.236	0.265	0.366	0.681
	ECog	0.242	0.230	0.221	0.336	0.659

sMRI and fMRI. For the fMRI data, we used the same processing procedure as that for the ABCD. MPRAGE scans were skull stripped using optiBET (Lutkenhoff et al., 2014) and nonlinearly aligned to the MNI-152 template using BioImage Suite.

Focal tau PET SUVR metrics. We used the PETSURFER within FreeSurfer for an integrated MRI-PET analysis (Greve et al., 2016; Greve et al., 2014). We derived the focal tau PET (^{18}F]flortaucipir) SUVR metrics from the A4 images using 90-110 minute (4x5-minute frames) post-injection images, preprocessed and analyzed using PETSURFER in FreeSurfer (v 6.0+). We summed and motion-corrected the five-minute tau PET frames. We then aligned the composite PET images to corresponding MRI images, parcellated using the Desikan-Killiany Atlas (DSK Desikan et al., 2006), and partial-volume corrected using FreeSurfer. We gathered the average tracer absorption values for each region defined by the atlas and computed standardized uptake value ratios (SUVR) using the whole cerebellar cortex as the reference region.

Cognition Outcome. To assess cognition changes, we included the Preclinical Alzheimer’s Cognitive Composite (PACC, Donohue et al. (2014)) collected as part of the A4 project. PACC is a composite cognitive score combining tests that assess episodic memory, executive function and general cognition, and it is the primary outcome measure for A4 targeting the preclinical AD population. PACC is found to be sensitive to the earliest disease-related change (Papp et al., 2021).

HCP-Aging

The Lifespan Human Connectome Project in Aging (HCP-Aging) aims to characterize how brain organization and connectivity changes during typical aging, compared to an ‘abnormal’ aging process, a potentially distinct but not inevitable condition of aging (Bookheimer et al., 2019).

fMRI. For the fMRI data, we used the same processing procedure as that for the ABCD.

Emotion/Cognition Outcomes. For the HCP-Aging project, we focused on cognition and emotion measures. To assess the cognitive capability of the healthy aging population, we included the composite scores for the Picture Sequence Memory Test as well as the Cognition Composite score including Fluid Composite and Crystallized Composite, derived from all National Institutes of Health Toolbox Cognition tasks (NIH TB-CT, Weintraub et al., 2013). For the emotion outcomes, we chose the Emotional Distress Depression and the PROMIS Anxiety to maintain relative consistency with the internalizing outcome. The Emotional Distress Depression is captured by the Sadness Survey, from National Institutes of Health Toolbox Emotion Battery (NIH TB-EB, NIH (2022)), which measures the negative mood and perceptions. The PROMIS Anxiety is captured by Fear-Affect Survey, a self-report measure assessing fear and anxious misery from NIH TB-EB.

Table 3: Prediction accuracy in independent samples with different types of outcomes

Outcome Measure Name	Source	Condition	Method					
			CPM	rCPM	GC	TNFA	LatentSNA	
Cognition	Ecog	ADNI-3	Rest	0.245	0.301	0.304	0.341	0.551
	PACCRN	A4	Rest	0.259	0.270	0.232	0.228	0.543
	Picture Sequence Memory Cognition Composite	HCP-Aging	Rest	0.331 0.273	0.381 0.610	0.363 0.394	0.247 0.332	0.663 0.615
Disorder	BSI	Transdiagnostic	nBack	0.099	0.143	0.032	0.194	0.494
	ADAS	ADNI-3	Rest	0.167	0.276	0.217	0.255	0.492
Emotion	Internalizing	ABCD-2	Rest	0.091	0.200	0.102	0.267	0.897
			nBack	0.153	0.162	0.080	0.156	0.774
			MID	0.124	0.273	0.361	0.276	0.721
			SST	0.083	0.229	0.099	0.239	0.692
	Emotional Distress Depression PROMIS Anxiety	HCP-Aging	Rest	0.416 0.249	0.302 0.285	0.198 0.386	0.286 0.222	0.658 0.650
Tau	Inferior Temporal	A4	Rest	0.211	0.349	0.276	0.284	0.599
	Middle Temporal			0.325	0.275	0.227	0.190	0.652
	Parahippocampal			0.254	0.141	0.229	0.222	0.624
	Fusiform			0.366	0.206	0.314	0.428	0.663
	Entorhinal			0.274	0.242	0.299	0.348	0.642
	Precuneus			0.303	0.208	0.271	0.369	0.653
	Supramarginal			0.327	0.317	0.288	0.299	0.613
	Posterior cingulate cortex			0.298	0.354	0.290	0.176	0.588
	Rosteral anterior cingulate			0.254	0.275	0.271	0.236	0.611
	Caudal anterior cingulate			0.270	0.274	0.225	0.243	0.569
Amygdala	0.252	0.200	0.170	0.263	0.588			

Transdiagnostic

The Transdiagnostic project aims to recruit clinically naturalistic and demographically diverse subjects to more effectively study the links between imaging and behaviors (Greene et al., 2022); the project was conducted at Yale between February 2018 and March 2021. Participants of the Transdiagnostic project tend to show a wide range of symptom severity and commonly have multiple psychiatric diagnoses. All imaging information was collected at the Yale Magnetic Resonance Research Center.

fMRI. The data preprocessing for the fMRI of the Transdiagnostic project is the same as the processing of the fMRI for the ABCD study.

Disorder Outcomes. We included the global severity index of the Brief Symptom Inventory (BSI, Derogatis, 1975), a rating scale aiming to identify clinically relevant psychological symptoms in adolescents and adults. The global indices measure the level of symptomatology, its intensity and number of occurrence.

Results

Results of the added imaging and outcomes are shown in Table 2 and Table 3. In Table 2, we show the method’s generalizability to detect other types of imaging biomarkers besides functional. In particular, in Table 2, we include structural imaging modalities measuring fiber density, number of fibers and fiber length. Note that we also included tau metrics from PET imaging from the A4 study in Table 3. We report the average correlation between the predicted and observed outcomes in independent samples across 10 runs with the same procedure as before. The results show that using LatentSNA, we can obtain satisfactory prediction accuracy across different imaging modalities with consistent improvement over existing methods. See details about the comparison methods in the response to comment 2, reviewer 1.

In Table 3, we show the method’s generalizability to different outcomes. In particular, we include 4 types of outcomes with 20 outcome variables focused on cognition, disorder, emotion and focal tau PET SUVR metrics. The results show that using LatentSNA, we can obtain satisfactory prediction accuracy across different outcome measures, and we consistently improve prediction accuracy of existing methods. More specifically, we see larger improvements in predicting emotion outcomes based on fMRI data in the children’s population, which corresponds with existing literature showing difficulties in phenotype prediction using children’s fMRI data given the high heterogeneity and strong motion artifact (Calhoun & Sui, 2016; Chahal et al., 2020; Greene et al., 2022). The predicability of the focal tau PET SUVR metrics is consistently higher than other types of outcomes. This result is as expected because the tau PET SUVR metrics, derived from the A4 images using post-injection images, more directly reflect brain changes than behavior outcomes.

2. The current paper compares LatentSNA only with some of the most basic methods. There are plenty of latent-variable graphical models that are applicable to brain network analysis. Even dynamic causal modeling methods can be considered as based on latent models. A common trait of these methods, like LatentSNA, is that they represent complex signals in a low-dimensional space encoded by latent variables. A significant amount of work is needed to evaluate LatentSNA against the true state of the art.

Response: We really appreciate the reviewer for raising this interesting point. In response to the concern about insufficient evaluation of the proposed method against existing alternatives, we have provided strong evidence demonstrating the proposed model’s superiority over 6 existing methods including 2 network-based brain analysis methods and 4 popular brain-behavior linking approaches. We have evaluated the proposed method against the penalized graph classification (GC) approach, tensor network factorization

analysis (TNFA), connectome-based predictive modeling (CPM), and machine learning approaches including ridge connectome-based predictive model (rCPM), least absolute shrinkage and selection operator (LASSO) and canonical-correlation analysis (CCA). In data situations with different imaging modalities, outcome measures and populations, our method consistently outperforms existing alternatives showing, for example, an average of 110% improvement over TNFA and an average of 150% improvement over CPM. The results show that the proposed LatentSNA is a valid and significant adaptation of social network analysis (SNA) concepts and methods for real-world networks and brain and behavior linking.

Directly addressing reviewer’s concern, we have added model evaluation results against two network-based brain analysis methods, the penalized graph classification (GC) approach (Reli3n et al., 2019) and the tensor network factorization analysis (TNFA). The GC approach uses the brain connectivity as predictors and adopts both L1 penalty, the absolute value of coefficient magnitudes and a generic group lasso penalty. We fitted the GC approach using the graphclass R package. The tuning of the penalty factor pair (λ , ρ) is conducted on a 3x4 grid, with λ selected from the set $\{10^{-6}, 10^{-5}, 10^{-4}\}$ and $\rho \in \{1, 10, 20, 30\}$. It is observed that a λ value exceeding 10^{-3} and a ρ value surpassing 40 result in the penalization of all coefficients to zero. For the TNFA approach, similar to the tensor network principal components analysis method (Zhang et al., 2019), we embedded the $V \times V$ symmetric adjacency matrices into a low dimensional matrix; each row contains subjects’ principal component scores and each column contains basis network; only significant basis networks are included as predictors. We then perform a network predictor regression with the embedded low-dimensional basis networks as predictors of the outcome variables.

In Table 2 and Table 3, we show the prediction accuracy of LatentSNA against GC, TNFA, CPM and rCPM. The results show that LatentSNA shows superior prediction accuracy for independent samples across a variety of imaging modalities and outcome variables. Larger improvement in predicting emotion outcomes are found with fMRI data in the children’s population, which suggests that the proposed method is particularly advantageous for heterogeneous and noisy data situations.

In addition to expanding the pool of alternative methods to demonstrate the proposed method’s superiority, we would like to provide detailed discussions about the method’s innovation and highlight the difference between LatentSNA and latent-variable graphical models. We would like to clarify that *the proposed method is a first attempt at extending social network analysis (SNA, Wasserman & Faust, 1994) to jointly model functional connectomes and continuous behavior outcomes with substantially improved power to detect region-specific imaging biomarkers*. In modeling objectives and approaches, the LatentSNA method is different from latent-variable graphical models (GMs).

The proposed LatentSNA uniquely contributes to the current literature of social network analysis (SNA) methods. SNA methods are used to understand the observed networks’ structure and interaction patterns, which include but are not limited to social relationships (e.g., Friel et al., 2016; Gollini & Murphy, 2016; Minhas et al., 2019; Wang & Edgerton, 2022). Information about connectivity (edges) between nodes is collected, often visualized and analyzed using mathematical and statistical methods. The interconnectedness (dependencies) of edges is a key network feature and a major hurdle for network modeling. Notable examples of statistical approaches for network modeling include the exponential random graph models (ERGMs, Robins et al., 2007; Wasserman & Pattison, 1996), stochastic block models (SBMs, Airoldi et al., 2008; Holland et al., 1983) and latent space models (LSMs, Hoff et al., 2002). SNA methods differ in how dependencies among the edges are addressed. Using ERGMs, researchers assume that dependencies among the edges can be accounted for by descriptive features of the network, e.g., nodes’ differences in popularity can be accounted for by node degrees. In other words, in ERGMs, researchers explicitly define dependent features in a network and estimate the impact of network features on the likelihood of connectivity to infer what drives relational patterns. Latent variables-based network models, on the other hand,

enforce natural network behaviors without explicit specification.

The proposed LatentSNA is a novel extension of the latent variables-based network models for imaging biomarker detection. Latent variables-based network models, also called the conditionally independent dyadic (CID) network models (Sweet, 2016), account for dependencies among connectivity edges with latent variables—in this way, latent variables in SNAs are different from latent variables in GMs. These models assume conditional independence among the edges—that the edge between brain region A and brain region B is conditionally independent of other edges, e.g., between A and C, between B and C—conditional on the latent variables. The proposed LatentSNA builds on existing CID methods and assumes conditional independence among the connectivity edges; in addition, it assumes conditional independence among outcome variables, similar to the psychometric Rasch model, and most importantly, it assumes conditional independence between brain connectivity and outcome variables. *The LatentSNA is a first SNA method that jointly accounts for dependencies among connectivity edges, among outcome variables and between brain connectivity and outcomes.*

In the following paragraphs, we outline the specific innovations of the LatentSNA to existing state of the art SNA methods as well as network-based brain analysis methods. While the current SNA methods mentioned above focus on modeling single networks, brain connectivity networks can be seen as multiplex networks with multiple layers of brain connections observed on a shared set of brain regions. To model the multiplex structures, Gollini and Murphy (2016) and D’Angelo et al. (2018) propose a shared set of latent variables across layers and assume a joint relational structure across sets of connectivity. Salter-Townshend and McCormick (2017) and MacDonald et al. (2022) separate the shared and individual components across layers. Different from these approaches, we capture the individual differences in brain connectivity networks across layers and identify specific brain regions, where the covariation between layers of brain networks and the outcome variables is significant.

The LatentSNA uniquely contributes to existing network-based brain analysis literature. To identify links between brain networks and outcomes, existing network-based brain analysis methods commonly incorporate individual behaviors and attributes as predictors and model the effect of individual attributes on brain networks by estimating the regression coefficient associated with the predictors on the edge probabilities of brain networks, called the network response regression (e.g., Simpson et al., 2024; Wang et al., 2017; Xia et al., 2020; Zhou & Müller, 2022). Alternatively, researchers use brain connectivity networks as the predictors (e.g., Guha & Rodriguez, 2021) and use matrix and tensor operations (e.g., Wang et al., 2021) to transform the networks into a linear term (e.g., Zhao et al., 2022)—called the scalar-on-network regression (Morris et al., 2022). Different from the network response and the scalar-on-network regressions, the LatentSNA theorizes a shared data generation process connecting the brain connectivity and outcomes.

With a shared data generation process for connectivity and outcomes, the LatentSNA poses a few advantages over the network response and the scalar-on-network regressions. First, the relationships between the brain and individual behaviors and outcomes are modeled as one-directional in regression models: estimated as the impact of brain on behavior or the impact of behavior on brain when in fact, the relationship is likely mutual. While changes in the brain are often associated with changes in behavior; given neuroplasticity, it is reasonable to assume that disordered behaviors and dysfunctional environment can lead to further deterioration in the brain. Second, by using the brain connectivity (behavior outcome) as predictor in the scalar-on-network (network response) regression, we assume that the brain connectivity (behavior outcome) is fully observed, which is problematic when we have partially missing brain connectivity and outcome observations. Regression methods are not able to accommodate data situations, where we have partially missing observations for both brain connectivity and individual outcomes. Lastly, regression methods do not gain strength for estimating parameters modeling brain connectivity (behavior) given that

the behavior (brain connectivity) is not modeled. In sum, we overcome these limitations by proposing a shared data generation process for multiplex brain networks and individual outcomes and allow the brain and behavior to mutually inform each during model estimation.

The LatentSNA represents a novel implementation of network science in a generative statistical process by innovating SNA concepts and methods for real-world networks and brain and behavior linking. In modeling objectives and approaches, the LatentSNA method is different from GMs. A GM is used to represent joint probability distribution and construct conditional dependence between random variables (Edwards, 2000; Lauritzen, 1996; Murphy, 2001). Using GMs, researchers construct a graph (network) based on whether or not two variables are conditionally independent of each other. The edge in a brain functional network for LatentSNA represents whether/how strongly two brain regions co-activate across time; in contrast, the edge in the covariance network for GMs represents whether/how strongly two variables are conditional dependent. Crucially, the covariance networks in GMs are created out of the positive definite covariance matrices; their relationships are of a certain type, and they are more specific than the real-world observed networks including functional brain networks. In contrast to GMs, SNAs accommodate different types of real-world network structures, structures that are impossible to have for covariance networks in GMs. Therefore, GMs and SNAs are not comparable; latent variables in SNAs have different functions than latent variables in GMs. See further discussions about the differences between the two in Steinley (2021), Vitale et al. (2022), and Wang (2021).

3. The authors claimed that LatentSNA uses “the symmetric bilinear interaction effect to capture possible higher-order dependencies, among three or more connectivity edges.” However, Equation 0.1 seems to only be modeling the interaction between two nodes. The effect of this interaction is also not validated. What if this term is omitted from the model? How much degradation in performance, if any, will it cost. Is it possible to go beyond modeling bilinear interactions? How would modeling even higher-order dependencies affect the performance?

Response: Thank you for your comments. In response to the concern about the model’s capability to capture network topological structures and higher-order dependencies, we would like to make two follow-up points. First, we would like to provide an example disproving the claim that Equation 0.1 only models pairwise interactions. Second, we would like to provide detailed discussions about higher-order dependencies and emphasize that third-order dependence structures such as balance and clusterability contribute to relational patterns across the whole network, not just in triads.

First, we would like to confirm that the proposed LatentSNA method models higher-order dependencies and the interconnectedness among three and more brain regions. In Equation 0.1, we specify the probabilistic structure of the observed connectivity between pairs of brain regions u and v , $u < v$, $u, v = 1, 2, \dots, V$, following the standard practice to specify the probabilistic structure for each observation in a statistical model. This formulation is equivalent to the probabilistic structure of the whole network:

$$\mathbf{X}_i = a_i \mathbf{1}^T \mathbf{1} + \mathbf{w}_i \boldsymbol{\beta} \mathbf{1}^T \mathbf{1} + \mathbf{z}_i \mathbf{z}_i^T + \mathbf{E}_i, \quad (1)$$

where \mathbf{X}_i is the $V \times V$ brain connectivity adjacency matrix for person i ; $\mathbf{1}$ is a $V \times 1$ column vector of 1s; a_i is the fixed connectivity intercept for subject i ; \mathbf{z}_i is the $V \times 1$ vector of latent variable values for subject i ; and \mathbf{E}_i is the $V \times V$ matrix of errors. We adjusted for Q covariates, denoted by \mathbf{w}_i with the first element to be 1 corresponding to the intercept with their effects on the connectivity matrix characterized by $\boldsymbol{\beta}$. In sum, the formulation in Equation 0.1 represents the probabilistic structure for the whole network accounting for topological structures, relational patterns across the whole network.

We would like to show that Equation 0.1 models dependencies in the triangular structure between nodes a , b , and c , shown in Figure 1 in the appendix, which contradicts the idea that Equation 0.1 only models pairwise interactions. The triangular structure signals the presence of third-order dependence (Wasserman & Faust, 1994)—the connectivity edge between brain regions a and b , between brain regions a and c and between brain regions b and c are related through their share of the triad, regions a , b and c . Following Equation 0.1, the probability of edge between brain regions a and b depends on z_a and z_b ; and the probability of connectivity between brain regions a and c depends on z_a and z_c . The probabilities of these two edges are not independent to the probability of the edge between brain regions b and c as z_b and z_c contribute to the probability of connectivity between brain regions b and c as well as connectivities between brain regions a and b and between brain regions a and c . In other words, the probabilities of connectivities among brain regions a , b and c are related to each other, and this relatedness is reflected by the model in Equation 0.1.

In the above, we have provided a simplified illustration contradicting the idea that Equation 0.1 only models pairwise interactions. In the following, we would like to provide detailed discussions about higher-order dependencies and bilinear effects. In Wasserman and Faust (1994), higher-order dependencies in real-world networks are defined as dependencies among three (triad) or more nodes. Common examples of higher-order dependencies in real-world networks include homophily and balance and clusterability (Heider, 1944; McPherson et al., 2001; Norman et al., 1965; Wasserman & Faust, 1994). Homophily is often associated with the transitive property of a network, how new connectivities are established upon existing connectivities, also known as transitivity. Balance suggests a state of harmony, where positive connectivities are found among nodes with similar attributes, and the negative connectivities are found among nodes with divergent attributes. Clusterability represents a state of harmony with a more relaxed criteria than balance (Hoff, 2005, see examples of balanced and clusterable triads in Figure 2A). With balanced cycles among triads, the whole network can be divided into cohesive groups with $x_{u,v} > 0$ if u and v are in the same group and $x_{u,v} < 0$ if they are in opposite groups (Harary, 1953; Hoff, 2005). Therefore, the presence of higher-order dependencies such as balance contribute to relational patterns and topology across the whole network—even higher-order dependencies. By modeling higher-order dependencies, the proposed LatentSNA captures the relational patterns across the whole network.

Bilinear effects account for transitive, balanced and clusterable network structures (Hoff, 2005, 2008). Bilinear effects are theoretically and empirically tested to capture higher-order dependencies in dependent networks following the existing SNA literature. In Hoff (2008), the author provides the theoretical evidence that vector-product based latent space models, which include bilinear effects models, capture higher-order dependencies such as homophily, balance and clusterability; and in Sosa and Buitrago (2021), the authors further demonstrate that such models show satisfactory model fit for networks with varying degrees of transitivity and clusterability. Given that brain functional networks are known to possess small world properties (Bassett & Bullmore, 2006) likely exhibiting both transitivity and clusterability. It is thus optimal for us to use the bilinear effects to model higher-order dependence structures.

4. The code is not accessible, making it more challenging to assess the validity of the method.

Response: Thank you for your comment. We have modified the published code, and shared the repository via the user-friendly github at <https://github.com/selenashuowang/LatentSNA>.

5. Figure 1 is poorly done. It does not clearly explain the core concepts of LatentSNA. Notations are not defined, making it impossible to make any sense out of the figure.

Response: Thank you for your comment. Following your comment, we have revised Figure 1 of the manuscript (Figure 2 in the letter appendix), added labels for key components of the schematic diagram and provided explanations for each component in the caption. We have added the following explanations for the figure. In the left panel, the individual functional networks and internalizing psychopathology measures are used as inputs to obtain region-specific covariance between connectivity and internalizing measures. In the right panel, we show examples of balanced and clusterable triads (A) and the diagram of LatentSNA model (B). The diagram shows that with the input of functional connectivity matrices, $\{\mathbf{X}_1, \mathbf{X}_2, \dots, \mathbf{X}_N\}$ and internalizing measures, \mathbf{Y} , we propose a joint latent variable model, where we allow the latent connectivity variables, \mathbf{Z} and latent behavior variables, Θ to co-vary with a shared covariance matrix, Σ in the generative Bayesian framework. Intercepts a and error term E are part of the data generation process for connectivity; and intercepts b and error term Ψ are part of the data generation process for behavior. Circles represent unknown quantities of the model, and squares represent the observed data. Non-informative priors are not included in this diagram. In the right panel section C, we show estimated internalizing network via marginal test (top, blue), estimated internalizing network via LatentSNA (bottom, blue) and a theoretical star-like structure in red.

6. On page 9 it is unclear what is meant by “the Averaging method simply takes the sample average connectivity as a prediction for a new subject’s connectivity.”

Response: Thank you for your comment. Following your comment, we would like to detail the procedure to predict connectivity using the Average method in independent samples. The Average method and its extensions represent one of the most common methods to capture group-level connectivity and to perform subsequent analysis (Achard et al., 2006; Sinke et al., 2016; Song et al., 2009) often with satisfactory prediction accuracy (e.g., Paquette et al., 2016). We first randomly divide the connectivity data into ten equal sizes, using one set of data points as the test data and other sets of data points as the training data. We then capture the group-level connectivity using the entry-wise sample mean of individual connectivity matrices in the training data. We perform predictions for connectivity in the test set using estimated connectivity from the training data. Figure 3A in the supplementary materials shows the average out-of-sample correlations between the observed and predicted connectivity values across 100 random samples. Our results suggest that LatentSNA shows satisfactory prediction accuracy for the brain connectivity using individual-level estimates, and it outperforms the Average method when the signal proportion is large. When predicting connectivity using the group-level estimates, LatentSNA and the Average method both show satisfactory performance for the whole graph; and LatentSNA outperforms the Average method for regions with strong relational ties with behavior.

Reviewer 2

A. The LatentSNA framework proposed in this paper combines brain functional connectivity with internalizing psychopathology. In particular, the proposed model was applied to the ABCD study, which achieved sufficiently accurate prediction of both children’s internalizing traits and functional connectivity, and substantially improved the ability to explain the individual internalizing differences compared with current approaches. B. This original work is of great significance, especially in detecting the neurologically meaningful network topology in the adolescents and children population. C. Both simulation studies and real applications to the ABCD study were conducted to demonstrate the outperformance of the proposed method. D. The Bayesian framework

and the proposed MCMC algorithm are solid and convincing. E. The findings provided by latentSNA are consistent with existing literature. G. Previous related works are mostly cited in this paper. H. The whole paper is well written and organized.

We really appreciate Reviewer 2 for the positive comments and for taking the time to review our work. We have made changes according to your suggestions and our response to each comment is as follows.

1. Figure 1 shows the big picture of the LatentSNA. However, it is not informative and is poorly illustrated and explained in the caption contents.

Response: Following your comment, we have revised Figure 1 of the manuscript (Figure 2 in the letter appendix), added labels for key components of the schematic diagram and provided explanations for each component in the caption. We have added the following explanations for the figure. In the left panel, the individual functional networks and internalizing psychopathology measures are used as inputs to obtain region-specific covariance between connectivity and internalizing measures. In the right panel, we show examples of balanced and clusterable triads (A) and the diagram of LatentSNA model (B). The diagram shows that with the input of functional connectivity matrices, $\{\mathbf{X}_1, \mathbf{X}_2, \dots, \mathbf{X}_N\}$ and internalizing measures, \mathbf{Y} , we propose a joint latent variable model, where we allow the latent connectivity variables, \mathbf{Z} and latent behavior variables, Θ to co-vary with a shared covariance matrix, Σ in the generative Bayesian framework. Intercepts a and error term E are part of the data generation process for connectivity; and intercepts b and error term Ψ are part of the data generation process for behavior. Circles represent unknown quantities of the model, and squares represent the observed data. Non-informative priors are not included in this diagram. In the right panel section C, we show estimated internalizing network via marginal test (top, blue), estimated internalizing network via LatentSNA (bottom, blue) and a theoretical star-like structure in red.

2. More discussions about the differences between LatentSNA (Z) and LatentSNA (THETA) are expected.

Response: Thank you for your comment. Following your suggestion, we include more discussions about the difference between LatentSNA (Z) and LatentSNA (THETA). The difference between LatentSNA (Z) and LatentSNA (THETA) lies in how predictions are performed. For LatentSNA (THETA), predicting the psychopathology outcome of a new subject amounts to additional draws for each new y_i from a distribution with probability determined by the model. For LatentSNA (Z), on the other hand, predicting the psychopathology outcome of a new subject is based on the estimated latent connectivity variable \mathbf{Z} from the training data. We evaluate the out-of-sample predictive performance for LatentSNA (Z) and LatentSNA (THETA) as follows:

- We randomly sample 100 subjects and their psychopathology outcome as the test data, and other sets of data points as the training data.
- We fit the training data to LatentSNA, obtain the posterior mean of the model parameters.
- For LatentSNA (THETA),
 - Predicting the psychopathology outcome of a new subject amounts to additional draws for each new y_i from a distribution with probability determined by the model.
 - The full conditional of the new observations $\mathcal{Y}^{(test)}$ is, for any $y_i \in \mathcal{Y}^{(test)}$, determined by $\pi(y_i|\theta, b_i, \Psi_i)$.
- For LatentSNA (Z),

- Predicting the psychopathology outcome of a new subject is based on the estimated latent connectivity variable, $\hat{\mathbf{Z}}$ from the training data.
- We first select significant imaging biomarkers based on 95% posterior credible intervals of the covariance parameters and use significant imaging biomarkers’ latent connectivity variables as predictors.
- Second, we split the estimated latent connectivity variables into the test set, $\hat{\mathbf{Z}}^{(test)}$ and the training set, $\hat{\mathbf{Z}}^{(train)}$ following the split of the data.
- Third, we fit the training model using $\hat{\mathbf{Z}}^{(train)}$ as the predictors and the observed psychopathology outcomes for the training subjects as the response.
- We obtain the estimated regression coefficients, $\hat{\beta}$ based on the training model.
- Lastly, we predict the psychopathology outcome of a new subject, for any $y_i \in \mathcal{Y}^{(test)}$, under LatentSNA (Z) following $y_i = \hat{\beta} \times \hat{\mathbf{Z}}^{(test)}$.

We repeated the process 10 times. Figure 3A shows the out-of-sample correlations between the observed and predicted internalizing values on the test data using LatentSNA (Z) and LatentSNA (THETA). Between LatentSNA (Z) and LatentSNA (THETA), the former does not directly—but indirectly—incorporate training internalizing information for prediction while the latter does. This shows that by constructing joint learning mechanisms using LatentSNA, we can effectively predict internalizing profiles for new subjects based on the available data.

3. What kind of cross-validation approach is adopted for the prediction tasks?

Response: Thank you for your comment. Following your suggestion, we provide a detailed discussion about the out-of-sample prediction procedure for predicting $\mathcal{Y}^{(test)}$ in the previous response. In addition, we would like to provide a detailed discussion about the out-of-sample prediction procedure for predicting connectivity. We evaluate the out-of-sample performance for predicting connectivity of new subjects as follows:

- We randomly sample 100 subjects and their connectivity values as the test data, and other part of data points as the training data.
- We fit the training data to LatentSNA, obtain the posterior mean of the model parameters.
- Predicting the connectivity of a new subject amounts to additional draws for each missing $x_i \in \mathcal{X}^{(test)}$ from a distribution with probability determined by the model.
- The full conditional of the new observations $\mathcal{X}^{(test)}$ is, for any $x_i \in \mathcal{X}^{(test)}$, determined by $\pi(x_i | z_i, a_i, \mathbf{E}_i)$.

We repeated the process 10 times. Figure 3B shows the average out-of-sample correlations between the observed and predicted connectivity values on the test data for predicting the whole graph, top ten internalizing regions and top 5 internalizing regions. The results show that LatentSNA provides sufficiently accurate prediction of the connectivity measurements, posing a unique opportunity to uncover brain connectivity for new subjects incorporating their internalizing measures.

4. I would like to see the sensitivity of the prior settings in the MCMC algorithm.

Response: Thank you for your comment. We would like to provide a detailed discussion about the sensitivity of the prior for the covariance matrix Σ . The identification of the imaging biomarkers is based

on whether the estimated covariances between connectivity and behavior are significantly different from zero. Therefore, it is of interest to expand on the sensitivity of the prior specification of the covariance parameters.

In LatentSNA, the approximation of the posterior distribution of Σ is facilitated by setting a prior distribution of inverse Wishart($\mathbf{I}_{V+1}, V+1+2$) with identity scale matrix, $\mathbf{S}_0 = \mathbf{I}$ and degree of freedom equaling to $m_0 = V + 1 + 2$. The use of an inverse Wishart (IW) distribution as a prior for the variance-covariance parameter matrix is fairly common in Bayesian analysis; see discussions in Barnard et al. (2000), Leonard and Hsu (1992), and Zhang (2021). The IW prior is a conjugate prior for the covariance matrix of the normal data. In LatentSNA, we are interested in estimating the covariance matrix Σ of the joint distribution of the latent connectivity and behavior variables, $\mathbf{D} = (z_{1,i}, z_{2,i}, \dots, z_{V,i}, \theta_i)$. With the IW prior, the posterior distribution of Σ can be obtained through the Bayes' Theorem:

$$p(\Sigma|\mathbf{D}) = \frac{p(\mathbf{D}|\Sigma)p(\Sigma)}{p(\mathbf{D})}. \quad (2)$$

From it, we can obtain the posterior distribution of Σ with the specified prior distribution as

$$\Sigma|\mathbf{D} \sim IW(\mathbf{S}_0 + \mathbf{F}'^T \mathbf{F}', m_0 + 2), \quad (3)$$

where \mathbf{F}' as a $N \times (V + 1)$ matrix with i th row as (z_i^T, θ_i^T) . Therefore, the posterior mean of Σ is a weighted average of the sample covariance matrix $\mathbf{F}'^T \mathbf{F}'$ and the prior mean \mathbf{S}_0 . When the sample size $N \rightarrow \infty$, the posterior mean approaches the sample mean.

In a sensitivity analysis conducted by Zhang (2021), the author sets the scale matrix as identity and vary the degrees of freedom by increasing m_0 . With the increase of m_0 , the posterior means become smaller and the posterior variances also become smaller. Thus, given the large sample size in the data, we expect the posterior mean of Σ to approach the sample mean.

5. The mechanism for generating simulation data is not very clear.

Response: Thank you for your comment. Following your suggestion, we would like to provide a detailed discussion about the data generation process for the simulation. The data are generated as follows. For simplicity and consistency, the number of behavior variables are assigned as one in all generated data. We first generate the connectivity latent variables as well as the latent behavior variables from the multivariate normal distribution with the mean zero and pre-defined covariance matrix with unit variances. To conduct a comprehensive assessment of the model performance, we create a range of data situations with varying sample sizes, connectivity scale, signal to noise ratio and signal proportions. To assess the model's ability to accurately identify true imaging biomarkers for outcomes that have both strong and weak biological signals, we vary the amount of true signals in the data by assigning signal proportion to 0.1 and 0.3. When the signal proportion equals to 0.1(0.3), we randomly assigned 10% (30%) of the covariance parameters between connectivity and psychopathology to be non-zero. To ensure the positive definiteness of Σ , we assigned both the covariances between connectivity and psychopathology and the corresponding dimensions in the latent connectivity covariance matrix as 0.9. We randomly sampled the errors for the connectivity from a normal distribution with mean 0 and variance defined by the signal-to-noise (S/N) ratio. The errors for the psychopathology are sampled from the normal distribution with the mean 0 and variance 0.5.

We consider three sample sizes, $N = 500$, $N = 1,000$ and $N = 2,000$ and two conditions for the number of nodes $V = 20$ and $V = 70$, and we specify two levels of the S/N ratio, 0.5 and 1 controlled by the error variance while keeping the variance of the latent variables constant. The individual-specific intercepts for

connectivity and behavior are set as 0. In total, we consider 24 different scenarios combining from different signal proportions, sample sizes, node numbers and S/Ns. Under each scenario, we simulate the 100 data.

References

- Achard, S., Salvador, R., Whitcher, B., Suckling, J., & Bullmore, E. (2006). A resilient, low-frequency, small-world human brain functional network with highly connected association cortical hubs. *Journal of Neuroscience*, *26*(1), 63–72.
- Airoldi, E. M., Blei, D. M., Fienberg, S. E., & Xing, E. P. (2008). Mixed membership stochastic blockmodels. *Journal of Machine Learning Research*, *9*(Sep), 1981–2014.
- Barnard, J., McCulloch, R., & Meng, X.-L. (2000). Modeling covariance matrices in terms of standard deviations and correlations, with application to shrinkage. *Statistica Sinica*, 1281–1311.
- Bassett, D. S., & Bullmore, E. (2006). Small-world brain networks. *The neuroscientist*, *12*(6), 512–523.
- Bookheimer, S. Y., Salat, D. H., Terpstra, M., Ances, B. M., Barch, D. M., Buckner, R. L., Burgess, G. C., Curtiss, S. W., Diaz-Santos, M., Elam, J. S., et al. (2019). The lifespan human connectome project in aging: An overview. *Neuroimage*, *185*, 335–348.
- Calhoun, V. D., & Sui, J. (2016). Multimodal fusion of brain imaging data: A key to finding the missing link (s) in complex mental illness. *Biological psychiatry: cognitive neuroscience and neuroimaging*, *1*(3), 230–244.
- Cammoun, L., Gigandet, X., Meskaldji, D., Thiran, J. P., Sporns, O., Do, K. Q., Maeder, P., Meuli, R., & Hagmann, P. (2012). Mapping the human connectome at multiple scales with diffusion spectrum mri. *Journal of Neuroscience Methods*, *203*(2), 386–397. <https://doi.org/10.1016/j.jneumeth.2011.09.031>
- Chahal, R., Gotlib, I. H., & Guyer, A. E. (2020). Research review: Brain network connectivity and the heterogeneity of depression in adolescence—a precision mental health perspective. *Journal of Child Psychology and Psychiatry*, *61*(12), 1282–1298.
- Cox, R. W. (1996). Afni: Software for analysis and visualization of functional magnetic resonance neuroimages. *Computers and Biomedical research*, *29*(3), 162–173.
- D’Angelo, S., Murphy, T. B., & Alfò, M. (2018). Latent space modelling of multidimensional networks with application to the exchange of votes in eurovision song contest. *The Annals of Applied Statistics*. <https://api.semanticscholar.org/CorpusID:88523060>
- Derogatis, L. R. (1975). Brief symptom inventory. *European Journal of Psychological Assessment*.
- Desikan, R. S., Ségonne, F., Fischl, B., Quinn, B. T., Dickerson, B. C., Blacker, D., Buckner, R. L., Dale, A. M., Maguire, R. P., Hyman, B. T., et al. (2006). An automated labeling system for subdividing the human cerebral cortex on mri scans into gyral based regions of interest. *Neuroimage*, *31*(3), 968–980.
- Donohue, M. C., Sperling, R. A., Salmon, D. P., Rentz, D. M., Raman, R., Thomas, R. G., Weiner, M., Aisen, P. S., et al. (2014). The preclinical alzheimer cognitive composite: Measuring amyloid-related decline. *JAMA neurology*, *71*(8), 961–970.
- Edwards, D. (2000). *Introduction to graphical modelling*. Springer Science & Business Media.
- Farias, S. T., Mungas, D., Reed, B. R., Cahn-Weiner, D., Jagust, W., Baynes, K., & DeCarli, C. (2008). The measurement of everyday cognition (ecog): Scale development and psychometric properties. *Neuropsychology*, *22*(4), 531.
- Friel, N., Rastelli, R., Wyse, J., & Raftery, A. E. (2016). Interlocking directorates in irish companies using a latent space model for bipartite networks. *Proceedings of the National Academy of Sciences*, *113*(24), 6629–6634.
- Gollini, I., & Murphy, T. B. (2016). Joint modeling of multiple network views. *Journal of Computational and Graphical Statistics*, *25*(1), 246–265.
- Gorgolewski, K., Burns, C. D., Madison, C., Clark, D., Halchenko, Y. O., Waskom, M. L., & Ghosh, S. S. (2011). Nipype: A flexible, lightweight and extensible neuroimaging data processing framework in python. *Frontiers in neuroinformatics*, *5*, 13.
- Greene, A. S., Shen, X., Noble, S., Horien, C., Hahn, C. A., Arora, J., Tokoglu, F., Spann, M. N., Carrión, C. I., Barron, D. S., et al. (2022). Brain–phenotype models fail for individuals who defy sample stereotypes. *Nature*, *609*(7925), 109–118.

- Greve, D. N., Salat, D. H., Bowen, S. L., Izquierdo-Garcia, D., Schultz, A. P., Catana, C., Becker, J. A., Svarer, C., Knudsen, G. M., Sperling, R. A., et al. (2016). Different partial volume correction methods lead to different conclusions: An 18f-fdg-pet study of aging. *Neuroimage*, *132*, 334–343.
- Greve, D. N., Svarer, C., Fisher, P. M., Feng, L., Hansen, A. E., Baare, W., Rosen, B., Fischl, B., & Knudsen, G. M. (2014). Cortical surface-based analysis reduces bias and variance in kinetic modeling of brain pet data. *Neuroimage*, *92*, 225–236.
- Guha, S., & Rodriguez, A. (2021). Bayesian regression with undirected network predictors with an application to brain connectome data. *Journal of the American Statistical Association*, *116*(534), 581–593.
- Harary, F. (1953). On the notion of balance of a signed graph. *Michigan Mathematical Journal*, *2*(2), 143–146.
- Heider, F. (1944). Social perception and phenomenal causality. *Psychological review*, *51*(6), 358.
- Hoff, P. (2005). Bilinear mixed-effects models for dyadic data. *J. Amer. Statist. Assoc.*, *100*(469), 286–295.
- Hoff, P. (2008). Modeling homophily and stochastic equivalence in symmetric relational data. In J. Platt, D. Koller, Y. Singer, & S. Roweis (Eds.), *Advances in neural information processing systems 20* (pp. 657–664). MIT Press.
- Hoff, P. D., Raftery, A. E., & Handcock, M. S. (2002). Latent space approaches to social network analysis. *Journal of the American Statistical Association*, *97*(460), 1090–1098.
- Holland, P. W., Laskey, K. B., & Leinhardt, S. (1983). Stochastic blockmodels: First steps. *Social networks*, *5*(2), 109–137.
- Jenkinson, M., Bannister, P., Brady, M., & Smith, S. (2002). Improved optimization for the robust and accurate linear registration and motion correction of brain images. *Neuroimage*, *17*(2), 825–841.
- Jenkinson, M., Beckmann, C. F., Behrens, T. E., Woolrich, M. W., & Smith, S. M. (2012). Fsl. *NeuroImage*, *62*(2), 782–790. <https://doi.org/10.1016/j.neuroimage.2011.09.015>
- Jenkinson, M., & Smith, S. (2001). A global optimisation method for robust affine registration of brain images. *Medical image analysis*, *5*(2), 143–156.
- Kueper, J. K., Speechley, M., & Montero-Odasso, M. (2018). The alzheimer’s disease assessment scale–cognitive subscale (adas-cog): Modifications and responsiveness in pre-dementia populations. a narrative review. *Journal of Alzheimer’s Disease*, *63*(2), 423–444.
- Lauritzen, S. L. (1996). *Graphical models* (Vol. 17). Clarendon Press.
- Leonard, T., & Hsu, J. S. (1992). Bayesian inference for a covariance matrix. *The Annals of Statistics*, *20*(4), 1669–1696.
- Lutkenhoff, E. S., Rosenberg, M., Chiang, J., Zhang, K., Pickard, J. D., Owen, A. M., & Monti, M. M. (2014). Optimized brain extraction for pathological brains (optibet). *PloS one*, *9*(12), e115551.
- MacDonald, P. W., Levina, E., & Zhu, J. (2022). Latent space models for multiplex networks with shared structure. *Biometrika*, *109*(3), 683–706.
- Manjón, J. V., Coupé, P., Concha, L., Buades, A., Collins, D. L., & Robles, M. (2013). Diffusion weighted image denoising using overcomplete local pca. *PLoS ONE*, *8*(9). <https://doi.org/10.1371/journal.pone.0073021>
- McPherson, M., Smith-Lovin, L., & Cook, J. M. (2001). Birds of a feather: Homophily in social networks. *Annual review of sociology*, *27*(1), 415–444.
- Minhas, S., Hoff, P. D., & Ward, M. D. (2019). Inferential approaches for network analysis: Amen for latent factor models. *Political Analysis*, *27*(2), 208–222.
- Moore, C., & Sciacca, F. (2019). Fiber assignment by continuous tracking algorithm (fact). *Radiopaedia.org*. <https://doi.org/10.53347/rid-72014>
- Morris, E. L., He, K., & Kang, J. (2022). Scalar on network regression via boosting. *The annals of applied statistics*, *16*(4), 2755.
- Murphy, K. (2001). An introduction to graphical models. *Rap. tech*, *96*, 1–19.

- NIH. (2022). Nih toolbox scoring and interpretation guide [Accessed on February 16, 2024]. https://www.nihtoolbox.org/app/uploads/2022/05/Toolbox_Scoring_and_Interpretation_Guide_for_iPad_v1.7-5.25.21.pdf
- Norman, R. Z., et al. (1965). Structural models: An introduction to the theory of directed graphs.
- Papp, K. V., Rentz, D. M., Maruff, P., Sun, C.-K., Raman, R., Donohue, M. C., Schembri, A., Stark, C., Yassa, M. A., Wessels, A., et al. (2021). The computerized cognitive composite (c3) in a4, an alzheimer's disease secondary prevention trial. *The journal of prevention of Alzheimer's disease*, 8, 59–67.
- Paquette, M., Girard, G., Chamberland, M., & Descoteaux, M. (2016). "noise" in diffusion tractography connectomes is not additive.
- Reli3n, J. D. A., Kessler, D., Levina, E., & Taylor, S. F. (2019). Network classification with applications to brain connectomics. *The annals of applied statistics*, 13(3), 1648.
- Robins, G., Pattison, P., Kalish, Y., & Lusher, D. (2007). An introduction to exponential random graph (p*) models for social networks. *Social networks*, 29(2), 173–191.
- Salter-Townshend, M., & McCormick, T. H. (2017). Latent space models for multiview network data. *The annals of applied statistics*, 11(3), 1217.
- Simpson, S. L., Shappell, H. M., & Bahrami, M. (2024). Statistical brain network analysis. *Annual Review of Statistics and Its Application*, 11.
- Sinke, M. R., Dijkhuizen, R. M., Caimo, A., Stam, C. J., & Otte, W. M. (2016). Bayesian exponential random graph modeling of whole-brain structural networks across lifespan. *NeuroImage*, 135, 79–91.
- Song, M., Liu, Y., Zhou, Y., Wang, K., Yu, C., & Jiang, T. (2009). Default network and intelligence difference. *IEEE Transactions on autonomous mental development*, 1(2), 101–109.
- Sosa, J., & Buitrago, L. (2021). A review of latent space models for social networks. *Revista Colombiana de Estadística*, 44(1), 171–200.
- Sperling, R. A., Rentz, D. M., Johnson, K. A., Karlawish, J., Donohue, M., Salmon, D. P., & Aisen, P. (2014). The a4 study: Stopping ad before symptoms begin? *Science translational medicine*, 6(228), 228fs13–228fs13.
- Steinley, D. (2021). Recent advances in (graphical) network models [PMID: 34029161]. *Multivariate Behavioral Research*, 56(2), 171–174. <https://doi.org/10.1080/00273171.2021.1911777>
- Sweet, T. (2016). Social network methods for the educational and psychological sciences. *Educational Psychologist*, 51(3-4), 381–394.
- Tourbier, S., Rue Queralt, J., Glomb, K., Aleman-Gomez, Y., Mullier, E., Griffa, A., Sch3ttner, M., Wirsich, J., Tuncel, A., Jancovic, J., Bach Cuadra, M., & Hagmann, P. (2022). Connectome mapper 3: A flexible and open-source pipeline software for multiscale multimodal human connectome mapping. *Journal of Open Source Software*, 7(74), 4248. <https://doi.org/10.21105/joss.04248>
- Vitale, M. P., Giordano, G., & Ragozini, G. (2022). Discussion to: Bayesian graphical models for modern biological applications by Y. Ni, V. Baladandayuthapani, M. Vannucci and F.C. Stingo. *Statistical Methods & Applications*, 31(2), 269–278. <https://doi.org/10.1007/s10260-021-00603->
- Wang, L., Durante, D., Jung, R. E., & Dunson, D. B. (2017). Bayesian network–response regression. *Bioinformatics*, 33(12), 1859–1866.
- Wang, L., Lin, F. V., Cole, M., & Zhang, Z. (2021). Learning clique subgraphs in structural brain network classification with application to crystallized cognition. *NeuroImage*, 225, 117493.
- Wang, S. (2021). Recent integrations of latent variable network modeling with psychometric models. *Frontiers in psychology*, 12, 773289.
- Wang, S., & Edgerton, J. (2022). Resilience to stress in bipartite networks: Application to the islamic state recruitment network. *Journal of Complex Networks*, 10(4), cnac017.
- Wasserman, S., & Faust, K. (1994). Social network analysis: Methods and applications.

- Wasserman, S., & Pattison, P. (1996). Logit models and logistic regressions for social networks: I. an introduction to markov graphs andp. *Psychometrika*, *61*(3), 401–425.
- Weintraub, S., Dikmen, S. S., Heaton, R. K., Tulskey, D. S., Zelazo, P. D., Bauer, P. J., Carlozzi, N. E., Slotkin, J., Blitz, D., Wallner-Allen, K., et al. (2013). Cognition assessment using the nih toolbox. *Neurology*, *80*(11 Supplement 3), S54–S64.
- Xia, C. H., Ma, Z., Cui, Z., Bzdok, D., Thirion, B., Bassett, D. S., Satterthwaite, T. D., Shinohara, R. T., & Witten, D. M. (2020). *Multi-scale network regression for brain-phenotype associations* (tech. rep.). Wiley Online Library.
- Xu, F., Garai, S., Duong-Tran, D., Saykin, A. J., Zhao, Y., & Shen, L. (2022). Consistency of graph theoretical measurements of alzheimer’s disease fiber density connectomes across multiple parcellation scales. *(BIBM) 2022 IEEE International Conference on Bioinformatics and Biomedicine IEEE, Regular Paper*.
- Yan, J., Raja, V. V., Huang, Z., Amico, E., Nho, K., Fang, S., Sporns, O., Wu, Y. C., Saykin, A., Goñi, J., & Shen, L. (2020). Brain-wide structural connectivity alterations under the control of alzheimer risk genes. *International Journal of Computational Biology and Drug Design*, *13*(1), 58. <https://doi.org/10.1504/ijcbdd.2020.105098>
- Zhang, Z., Allen, G. I., Zhu, H., & Dunson, D. (2019). Tensor network factorizations: Relationships between brain structural connectomes and traits. *Neuroimage*, *197*, 330–343.
- Zhang, Z. (2021). A note on wishart and inverse wishart priors for covariance matrix. *Journal of Behavioral Data Science*, *1*(2), 119–126.
- Zhao, Y., Li, T., & Zhu, H. (2022). Bayesian sparse heritability analysis with high-dimensional neuroimaging phenotypes. *Biostatistics*, *23*(2), 467–484.
- Zhou, Y., & Müller, H.-G. (2022). Network regression with graph laplacians. *The Journal of Machine Learning Research*, *23*(1), 14383–14423.

Appendix

Figure 1: A triangle structure between nodes a, b and c for illustration.

Figure 2: The schematic diagram of LatentSNA. In the left panel, the individual functional networks and internalizing psychopathology measures are used as inputs to obtain region-specific covariance between connectivity and internalizing measures. In the right panel, we show examples of balanced and clusterable triads (A) and the diagram of LatentSNA model (B). With the input of functional connectivity matrices, $\{X_1, X_2, \dots, X_N\}$ and internalizing measures, Y , we propose a joint latent variable model, where we allow the latent connectivity variables, Z and latent behavior variables, Θ to co-vary with a shared covariance matrix, Σ in the generative Bayesian framework. Intercepts a and error term E are part of the data generation process for connectivity; and intercepts b and error term Ψ are part of the data generation process for behavior. Circles represent unknown quantities of the model, and squares represent the observed data. Non-informative priors are not included in this diagram. In the right panel section C, we show estimated internalizing network via marginal test (top, blue), estimated internalizing network via LatentSNA (bottom, blue) and a theoretical star-like structure in red.

Authors' Response to Reviewers' Comments: “Neuroimaging connectivity analysis needs network science for brain-behavior linking” (NMETH-A54306)

Dear Editor,

Thank you for giving us the opportunity to further revise our manuscript. We have responded to all points from the new round of reviews below. In the sections below, we present the verbatim text of the reviews in the order of the review. Following each review comment, we discuss how we improved the manuscript to address the point raised by the reviewer in blue. We thank the reviewers for their thoughtful and productive suggestions and believe that the manuscript is substantially improved as a result. In addition to this letter, we include a changed version of the edited manuscript with changes highlighted in blue italics for ease of spotting them.

Best wishes,

Authors

Reviewer 1

In this paper, Wang et al. introduces a method, called latent variable-based statistical network analysis (LatentSNA), to improve the statistical power for detection imaging biomarkers. The method is applied to model multi-state functional networks with multivariate internalizing profiles using of a few thousand children in the Adolescent Brain Cognitive Development (ABCD) study. The approach was demonstrated to yield markedly improved statistical power in detecting biomarkers and the latent variables were shown to explain the internalizing profiles better.

We really appreciate Reviewer 1 for the comments and for taking the time to review our work. We have made changes according to your suggestions and our response to each comment is as follows.

1. My main concern is with the lack of focus of the paper. What is the main contribution? LatentSNA or the neuroscientific findings associated with internalizing psychopathology?

If the main contribution is LatentSNA, the method has not been shown to be generalizable. The authors claimed: “Our approach can be used to detect other types of biomarkers using positron emission tomography, T1-weighted structural MRI and Diffusion Tensor Imaging, etc. Our approach can also detect imaging biomarkers of other behavioral, cognitive and psychopathology measures.” However, judging based on the limited results presented, these claims are at best unsubstantiated.

If the contribution is on the neuroscientific findings, the findings are overly narrow being focusing on internalizing psychopathology. Observations of star-like architectures are not necessarily generalizable to other conditions. The paper does not appear to be a good fit for Nature Methods.

Response: Thanks for the comments. To address the concern about the lack of focus of the paper, we have rewritten the manuscript including the title to emphasize our contribution to the current literature on brain

connectivity-to-behavior modeling by proposing a novel method that models the correspondence between latent structures from brain connectivity and behavior profiles. The proposed method addresses the lack of power and inflated Type II errors in current analytic approaches in this field when detecting brain network neuromarkers, allows unbiased estimation of neuromarkers' relationship with behavioral variants, quantifies the uncertainty, and evaluates the likelihood of the estimated neuromarkers' effects against chance. Ultimately, it improves brain-behavior prediction in novel samples and enhances the clinical utility of neuroimaging findings. To better clarify the focus on the innovation of the method, we have added relevant descriptions in the **Abstract**, **Introduction Section**, and **Discussion Section**.

In the **Abstract**:

We propose a latent statistical network analysis (LatentSNA) that implements network science in a generative Bayesian framework, preserves the neurologically meaningful brain topology and improves the statistical power for imaging biomarker detection. LatentSNA (1) addresses the lack of power and inflated Type II errors in current analytic approaches when detecting imaging biomarkers, (2) allows unbiased estimation of biomarkers' influence on behavior variants, (3) quantifies the uncertainty and evaluates the likelihood of the estimated biomarker effects against chance and (4) ultimately improves brain-behavior prediction in novel samples and the clinical utilities of neuroimaging findings. LatentSNA is broadly applied to multiple neuroimaging landmark studies, imaging modalities, outcome measures with developing, aging and transdiagnostic populations, totaling 8,003 to 11,861 participants. When applied to the Adolescent Brain Cognitive Development Study with 5,000 to 7,000 children, LatentSNA provides an unprecedented view of how network topology is implicated in brain dysfunction during neurodevelopment.

In the **Section Introduction**:

Neuroimaging encompasses techniques that provide in vivo depiction of the anatomy and function of the central nervous system to scientifically and objectively study the human brain in a non-invasive manner. Some imaging techniques focus on the structure of the brain (e.g., CAT¹, MR², DTI³), while others allow us to characterize brain activity or function (e.g., fMRI⁴, PET⁵). A major hurdle for modeling neuroimaging data is the highly correlated and connected nature of measurements throughout the brain, not dissimilar to networks (Barabási, 2013; Newman, 2003; Wasserman & Faust, 1994), which contributes to low statistical power for identifying brain-behavior links. Given the networked nature of the brain, a marriage between network science, a complexity-driven discipline focused on the shared architecture of networks emerging across physical, biological, and social domains, and neuroimaging shows great promise to provide a rich substrate to understand the brain and its decisions.

Neuroimaging connectivity models recognize and select meaningful patterns from neuroimages that explain individual differences in behavior, cognition, and other outcomes. For example, case-control comparisons measure differences in connectivity between healthy individuals and

¹Computerized Axial Tomography

²Magnetic Resonance Imaging

³Diffusion Tensor Imaging

⁴Functional Magnetic Resonance Imaging

⁵Positron Emission Tomography

patients to identify markers of dysfunction (e.g., Duffy & Als, 2012; Pornpattananangkul et al., 2019). Univariate and marginal association analyses calculate associations between connectivity and outcomes to identify significant links (e.g., Marek et al., 2019; Satterthwaite et al., 2015). By vectorizing unique pairwise edges from symmetric functional connectomes, CPM⁶ (Finn et al., 2015; Shen et al., 2017) achieves functional imaging biomarker detection using a multivariate regression model controlling overfitting with cross-validation. Machine learning algorithms such as Ridge (Gao et al., 2019; Mihalik et al., 2019; Rosenblatt et al., 2023), LASSO⁷ (Gao et al., 2019), SVM⁸ (Hearst et al., 1998), RF⁹ (Belgiu & Drăguț, 2016), and CNN¹⁰ (Albawi et al., 2017) are integrated to improve the connectivity model's predictability for individual outcomes.

A critical challenge remains: connectivity edges are treated as independent observations, whereas evidence supports the dependent organization of brain networks as informative neurobiological indicators (Bassett & Sporns, 2017; Bullmore & Sporns, 2009; Zuo et al., 2012). Graphical models, consisting of both undirected Gaussian graphical models (e.g., Dobra et al., 2004; Dobra et al., 2011; Friedman et al., 2008; Friedman, 2004) and directed acyclic graphs (e.g., Friedman et al., 2000; Geiger & Heckerman, 2002), describe the conditional dependence among random variables and directly address the violation of the independence assumption. A key task of graphical models, when applied to neuroimaging data, is to estimate and create brain connectivity networks based on whether signals from two brain regions are conditionally independent of each other (Ni et al., 2022). Individual behaviors and outcomes can be incorporated to influence the estimation of brain connectivity networks (Ni et al., 2022; Ni et al., 2019). While the graphical model addresses the unrealistic independence assumption in most existing connectivity models, it (1) does not specifically address the unique topological structures of networks, as the method is built for multidimensional random variables, not necessarily with real-world network structures among them, (2) does not directly achieve neuroimaging biomarker detection, and (3) cannot be readily used for imaging-based outcome prediction.

What differentiates a network-science-driven analytic approach is that it draws on insights regarding the universality of the communicative structures of real-world networks (Barabási, 2013). Characteristics such as the small-world property and sparsity are universal properties found in social networks (Wohlgemuth, 2012), political networks (Wang & Edgerton, 2022), the World Wide Web (Fatta et al., 2016), and human connectomes (Amaral et al., 2000; Bassett & Bullmore, 2006; Dubitzky et al., 2013). Shared network architectures, as a result of being governed by universal principles (Barabási, 2013; Wohlgemuth, 2012), allow us to use a common set of mathematical and statistical instruments for network modeling. Analytic frameworks aimed at accommodating dependent and topological characteristics of networks are called statistical network-based analysis (SNA). See reviews on SNA methods (Desmarais & Cranmer, 2017; Goldenberg et al., 2010; Kim et al., 2018; Matias & Robin, 2014; Newman, 2003; Smith et al., 2019; Wasserman & Faust, 1994).

LatentSNA, a novel inference-focused generative Bayesian framework capturing universal network topologies and leveraging state-of-the-art latent space estimation techniques, is particularly designed to analyze human connectomes and identify meaningful neuroimaging biomarkers of individual outcomes (Figure 2). It comprises an integrated workflow containing three

⁶Connectome-based Predictive Modeling

⁷Least Absolute Shrinkage and Selection Operator

⁸Support Vector Machines

⁹Random Forest

¹⁰Convolutional Neural Networks

modules: networked connectome modeling (preserving transitivity and modularity), psychometric behavior profiling, and two-way brain-behavior linking. We achieve robust neuroimaging biomarker detection with markedly improved statistical power, demonstrating the method’s generalizability across seven neuroimaging landmark studies: Alzheimer’s Disease Neuroimaging Initiative Grand Opportunities and ADNI Phase 2 (ADNI-GO/2) and ADNI Phase 3 (ADNI-3, Jack Jr et al., 2008; Petersen et al., 2010), Anti-Amyloid Treatment in Asymptomatic Alzheimer’s Disease (A4) (Sperling et al., 2014), Human Connectome Project Aging (HCP-A) (Bookheimer et al., 2019), Adolescent Brain Cognitive Development Study Baseline (ABCD-B) and second release (ABCD-2) (Casey et al., 2018), and transdiagnostic data collected at Yale (Greene et al., 2022). These studies involve eight different imaging modalities and 20 outcome measures with a total of 8,003 to 11,861 participants. LatentSNA consistently improves model fit performance over nine existing methods, including three state-of-the-art deep learning techniques (SVM, RF, and CNN), two network-based brain analysis methods (GC¹¹ and TNFA¹²), and four popular brain-behavior linking approaches (CPM, rCPM, LASSO, and CCA¹³). It enhances the predictability and potential clinical utility of various imaging techniques, including fMRI, T1-weighted structural MRI, DTI, and PET. Moreover, it is generalizable to different outcome measures, including but not limited to cognition, emotion, assessment of mental disorders, focal tau PET (¹⁸F]flortaucipir) SUVR metrics, and different participant demographics across developing, aging, and transdiagnostic populations.

Using LatentSNA in the largest children’s developmental study in the U.S. (ABCD-B), we gain an unprecedented view of how network topology is implicated in brain dysfunction during neurodevelopment. Large-scale disruptions in the functional communicative patterns of brain connectomes across multiple interconnected functional systems are found to explain differences in internalizing symptoms among children (Krueger & Markon, 2006). Star-like topological architectures, known for their efficiency in information dissemination, resiliency with local transmission failure, and affiliation with congestion (Guimerà et al., 2002; Sawai, 2012, 2014), are identified as the fingerprints of internalizing psychopathology and its deterioration in children. Overall, LatentSNA demonstrates high-quality fit to various imaging data, generates novel scientific insights, and enriches discussions surrounding existing neuroscience questions.

In the Section Discussion:

In this study, we developed a network-science driven analytic method that addresses the lack of power and inflated Type II errors in neuroimaging biomarker detection. LatentSNA allows unbiased estimation of biomarkers’ influence on behavior variants, quantifies the uncertainty and evaluates the likelihood of the estimated biomarker effects against chance and ultimately improves brain-behavior prediction in novel samples and the clinical utilities of neuroimaging findings. The LatentSNA method achieves robust neuroimaging biomarker detection with markedly improved statistical power; and it has been demonstrated to show broad generalizability with multiple neuroimaging landmark studies, different imaging modalities, outcome measures and participant populations.

To address concerns about the generalizability of the method, we extensively evaluated our approach in

¹¹Penalized Graph Classification (Reli3n et al., 2019)

¹²Tensor Network Factorization Analysis (Zhang et al., 2019)

¹³Canonical Correlation Analysis

new neuroimaging landmark studies with diverse populations. We outline the study cohorts and datasets in Table 1 and demonstrate the method’s generalizability to additional imaging biomarker types in Table 2, and to different outcome measures in Table 3. In Table 1, we outline the study cohorts and datasets included in the study: Alzheimer’s Disease Neuroimaging Initiative Grand Opportunities and ADNI Phase 2 (ADNI-GO/2), ADNI Phase 3, Anti-Amyloid Treatment in Asymptomatic Alzheimer’s Disease (A4), Human Connectome Project Aging (HCP-A), Adolescent Brain Cognitive Development Baseline (ABCD-B), the 2-year follow-up (ABCD-2), and Transdiagnostic data collected at Yale. We fitted the model to each imaging modality and outcome measure outlined in Tables 2 and 3. We focus on cognition outcomes, which are often used to assess the performance of novel methods; emotion outcomes closely aligned with the internalizing outcomes in the original manuscript, such as depression and anxiety; disorder; and focal tau PET SUVR outcomes that directly reflect biological changes in the brain.

Collectively, we have applied the method to 7 different datasets, involving 8 different imaging modalities and 20 outcome measures, with a total of 8,003–11,861 subjects’ information included. The results demonstrate that the proposed method is broadly applicable to a variety of imaging techniques, including functional MRI (fMRI), T1-weighted structural MRI, Diffusion Tensor Imaging (DTI), and positron emission tomography (PET). It is also generalizable to different types of outcome measures, which include, but are not limited to, cognition, emotion, assessment of mental disorders, and focal tau PET SUVR metrics. We use disorder outcomes to represent scales measuring psychopathological and psychological symptoms, although many disorder symptoms manifest as dysfunctions in one or more cognitive and emotional dimensions. The proposed method consistently improves the model fit across various data situations compared to available alternatives, with satisfactory prediction accuracy for a variety of outcomes using functional and structural imaging modalities in developing, aging, and transdiagnostic populations.

Table 1: Study cohorts and data

Data	Outcome				Neuroimaging			Sample Size
	Cognition	Disorder	Emotion	Tau	Structural ^a	fMRI/R ^b	fMRI/T ^c	
ADNI-GO/2	✓	✓			✓			410
ADNI-3	✓	✓				✓		174
A4	✓			✓		✓		394
HCP-Aging	✓		✓			✓		529
ABCD-B			✓			✓	✓	4,871-7,606
ABCD-2			✓			✓	✓	1,435-2,558
Transdiagnostic		✓					✓	190
Total	✓	✓	✓	✓	✓	✓	✓	8,003 - 11,861

^aThree types of structural imaging information are used including the fiber density, the number of fibers and fiber length. ^bfMRI collected when subjects are asked to rest. ^cfMRI collected when subjects are asked to perform cognitive and emotional tasks. We include emotional n-back task (EN-back), the Stop Signal task (SST) and the Monetary Incentive Delay (MID) task conditions.

Description of Data

We now provide details about the neuroimaging data and outcome measures for each study. For the ABCD study, the neuroimaging and internalizing outcomes have been discussed in the original submission, and are thus excluded here.

ADNI

Data used in the preparation of this article were obtained from the Alzheimer’s Disease Neuroimaging Initiative (ADNI) database (adni.loni.usc.edu). The ADNI was launched in 2003 as a public-private partnership, led by Principal Investigator Michael W. Weiner, MD. The primary goal of ADNI has been to test whether serial magnetic resonance imaging (MRI), positron emission tomography (PET), other biological markers, and clinical and neuropsychological assessment can be combined to measure the progression of mild cognitive impairment (MCI) and early Alzheimer’s disease (AD).

sMRI and DTI. We downloaded the T1-weighted structural MRI (sMRI) and Diffusion Tensor Imaging (DTI) from the ADNI-GO/2 databases from 174 subjects. We applied an overcomplete local principal components analysis (Manjón et al., 2013) to process DTI data following standard steps including denoising, motion-correction and distortion-correction. We performed the probabilistic white matter fiber tractography using the fiber assignment by continuous tracking (Moore & Sciacca, 2019). We registered sMRI scans to the lower resolution b0 volume of the DTI data using the FLIRT toolbox in the FMRIB Software Library (Jenkinson et al., 2012), and we then defined the cortical ROIs in the FreeSurfer space using the Lausanne 2008 parcellation with 68 cortical regions of interest (Cammoun et al., 2012). We obtained the number of the fibers (NOF) connecting each pair of ROIs as well as the regions’ surface area (SA). The fiber density-based structural connectivity was calculated by dividing NOF between two ROIs with their average surface areas (Xu et al., 2022; Yan et al., 2020). Three types of structural brain networks were constructed as the number of fibers between a pair of brain regions, the length of the fibers as well as the fiber density of tracts connecting pairs of ROIs.

fMRI. We used resting state functional neuroimaging data from the third release of the ADNI study. We processed the images using the ConnectomeMapper3 pipeline (Toumbier et al., 2022) built in Nipype (Gorgolewski et al., 2011). The resting-state fMRI images were processed with despiking and slice timing correction following Cox (1996); the images also were motion-corrected and distortion-corrected using FSL. The rs-fMRI images were registered to the b0 sMRI using the FLIRT toolbox (Jenkinson et al., 2002; Jenkinson & Smith, 2001). The BOLD time signals of each ROI were band-pass filtered and then detrended using a linear regression. We constructed functional brain networks for each subject as the Pearson correlation between the BOLD time signals for pairs of ROIs.

Disorder/Cognition Outcomes. For outcomes, we included the Alzheimer’s Disease Assessment Scale, cognitive subscale (ADAS, Kueper et al., 2018), a rating of dysfunction by AD. We included the ADAS score as the sum of 13 diagnostic questions collected at baseline. In addition, we included the sum score of the Everyday Cognition Scale (Ecog, Farias et al., 2008), a questionnaire measuring the patient’s cognitive function, which is often used to aid the diagnosis of dementia and other neurodegenerative aging-related disorders. We applied the proposed LatentSNA to both ADAS and Ecog to assess the model’s generalizability to alternative outcomes for the aging population.

A4

Anti-Amyloid Treatment in Asymptomatic Alzheimer’s Disease (A4) study is a secondary prevention trial targeted towards older people with amyloid accumulation and are at high risk for AD dementia (Sperling et al., 2014).

sMRI and fMRI. For the fMRI data, we used the same processing procedure as that for the ABCD. MPRAGE scans were skull stripped using optiBET (Lutkenhoff et al., 2014) and nonlinearly aligned to the MNI-152 template using BioImage Suite.

Table 2: Prediction accuracy in independent samples with different types of imaging biomarkers

Biomarker Type	Outcome	Method							
		CPM	rCPM	GC	TNFA	SVM	RF	CNN	LatentSNA
Fiber Density	ADAS	0.223	0.256	0.238	0.265	0.279	0.330	0.279	0.646
	ECog	0.234	0.176	0.212	0.362	0.236	0.326	0.304	0.572
# Fibers	ADAS	0.226	0.381	0.692	0.297	0.258	0.373	0.381	0.692
	ECog	0.256	0.258	0.326	0.448	0.167	0.342	0.243	0.663
Fiber Length	ADAS	0.421	0.236	0.265	0.366	0.242	0.274	0.307	0.681
	ECog	0.242	0.230	0.221	0.336	0.235	0.264	0.284	0.659

Focal tau PET SUVR metrics. We used the PETSURFER within FreeSurfer for an integrated MRI-PET analysis (Greve et al., 2016; Greve et al., 2014). We derived the focal tau PET ($[^{18}\text{F}]\text{flortaucipir}$) SUVR metrics from the A4 images using 90-110 minute (4x5-minute frames) post-injection images, preprocessed and analyzed using PETSURFER in FreeSurfer (v 6.0+). We summed and motion-corrected the five-minute tau PET frames. We then aligned the composite PET images to corresponding MRI images, parcellated using the Desikan-Killiany Atlas (DSK Desikan et al., 2006), and partial-volume corrected using FreeSurfer. We gathered the average tracer absorption values for each region defined by the atlas and computed standardized uptake value ratios (SUVR) using the whole cerebellar cortex as the reference region.

Cognition Outcome. To assess cognition changes, we included the Preclinical Alzheimer’s Cognitive Composite (PACC, Donohue et al. (2014)) collected as part of the A4 project. PACC is a composite cognitive score combining tests that assess episodic memory, executive function and general cognition, and it is the primary outcome measure for A4 targeting the preclinical AD population. PACC is found to be sensitive to the earliest disease-related change (Papp et al., 2021).

HCP-Aging

The Lifespan Human Connectome Project in Aging (HCP-Aging) aims to characterize how brain organization and connectivity changes during typical aging, compared to an ‘abnormal’ aging process, a potentially distinct but not inevitable condition of aging (Bookheimer et al., 2019).

fMRI. For the fMRI data, we used the same processing procedure as that for the ABCD.

Emotion/Cognition Outcomes. For the HCP-Aging project, we focused on cognition and emotion measures. To assess the cognitive capability of the healthy aging population, we included the composite scores for the Picture Sequence Memory Test as well as the Cognition Composite score including Fluid Composite and Crystallized Composite, derived from all National Institutes of Health Toolbox Cognition tasks (NIH TB-CT, Weintraub et al., 2013). For the emotion outcomes, we chose the Emotional Distress Depression and the PROMIS Anxiety to maintain relative consistency with the internalizing outcome. The Emotional Distress Depression is captured by the Sadness Survey, from National Institutes of Health Toolbox Emotion Battery (NIH TB-EB, NIH (2022)), which measures the negative mood and perceptions. The PROMIS Anxiety is captured by Fear-Affect Survey, a self-report measure assessing fear and anxious misery from NIH TB-EB.

Transdiagnostic

The Transdiagnostic project aims to recruit clinically naturalistic and demographically diverse subjects to more effectively study the links between imaging and behaviors (Greene et al., 2022); the project was conducted at Yale between February 2018 and March 2021. Participants of the Transdiagnostic project

Table 3: Prediction accuracy in independent samples with different types of outcomes

Outcome Measure	Name	Source	Condition	Method							
				CPM	rCPM	GC	TNFA	SVM	RF	CNN	LatentSNA
Cognition	Ecog	ADNI-3	Rest	0.245	0.301	0.304	0.341	0.366	0.315	0.299	0.551
	PACCRN	A4	Rest	0.259	0.270	0.232	0.228	0.250	0.270	0.201	0.543
	Picture Sequence Memory Cognition Composite	HCP-Aging	Rest	0.331	0.381	0.363	0.247	0.329	0.394	0.295	0.663
Disorder	BSI	Transdiagnostic	nBack	0.099	0.143	0.032	0.194	0.153	0.191	0.221	0.494
	ADAS	ADNI-3	Rest	0.167	0.276	0.217	0.255	0.182	0.333	0.371	0.492
Emotion	Internalizing	ABCD-2	Rest	0.091	0.200	0.102	0.267	0.118	0.149	0.149	0.897
			nBack	0.153	0.162	0.080	0.156	0.198	0.149	0.175	0.774
			MID	0.124	0.273	0.361	0.276	0.203	0.148	0.167	0.721
			SST	0.083	0.229	0.099	0.239	0.176	0.208	0.160	0.692
	Emotional Distress Depression PROMIS Anxiety	HCP-Aging	Rest	0.416	0.302	0.198	0.286	0.265	0.295	0.259	0.658
Tau	Inferior Temporal	A4	Rest	0.211	0.349	0.276	0.284	0.303	0.298	0.326	0.599
	Middle Temporal			0.325	0.275	0.227	0.190	0.344	0.277	0.247	0.652
	Parahippocampal			0.254	0.141	0.229	0.222	0.152	0.280	0.276	0.624
	Fusiform			0.366	0.206	0.314	0.428	0.320	0.288	0.340	0.663
	Entorhinal			0.274	0.242	0.299	0.348	0.232	0.146	0.200	0.642
	Precuneus			0.303	0.208	0.271	0.369	0.278	0.411	0.300	0.653
	Supramarginal			0.327	0.317	0.288	0.299	0.380	0.222	0.306	0.613
	Posterior cingulate cortex			0.298	0.354	0.290	0.176	0.159	0.354	0.303	0.588
	Rosteral anterior cingulate			0.254	0.275	0.271	0.236	0.249	0.297	0.384	0.611
	Caudal anterior cingulate			0.270	0.274	0.225	0.243	0.301	0.295	0.317	0.569
Amygdala	0.252	0.200	0.170	0.263	0.303	0.236	0.231	0.588			

tend to show a wide range of symptom severity and commonly have multiple psychiatric diagnoses. All imaging information was collected at the Yale Magnetic Resonance Research Center.

fMRI. The data preprocessing for the fMRI of the Transdiagnostic project is the same as the processing of the fMRI for the ABCD study.

Disorder Outcomes. We included the global severity index of the Brief Symptom Inventory (BSI, Derogatis, 1975), a rating scale aiming to identify clinically relevant psychological symptoms in adolescents and adults. The global indices measure the level of symptomatology, its intensity and number of occurrence.

Results

Results of the additional imaging and outcomes are shown in Table 2 and Table 3. In Table 2, we demonstrate the method’s generalizability to detect various types of imaging biomarkers beyond functional MRI. Specifically, Table 2 includes structural imaging modalities that measure fiber density, number of fibers, and fiber length. Additionally, tau metrics from PET imaging in the A4 study are included in Table 3. We report the average correlation between predicted and observed outcomes in independent samples across 10 runs using the same procedure as before. The results indicate that using LatentSNA, we achieve satisfactory prediction accuracy across different imaging modalities, consistently improving upon existing methods. Further details about the comparison methods can be found in the response to comment 2 from reviewer 1.

In Table 3, we demonstrate the method’s generalizability to different outcomes. Specifically, we include four types of outcomes with 20 outcome variables focused on cognition, disorder, emotion, and focal tau PET SUVR metrics. The results show that using LatentSNA, we achieve satisfactory prediction accuracy across various outcome measures, consistently improving upon existing methods. More specifically, we observe significant improvements in predicting emotion outcomes based on fMRI data in the children’s population, which aligns with existing literature highlighting challenges in phenotype prediction using children’s fMRI data due to high heterogeneity and strong motion artifacts (Calhoun & Sui, 2016; Chahal et al., 2020; Greene et al., 2022). Predictability of focal tau PET SUVR metrics consistently exceeds that of other types of outcomes. This outcome is expected, as tau PET SUVR metrics, derived from A4 images using post-injection data, more directly reflect brain changes compared to behavioral outcomes.

To better clarify, we have rewritten the section **Section Methods**, **Section Data** that reads,

In the **Section Methods**:

There are two primary goals for our analyses. First, we would like to illustrate that LatentSNA is generalizable to different imaging modalities and individual outcome measures with high quality fit. Second, we would like to provide an in-depth analysis identifying imaging biomarkers of internalizing symptoms in children using the largest children’s developmental study (ABCD). We believe that improved model performance with LatentSNA can translate to potential enrichment of scientific discussions.

In the **Section Data**, we have added sections **ABCD**, **ADNI**, **A4**, **HCP-Aging** and **Transdiagnostic** that read,

We provide details about the neuroimaging data and outcome measures for each study. In Table 1, we outline study cohorts and datasets included in the study: Alzheimer’s Disease Neuroimaging Initiative Grand Opportunities and ADNI Phase 2 (ADNI-GO/2), ADNI Phase 3, Anti-Amyloid Treatment in Asymptomatic Alzheimer’s Disease (A4), Human Connectome Project

Aging (HCP-A), Adolescent Brain Cognitive Development Baseline (ABCD-B) and the 2-year follow-up (ABCD-2) and the Transdiagnostic data collected at Yale. We fit the model to each imaging modality and outcome measures. We focus on cognition outcomes that are often used to assess the performance of novel methods, emotion outcomes that are closely aligned with the internalizing outcomes such as depression and anxiety, disorder and focal tau PET SUVR outcomes that directly reflect biological changes in the brain.

ABCD

We used brain imaging data from the first and second releases of the largest long-term study of brain development and child health in the United States, collected from 11,875 children aged between 9 to 10 years old (Casey et al., 2018)

fMRI.*To investigate links The blood-oxygen-level-dependent (BOLD) functional activation was recorded for children during resting state (RS) and when they performed three emotional and cognitive tasks. The fMRI data were first preprocessed using BioImage Suite (Joshi et al., 2011). The standard preprocessing procedures, such as slice time and motion correction, registration to the MNI template, were described in Greene et al. (2018) and Horien et al. (2019). The eligible scans had no more than 0.10 mm mean frame-to-frame displacement. Brain images were parceled into 268 regions of interest (ROIs) or nodes using the Shen atlas including the cortex, subcortex and cerebellum (Shen et al., 2013). Within each node, the voxel-level time courses were aggregated. Functional connectivity (FC) was constructed for each child in the study during RS and each task state. The FC is constructed by creating a matrix, where each row and column represent all the nodes, and the value of the (i, j) th entry of the matrix is the Pearson correlation coefficient between the i th and j th nodes scaled to be normally distributed by a Fisher's z -transformation.*

To investigate whether there exists a shared set of neural substrates for internalizing psychopathology across different emotional and cognitive tasks and if the substrates are different from those during rest, we fitted LatentSNA separately to resting functional connectivity and functional connectivity during each task state. We included 7,606 adolescents with RS functional connectivity capturing intrinsic brain functional activity. We separately investigated the functional connectivity of 4,871 adolescents who are asked to perform the emotional n-back task (EN-back), 5,096 adolescents who are asked to perform the Stop Signal task (SST) and 5,298 adolescents who are asked to perform the Monetary Incentive Delay (MID) task.

Internalizing psychopathology.*Internalizing psychopathology represents a spectrum of conditions characterized by negative emotion including depression, anxiety and phobias (Krueger & Markon, 2006). In the ABCD study, the internalizing psychopathology is collected via self-reported survey using the Child Behavior Checklist (CBCL, Stavropoulos et al. (2017)), which includes 119 items aggregated into 8 empirical sub-scales. Three sub-scales of CBCL, anxious-depressed (13 items), withdrawn-depressed (8 items) and somatic complaints (11 items) are parts of the internalizing psychopathology. The multivariate representation of the internalizing psychopathology with anxious-depressed, withdrawn-depressed and somatic complaints variables likely outperforms the univariate representation (sum of the three variables) due to the loss of information in the latter. To investigate if this is true, we applied the proposed LatentSNA to both multivariate and univariate representation, and interpret the results based on the model with superior fit, the multivariate internalizing measures.*

ADNI

Data used in the preparation of this article were obtained from the Alzheimer's Disease Neuroimaging Initiative (ADNI) database (adni.loni.usc.edu). The ADNI was launched in 2003 as

a public-private partnership, led by Principal Investigator Michael W. Weiner, MD. The primary goal of ADNI has been to test whether serial magnetic resonance imaging (MRI), positron emission tomography (PET), other biological markers, and clinical and neuropsychological assessment can be combined to measure the progression of mild cognitive impairment (MCI) and early Alzheimer’s disease (AD).

smMRI and DTI. *We downloaded the T1-weighted structural MRI (smMRI) and Diffusion Tensor Imaging (DTI) from the ADNI-GO/2 databases from 174 subjects. We applied an overcomplete local principal components analysis (Manjón et al., 2013) to process DTI data following standard steps including denoising, motion-correction and distortion-correction. We performed the probabilistic white matter fiber tractography using the fiber assignment by continuous tracking (Moore & Sciacca, 2019). We registered SMRI scans to the lower resolution b0 volume of the DTI data using the FLIRT toolbox in the FMRIB Software Library (Jenkinson et al., 2012), and we then defined the cortical ROIs in the FreeSurfer space using the Lausanne 2008 parcellation with 68 cortical regions of interest (Cammoun et al., 2012). We obtained the number of the fibers (NOF) connecting each pair of ROIs as well as the regions’ surface area (SA). The fiber density-based structural connectivity was calculated by dividing NOF between two ROIs with their average surface areas (Xu et al., 2022; Yan et al., 2020). Three types of structural brain networks were constructed as the number of fibers between a pair of brain regions, the length of the fibers as well as the fiber density of tracts connecting pairs of ROIs.*

fmMRI. *We used resting state functional neuroimaging data from the third release of the ADNI study. We processed the images using the ConnectomeMapper3 pipeline (Tourbier et al., 2022) built in Nipype (Gorgolewski et al., 2011). The resting-state fmMRI images were processed with despiking and slice timing correction following Cox (1996); the images also were motion-corrected and distortion-corrected using FSL. The rs-fmMRI images were registered to the b0 smMRI using the FLIRT toolbox (Jenkinson et al., 2002; Jenkinson & Smith, 2001). The BOLD time signals of each ROI were band-pass filtered and then detrended using a linear regression. We constructed functional brain networks for each subject as the Pearson correlation between the BOLD time signals for pairs of ROIs.*

Disorder/Cognition Outcomes. *For outcomes, we included the Alzheimer’s Disease Assessment Scale, cognitive subscale (ADAS, Kueper et al., 2018), a rating of dysfunction by AD. We included the ADAS score as the sum of 13 diagnostic questions collected at baseline. In addition, we included the sum score of the Everyday Cognition Scale (Ecog, Farias et al., 2008), a questionnaire measuring the patient’s cognitive function, which is often used to aid the diagnosis of dementia and other neurodegenerative aging-related disorders. We applied the proposed LatentSNA to both ADAS and Ecog to assess the model’s generalizability to alternative outcomes for the aging population.*

A4

Anti-Amyloid Treatment in Asymptomatic Alzheimer’s Disease (A4) study is a secondary prevention trial targeted towards older people with amyloid accumulation and are at high risk for AD dementia (Sperling et al., 2014).

smMRI and fmMRI. *For the fmMRI data, we used the same processing procedure as that for the ABCD. MPRAGE scans were skull stripped using optiBET (Lutkenhoff et al., 2014) and nonlinearly aligned to the MNI-152 template using BioImage Suite.*

Focal tau PET SUVR metrics. *We used the PETSURFER within FreeSurfer for an integrated MRI-PET analysis (Greve et al., 2016; Greve et al., 2014). We derived the focal tau PET*

($[^{18}\text{F}]$ flortaucipir) SUVR metrics from the A4 images using 90-110 minute (4x5-minute frames) post-injection images, preprocessed and analyzed using PETSURFER in FreeSurfer (v 6.0+). We summed and motion-corrected the five-minute tau PET frames. We then aligned the composite PET images to corresponding MRI images, parcellated using the Desikan-Killiany Atlas (DSK Desikan et al., 2006), and partial-volume corrected using FreeSurfer. We gathered the average tracer absorption values for each region defined by the atlas and computed standardized uptake value ratios (SUVR) using the whole cerebellar cortex as the reference region.

Cognition Outcome. To assess cognition changes, we included the Preclinical Alzheimer's Cognitive Composite (PACC, Donohue et al. (2014)) collected as part of the A4 project. PACC is a composite cognitive score combining tests that assess episodic memory, executive function and general cognition, and it is the primary outcome measure for A4 targeting the preclinical AD population. PACC is found to be sensitive to the earliest disease-related change (Papp et al., 2021).

HCP-Aging

The Lifespan Human Connectome Project in Aging (HCP-Aging) aims to characterize how brain organization and connectivity changes during typical aging, compared to an 'abnormal' aging process, a potentially distinct but not inevitable condition of aging (Bookheimer et al., 2019).

fMRI. For the fMRI data, we used the same processing procedure as that for the ABCD.

Emotion/Cognition Outcomes. For the HCP-Aging project, we focused on cognition and emotion measures. To assess the cognitive capability of the healthy aging population, we included the composite scores for the Picture Sequence Memory Test as well as the Cognition Composite score including Fluid Composite and Crystallized Composite, derived from all National Institutes of Health Toolbox Cognition tasks (NIH TB-CT, Weintraub et al., 2013). For the emotion outcomes, we chose the Emotional Distress Depression and the PROMIS Anxiety to maintain relative consistency with the internalizing outcome. The Emotional Distress Depression is captured by the Sadness Survey, from National Institutes of Health Toolbox Emotion Battery (NIH TB-EB, NIH (2022)), which measures the negative mood and perceptions. The PROMIS Anxiety is captured by Fear-Affect Survey, a self-report measure assessing fear and anxious misery from NIH TB-EB.

Transdiagnostic

The Transdiagnostic project aims to recruit clinically naturalistic and demographically diverse subjects to more effectively study the links between imaging and behaviors (Greene et al., 2022); the project was conducted at Yale between February 2018 and March 2021. Participants of the Transdiagnostic project tend to show a wide range of symptom severity and commonly have multiple psychiatric diagnoses. All imaging information was collected at the Yale Magnetic Resonance Research Center.

fMRI. The data preprocessing for the fMRI of the Transdiagnostic project is the same as the processing of the fMRI for the ABCD study.

Disorder Outcomes. We included the global severity index of the Brief Symptom Inventory (BSI, Derogatis, 1975), a rating scale aiming to identify clinically relevant psychological symptoms in adolescents and adults. The global indices measure the level of symptomatology, its intensity and number of occurrence.

2. The current paper compares LatentSNA only with some of the most basic methods. There are plenty of latent-variable graphical models that are applicable to brain network analysis. Even dynamic causal modeling methods can be considered as based on latent models. A common trait of these methods, like LatentSNA, is that they represent complex signals in a low-dimensional space encoded by latent variables. A significant amount of work is needed to evaluate LatentSNA against the true state of the art.

Response: We greatly appreciate the reviewer for raising this interesting point. In response to the concern about insufficient evaluation of the proposed method against existing alternatives, we have provided strong evidence demonstrating the superiority of the proposed model over 9 existing methods, including 2 network-based brain analysis methods and 7 popular brain-behavior linking approaches. We evaluated the proposed method against penalized graph classification (GC), tensor network factorization analysis (TNFA), connectome-based predictive modeling (CPM), ridge connectome-based predictive model (rCPM), least absolute shrinkage and selection operator (LASSO), canonical-correlation analysis (CCA), support vector machines (SVM), random forest (RF), and convolutional neural network (CNN). In various data situations with different imaging modalities, outcome measures, and populations, our method consistently outperforms existing alternatives, showing, for example, an average improvement of 110% over TNFA and an average improvement of 150% over CPM. These results demonstrate that the proposed LatentSNA represents a valid and significant adaptation of social network analysis (SNA) concepts and methods for real-world networks and their application in linking brain and behavior.

Directly addressing the reviewer’s concern, we have added model evaluation results against two network-based brain analysis methods: the penalized graph classification (GC) approach (Reli3n et al., 2019) and the tensor network factorization analysis (TNFA). Additionally, we have included comparisons with three widely-used machine learning techniques—Support Vector Machines (SVM) (Hearst et al., 1998), Random Forest (RF) (Belgiu & Dr3gu7, 2016), and Convolutional Neural Networks (CNN) (Albawi et al., 2017).

The GC approach uses brain connectivity as predictors and applies both L1 penalty (absolute value of coefficient magnitudes) and a generic group lasso penalty. We implemented the GC approach using the graphclass R package (Aine et al., 2017; Reli3n et al., 2019). The tuning of the penalty factor pair (λ , ρ) was conducted on a 3x4 grid, with λ selected from the set $\{10^{-6}, 10^{-5}, 10^{-4}\}$ and ρ from $\{1, 10, 20, 30\}$. It was observed that a λ value exceeding 10^{-3} and a ρ value surpassing 40 result in the penalization of all coefficients to zero.

For the TNFA approach, similar to the tensor network principal components analysis method (Zhang et al., 2019), we embedded the $V \times V$ symmetric adjacency matrices into a low-dimensional matrix. Each row contains subjects’ principal component scores, and each column represents a basis network; only significant basis networks were included as predictors. Subsequently, we performed network predictor regression with the embedded low-dimensional basis networks predicting the outcome variables.

The SVM predicted behavioral outcomes based on a low-dimensional matrix derived from the $V \times V$ symmetric adjacency matrices, similar to the TNFA approach. This process involved embedding the adjacency matrices into reduced space, retaining only significant basis networks as predictors. We trained an SVM model with a linear kernel using the e1071 R package (Meyer et al., 2023). The model underwent parameter tuning via grid search to optimize the cost parameter, and the best model was used for predicting behavioral outcomes from the test data.

The RF method was implemented using the ranger package within the caret framework in R (Wright & Ziegler, 2017). Similar to SVM, it utilized features derived from a low-dimensional matrix of brain connectivity data. A grid search strategy optimized key parameters including the number of variables per split (mtry), node splitting criterion (splitrule), and minimum node size (min.node.size).

For CNN, we employed the torch package in R (Falbel & Luraschi, 2023) to fit the model. Our CNN architecture comprised sequential dense layers with ReLU activations, specifically designed for processing features extracted from low-dimensional connectivity data. The model underwent training using an Adam optimizer and cross-entropy loss function over multiple epochs to ensure optimal learning from the training data. Post-training, the CNN was utilized to predict outcomes on the test dataset.

In Table 2 and Table 3, we present the prediction accuracy of LatentSNA compared to the aforementioned methods. The results demonstrate that LatentSNA achieves superior prediction accuracy in independent samples across various imaging modalities and outcome variables. Particularly notable is the larger improvement observed in predicting emotion outcomes using fMRI data in the children’s population, indicating that the proposed method is particularly advantageous in heterogeneous and noisy data scenarios.

To provide further clarification, we have included relevant descriptions in the **Section Comparison Methods**.

In the **Section Comparison methods**:

We have added model evaluation results against two network-based brain analysis methods, the penalized graph classification (GC) approach (Reli3n et al., 2019) and the tensor network factorization analysis (TNFA). Additionally, we have incorporated comparisons with three widely-used machine learning techniques—Support Vector Machines (SVM) (Hearst et al., 1998), Random Forest (RF) (Belgiu & Drăguș, 2016), and Convolutional Neural Networks (CNN) (Albawi et al., 2017)—to provide a comprehensive assessment of our methods’ performance. The GC approach uses the brain connectivity as predictors and adopts both L1 penalty, the absolute value of coefficient magnitudes and a generic group lasso penalty. We fitted the GC approach using the graphclass R package (Aine et al., 2017; Reli3n et al., 2019). The tuning of the penalty factor pair (λ , ρ) was conducted on a 3×4 grid, with λ selected from the set $\{10^{-6}, 10^{-5}, 10^{-4}\}$ and $\rho \in \{1, 10, 20, 30\}$. It was observed that a λ value exceeding 10^{-3} and a ρ value surpassing 40 result in the penalization of all coefficients to zero.

For the TNFA approach, similar to the tensor network principal components analysis method (Zhang et al., 2019), we embedded the $V \times V$ symmetric adjacency matrices into a low dimensional matrix; each row contains subjects’ principal component scores and each column contains basis network; only significant basis networks were included as predictors. We then performed a network predictor regression with the embedded low-dimensional basis networks as predictors of the outcome variables. SVM predicted behavioral outcomes based on low dimensional matrix derived from the $V \times V$ symmetric adjacency matrices, akin to the TNFA approach. This process involves embedding the adjacency matrices into a reduced space, where only significant basis networks were retained as predictors. These features were then used to train an SVM model with a linear kernel using e1071 R package (Meyer et al., 2023). The model undergoes parameter tuning using a grid search to optimize the cost parameter, and the best model is used to predict behavioral outcomes from test data. The RF method is implemented using the ranger package within the caret framework in R (Wright & Ziegler, 2017), and similarly to SVM, it utilizes features derived from a low-dimensional matrix of brain connectivity data. A grid search strategy optimizes key parameters: the number of variables per split (mtry), the node splitting criterion (splitrule), and the minimum node size (min.node.size). For CNN, we fitted the model with the torch package in R (Falbel & Luraschi, 2023). Our CNN architecture consists of sequential dense layers with ReLU activations, specifically designed to handle the features extracted from the low-dimensional connectivity data. The model under-

goes training using an Adam optimizer and cross-entropy loss function across multiple epochs, ensuring optimal learning from the training data. Post-training, the CNN is used to predict outcomes on the test dataset.

In addition to expanding the pool of alternative methods to demonstrate the superiority of the proposed method, we aim to provide detailed discussions about the method’s innovation and emphasize the distinction between LatentSNA and latent-variable graphical models. We wish to clarify that the proposed method represents a pioneering effort to extend social network analysis (SNA, Wasserman & Faust, 1994) to jointly model functional connectomes and continuous behavioral outcomes, significantly enhancing the ability to detect region-specific imaging biomarkers.

The proposed LatentSNA makes a unique contribution to the current literature on social network analysis (SNA) methods. SNA methods are utilized to understand the structural and interaction patterns within observed networks, encompassing various domains such as social relationships (e.g., Friel et al., 2016; Gollini & Murphy, 2016; Minhas et al., 2019; Wang & Edgerton, 2022). These methods involve collecting information about connectivity (edges) between nodes, often visualizing and analyzing these networks using mathematical and statistical techniques. The interconnectedness (dependencies) among edges represents a significant network feature and a challenge in network modeling. Prominent statistical approaches for network modeling include exponential random graph models (ERGMs) (Robins et al., 2007; Wasserman & Pattison, 1996), stochastic block models (SBMs) (Airoldi et al., 2008; Holland et al., 1983), and latent space models (LSMs) (Hoff et al., 2002). These methods vary in how they address dependencies among edges. ERGMs, for instance, assume that dependencies among edges can be explained by descriptive features of the network, such as differences in node popularity (node degrees). In ERGMs, researchers explicitly specify dependent features within a network and estimate how network features influence the likelihood of connectivity, thereby inferring relational patterns. In contrast, latent variable-based network models impose natural network behaviors without explicit specification.

The proposed LatentSNA represents a novel extension of latent variable-based network models specifically tailored for detecting imaging biomarkers. Latent variable-based network models, also known as conditionally independent dyadic (CID) network models (Sweet, 2016), address dependencies among connectivity edges using latent variables. It’s important to note that latent variables in social network analysis (SNAs) differ from those in graphical models (GMs). These models assume conditional independence among edges—for example, the edge between brain region A and B is conditionally independent of other edges (e.g., between A and C, B and C) given the latent variables. The proposed LatentSNA builds upon existing CID methods by assuming conditional independence among connectivity edges. Moreover, it extends this framework by assuming conditional independence among outcome variables, akin to the psychometric Rasch model, and crucially, by assuming conditional independence between brain connectivity and outcome variables. LatentSNA represents the first SNA method to jointly account for dependencies among connectivity edges, among outcome variables, and between brain connectivity and outcomes.

In the following paragraphs, we outline the specific innovations of LatentSNA compared to existing state-of-the-art SNA methods and network-based brain analysis methods. While the current SNA methods mentioned above primarily focus on modeling single networks, brain connectivity networks can be viewed as multiplex networks with multiple layers of brain connections observed across a shared set of brain regions. To model these multiplex structures, Gollini and Murphy (2016) and D’Angelo et al. (2018) propose utilizing a shared set of latent variables across layers, assuming a joint relational structure across sets of connectivity. Salter-Townshend and McCormick (2017) and MacDonald et al. (2022) alternatively distinguish between shared and individual components across layers. In contrast to these approaches, LatentSNA captures individual differences in brain connectivity networks across layers and identifies specific

brain regions where the covariation between layers of brain networks and outcome variables is significant. LatentSNA makes a unique contribution to the existing literature on network-based brain analysis. Current network-based brain analysis methods typically involve incorporating individual behaviors and attributes as predictors to model the impact of these attributes on brain networks. This is often achieved by estimating regression coefficients that relate predictors to edge probabilities in brain networks, known as network response regression (e.g., Simpson et al., 2024; Wang et al., 2017; Xia et al., 2020; Zhou & Müller, 2022). Alternatively, some studies use brain connectivity networks themselves as predictors (e.g., Guha & Rodriguez, 2021), employing matrix and tensor operations to transform these networks into a linear form (e.g., Wang et al., 2021; Zhao et al., 2022), a method referred to as scalar-on-network regression (Morris et al., 2022).

In contrast to both network response regression and scalar-on-network regression, LatentSNA posits a shared data generation process that links brain connectivity and outcomes. LatentSNA offers several advantages over network response and scalar-on-network regressions by positing a shared data generation process for connectivity and outcomes. First, unlike regression models that typically assume one-directional relationships between brain and behaviors or outcomes—estimating either the impact of brain on behavior or vice versa—LatentSNA acknowledges the mutual relationship between them. Changes in the brain often correlate with changes in behavior, but neuroplasticity suggests that disordered behaviors and dysfunctional environments can also influence brain function over time. Second, in scalar-on-network and network response regressions, using brain connectivity (or behavior outcomes) as predictors assumes that these variables are fully observed. This assumption becomes problematic when data includes partially missing observations for brain connectivity and individual outcomes. Regression methods struggle to handle situations where data is incomplete for both brain connectivity and outcomes. Lastly, traditional regression methods lack robustness in estimating parameters related to brain connectivity or behavior when they do not simultaneously model the reciprocal influence between them. In contrast, LatentSNA integrates both brain and behavior within a unified modeling framework, allowing mutual information exchange during model estimation. In summary, LatentSNA addresses these limitations by proposing a shared data generation process for multiplex brain networks and individual outcomes, facilitating a more comprehensive understanding of their interrelationships.

LatentSNA introduces a novel application of network science through a generative statistical process, innovating social network analysis (SNA) concepts and methods for real-world networks and their connection to brain and behavior. In terms of modeling objectives and approaches, LatentSNA methodologically differs from graphical models (GMs). GMs are utilized to represent joint probability distributions and establish conditional dependencies between random variables (Edwards, 2000; Lauritzen, 1996; Murphy, 2001). In GMs, researchers construct a graph (network) based on whether or not two variables are conditionally independent of each other. For LatentSNA, the edges in a brain functional network indicate the strength and nature of co-activation between two brain regions over time. In contrast, edges in covariance networks for GMs represent the conditional dependence strength between variables. Crucially, covariance networks in GMs are derived from positive definite covariance matrices, specifying a particular type of relationship that is more constrained than real-world observed networks, including functional brain networks. Unlike GMs, SNAs accommodate a broader array of real-world network structures that are not feasible for covariance networks in GMs. Therefore, GMs and SNAs serve distinct purposes and are not directly comparable; latent variables in SNAs serve different functions than those in GMs. For further exploration of the differences between these methodologies, refer to discussions in Steinley (2021), Vitale et al. (2022), and Wang (2021).

To better clarify, we have relevant descriptions in the **Section Introduction** and **Supplementary Materials**

that read,

In the Section Introduction:

A critical challenge remains: connectivity edges are treated as independent observations, whereas evidence supports the dependent organization of brain networks as informative neurobiological indicators (Bassett & Sporns, 2017; Bullmore & Sporns, 2009; Zuo et al., 2012). Graphical models, consisting of both undirected Gaussian graphical models (e.g., Dobra et al., 2004; Dobra et al., 2011; Friedman et al., 2008; Friedman, 2004) and directed acyclic graphs (e.g., Friedman et al., 2000; Geiger & Heckerman, 2002), describe the conditional dependence among random variables and directly address the violation of the independence assumption. A key task of graphical models, when applied to neuroimaging data, is to estimate and create brain connectivity networks based on whether signals from two brain regions are conditionally independent of each other (Ni et al., 2022). Individual behaviors and outcomes can be incorporated to influence the estimation of brain connectivity networks (Ni et al., 2022; Ni et al., 2019). While the graphical model addresses the unrealistic independence assumption in most existing connectivity models, it (1) does not specifically address the unique topological structures of networks, as the method is built for multidimensional random variables, not necessarily with real-world network structures among them, (2) does not directly achieve neuroimaging biomarker detection, and (3) cannot be readily used for imaging-based outcome prediction.

What differentiates a network-science-driven analytic approach is that it draws on insights regarding the universality of the communicative structures of real-world networks (Barabási, 2013). Characteristics such as the small-world property and sparsity are universal properties found in social networks (Wohlgemuth, 2012), political networks (Wang & Edgerton, 2022), the World Wide Web (Fatta et al., 2016), and human connectomes (Amaral et al., 2000; Bassett & Bullmore, 2006; Dubitzky et al., 2013). Shared network architectures, as a result of being governed by universal principles (Barabási, 2013; Wohlgemuth, 2012), allow us to use a common set of mathematical and statistical instruments for network modeling. Analytic frameworks aimed at accommodating dependent and topological characteristics of networks are called statistical network-based analysis (SNA). See reviews on SNA methods (Desmarais & Cranmer, 2017; Goldenberg et al., 2010; Kim et al., 2018; Matias & Robin, 2014; Newman, 2003; Smith et al., 2019; Wasserman & Faust, 1994).

In the Supplementary Materials:

LatentSNA contributes to the current neuroimaging connectivity model literature by offering a high-power whole-brain SNA method for identifying brain-behavior links. Concentrating solely on localized effects may overlook the multifaceted nature of developing psychopathology during critical developmental stages and fail to capture substantial whole-brain changes due to widespread restructuring as the brain matures (Benes, 1989; Krasnegor et al., 1997; Paus et al., 1999). Disjointed and disconnected connectivity biomarkers, often coupled with inconsistencies across studies and low prediction accuracy and replicability, may be attributable to low statistical power. LatentSNA addresses these limitations, improves the discovery of the exact neurobiological mechanisms underlying childhood and adolescent psychopathology and encourages the development of effective interventions.

The proposed LatentSNA uniquely contributes to the current literature of SNA methods. It is a novel extension of latent variables-based SNA models for imaging biomarker detection. While the current SNA methods often focus on modeling single networks, brain connectivity networks can be seen as multiplex networks with multiple layers of brain connections observed on a shared set of brain regions. To model the multiplex structures, Gollini and Murphy (2016) and D’Angelo et al. (2018) propose a shared set of latent variables across layers and assume a joint relational structure across sets of connectivity. Salter-Townshend and McCormick (2017) and MacDonald et al. (2022) separate the shared and individual components across layers. Different from these approaches, we capture the individual differences in brain connectivity networks across layers and identify specific brain regions, where the covariation between layers of brain networks and the outcome variables is significant.

The LatentSNA represents a novel implementation of network science in a generative statistical process by innovating SNA concepts and methods for real-world networks and brain and behavior linking. In modeling objectives and approaches, the LatentSNA method is different from graphical models (GMs). A GM is used to represent joint probability distribution and construct conditional dependence between random variables (Edwards, 2000; Lauritzen, 1996; Murphy, 2001). Using GMs, researchers construct a graph (network) based on whether or not two variables are conditionally independent of each other. The edge in a brain functional network for LatentSNA represents whether/how strongly two brain regions co-activate across time; in contrast, the edge in the covariance network for GMs represents whether/how strongly two variables are conditional dependent. Crucially, the covariance networks in GMs are created out of the positive definite covariance matrices; their relationships are of a certain type, and they are more specific than the real-world observed networks including functional brain networks. In contrast to GMs, SNAs accommodate different types of real-world network structures, structures that are impossible to have for covariance networks in GMs. Therefore, GMs and SNAs are not comparable; latent variables in SNAs have different functions than latent variables in GMs. See further discussions about the differences between the two in Steinley (2021), Vitale et al. (2022), and Wang (2021).

3. The authors claimed that LatentSNA uses “the symmetric bilinear interaction effect to capture possible higher-order dependencies, among three or more connectivity edges.” However, Equation 0.1 seems to only be modeling the interaction between two nodes. The effect of this interaction is also not validated. What if this term is omitted from the model? How much degradation in performance, if any, will it cost. Is it possible to go beyond modeling bilinear interactions? How would modeling even higher-order dependencies affect the performance?

Response: Thank you for your comments. In response to concerns about the model’s capability to capture network topological structures and higher-order dependencies, we would like to clarify two points. First, Equation 0.1 in our model does not solely model pairwise interactions. As detailed in our analysis, Equation 0.1 also incorporates higher-order dependencies beyond pairwise interactions. Second, we will provide a detailed discussion on higher-order dependencies, particularly emphasizing that third-order dependence structures such as balance and clusterability contribute to relational patterns across the entire network, not just in triads.

First, we would like to confirm that the proposed LatentSNA method models higher-order dependencies and the interconnectedness among three or more brain regions. In Equation 0.1, we specify the probabilistic structure of the observed connectivity between pairs of brain regions u and v , where $u < v$, and $u, v =$

$1, 2, \dots, V$, following the standard practice to specify the probabilistic structure for each observation in a statistical model. This formulation is equivalent to the probabilistic structure of the whole network:

$$\mathbf{X}_i = a_i \mathbf{1}^T \mathbf{1} + \mathbf{w}_i \boldsymbol{\beta} \mathbf{1}^T \mathbf{1} + \mathbf{z}_i \mathbf{z}_i^T + \mathbf{E}_i, \quad (1)$$

where \mathbf{X}_i is the $V \times V$ brain connectivity adjacency matrix for person i ; $\mathbf{1}$ is a $V \times 1$ column vector of 1s; a_i is the fixed connectivity intercept for subject i ; \mathbf{z}_i is the $V \times 1$ vector of latent variable values for subject i ; and \mathbf{E}_i is the $V \times V$ matrix of errors. We adjusted for Q covariates, denoted by \mathbf{w}_i , with the first element set to 1, corresponding to the intercept, and their effects on the connectivity matrix characterized by $\boldsymbol{\beta}$. In sum, the formulation in Equation 0.1 represents the probabilistic structure for the entire network, accounting for topological structures and relational patterns across the entire network.

Figure 1: A triangle structure between nodes a , b and c for illustration.

We would like to demonstrate that Equation 0.1 models dependencies in the triangular structure among nodes a , b , and c , as illustrated in Figure 1, contradicting the notion that Equation 0.1 solely captures pairwise interactions. The triangular structure indicates the presence of third-order dependence (Wasserman & Faust, 1994)—the connectivity edges between brain regions a and b , a and c , and b and c are interrelated through their shared triad of regions a , b , and c .

According to Equation 0.1, the probability of an edge between brain regions a and b depends on z_a and z_b , and similarly, the probability of connectivity between brain regions a and c depends on z_a and z_c . These probabilities are not independent of the probability of an edge between brain regions b and c , as z_b and z_c also influence the probability of connectivity between brain regions b and c , in addition to their effects on the connectivities between a and b , and a and c . In other words, the probabilities of connectivity among brain regions a , b , and c are interconnected, and this interconnectedness is captured by the model in Equation 0.1.

In the above, we have provided a simplified illustration contradicting the idea that Equation 0.1 only models pairwise interactions. In the following, we would like to provide detailed discussions about higher-order dependencies and bilinear effects. According to Wasserman and Faust (1994), higher-order dependencies in real-world networks are defined as dependencies among three (triad) or more nodes. Common examples of higher-order dependencies in real-world networks include homophily, balance, and clusterability (Heider, 1944; McPherson et al., 2001; Norman et al., 1965; Wasserman & Faust, 1994).

Homophily is often associated with the transitive property of a network, where new connections are established based on existing connections. Balance suggests a state of harmony, where positive connections

are found among nodes with similar attributes, and negative connections are found among nodes with divergent attributes. Clusterability represents a relaxed criterion compared to balance (Hoff, 2005a, see examples of balanced and clusterable triads in Figure 2A). Balanced cycles among triads can divide the whole network into cohesive groups, where $x_{u,v} > 0$ if nodes u and v are in the same group, and $x_{u,v} < 0$ if they are in opposite groups (Harary, 1953; Hoff, 2005a). Therefore, the presence of higher-order dependencies such as balance contributes to relational patterns and topology across the entire network. By modeling higher-order dependencies, the proposed LatentSNA captures these relational patterns across the entire network.

Bilinear effects account for transitive, balanced, and clusterable network structures (Hoff, 2005a, 2008a). Theoretical and empirical evidence supports that bilinear effects models capture higher-order dependencies in dependent networks, as documented in the existing Social Network Analysis (SNA) literature. According to Hoff (2008a), vector-product based latent space models, which include bilinear effects, effectively capture higher-order dependencies such as homophily, balance, and clusterability. Moreover, Sosa and Buitrago (2021) demonstrate that such models exhibit satisfactory model fit for networks with varying degrees of transitivity and clusterability. Given that brain functional networks are known to exhibit small-world properties (Bassett & Bullmore, 2006), which often include both transitivity and clusterability, it is optimal for us to utilize bilinear effects to model these higher-order dependence structures.

To better clarify the focus on the innovation of the method, we have relevant descriptions in the **Section Conceptual framework** that read,

In the **Section Conceptual framework**:

LatentSNA builds on the statistical network modeling literature and preserves the topological structure of the brain network (Figure 2C). In Wasserman and Faust (1994), higher-order dependencies in real-world networks are defined as dependencies among three (triad) or more nodes. Common examples of higher-order dependencies in real-world networks include homophily, balance, and clusterability (Heider, 1944; McPherson et al., 2001; Norman et al., 1965; Wasserman & Faust, 1994). Homophily is often associated with the transitive property of a network, explaining how new connections are established based on existing connections, also known as transitivity. Balance suggests a state of harmony, where positive connections are found among nodes with similar attributes, and negative connections are found among nodes with divergent attributes. Clusterability represents a more relaxed criteria for harmony than balance (Hoff, 2005a). With balanced cycles among triads, the entire network can be divided into cohesive groups with $x_{u,v} > 0$ if nodes u and v are in the same group, and $x_{u,v} < 0$ if they are in opposite groups (Harary, 1953; Hoff, 2005a). Therefore, the presence of higher-order dependencies such as balance contributes to relational patterns and topology across the whole network, including higher-order dependencies. By modeling higher-order dependencies, the proposed LatentSNA captures relational patterns across the entire network.

Bilinear effects account for transitive, balanced, and clusterable network structures (Hoff, 2005a, 2008a). In Hoff (2008a), the author provides theoretical evidence that vector-product-based latent space models, which include bilinear effects models, capture higher-order dependencies such as homophily, balance, and clusterability. Furthermore, in Sosa and Buitrago (2021), the authors demonstrate that such models show satisfactory model fit for networks with varying degrees of transitivity and clusterability. Given that brain functional networks are known to possess small-world properties (Bassett & Bullmore, 2006), likely exhibiting both transitivity and clusterability, it is optimal for us to use bilinear effects to model higher-order

dependence structures. Consequently, LatentSNA uniquely captures how network topology is implicated in brain dysfunction (Figure 2C).

4. The code is not accessible, making it more challenging to assess the validity of the method.

Response: Thank you for your comment. We have modified the published code, and shared the repository via the user-friendly github at <https://github.com/selenashuowang/latentSNA> with tutorial. The repository will be available via CRAN pending review.

To better clarify, we have relevant descriptions in the **Section Methods** that read,

In the **Section Methods**:

The estimation algorithm for this paper was implemented in the open sourced programming language for statistical computing and graphics, R. The code is available upon the user-friendly github at <https://github.com/selenashuowang/latentSNA> with tutorial. The repository will be available via CRAN pending review.

5. Figure 1 is poorly done. It does not clearly explain the core concepts of LatentSNA. Notations are not defined, making it impossible to make any sense out of the figure.

Response: Following your comment, we have revised Figure 1 of the manuscript (Figure 2 in the letter appendix), added labels for key components of the schematic diagram and provided explanations for each component in the caption.

To better clarify, we have added relevant **figure caption** and descriptions in the **Section Conceptual framework** that read,

In the **figure caption**:

The schematic diagram of LatentSNA. A. LatentSNA allows multivariate modeling of latent outcome variable (e.g., internalizing psychopathology) with multiple components (e.g., anxious-depressed, withdrawn-depressed and somatic complaints to improve precision. The observed psychopathology is generated following a modified version of a psychometric rasch model (Fischer & Molenaar, 2012), where outcomes are decomposed into the item and person components. B. LatentSNA uses the symmetric bilinear interaction effect to capture network topology (transitivity, balance and clusterability) (Hoff, 2008b; Hoff, 2005b). C. LatentSNA-estimated internalizing network shows star-like structure (Guimerà et al., 2002; Sawai, 2012, 2014). In contrast, this topological structure is lost when the the internalizing network is estimated via marginal association tests. D. LatentSNA makes inference of the relationships between brain and behavior, e.g., internalizing psychopathology. We propose a joint latent variable model, where we allow the latent connectivity variables, \mathbf{Z} and latent behavior variables, $\boldsymbol{\theta}$ to covary with a shared covariance matrix, $\boldsymbol{\Sigma}$. E. The LatentSNA Bayesian diagram demonstrates a wholistic model for multivariate outcomes \mathbf{Y} and brain networks \mathbf{X} . Intercepts a and error term E are part of the data generation process for connectivity; and intercepts b and error term Ψ are part of the data generation process for behavior. Circles represent unknown quantities of the model, and squares represent the observed data. Non-informative priors are not included in this diagram.

Figure 2: The schematic diagram of LatentSNA. A. LatentSNA allows multivariate modeling of a latent outcome variable (e.g., internalizing psychopathology) with multiple components (e.g., anxious-depressed, withdrawn-depressed, and somatic complaints) to improve precision. The observed psychopathology is generated following a modified version of a psychometric Rasch model (Fischer & Molenaar, 2012), where outcomes are decomposed into item and person components. B. LatentSNA uses the symmetric bilinear interaction effect to capture network topology (transitivity, balance, and clusterability) (Hoff, 2008b; Hoff, 2005b). C. LatentSNA-estimated internalizing network shows a star-like structure (Guimerà et al., 2002; Sawai, 2012, 2014). In contrast, this topological structure is lost when the internalizing network is estimated via marginal association tests. D. LatentSNA makes inference of the relationships between brain and behavior, e.g., internalizing psychopathology and functional connectivity. We propose a joint latent variable model, where we allow the latent connectivity variables, \mathbf{Z} , and latent behavior variables, $\boldsymbol{\theta}$, to co-vary with a shared covariance matrix, $\boldsymbol{\Sigma}$. F. The LatentSNA Bayesian diagram demonstrates a holistic model for multivariate outcomes \mathbf{Y} and brain networks \mathbf{X} . Intercepts a and error term E are part of the data generation process for connectivity, and intercepts b and error term Ψ are part of the data generation process for behavior. Circles represent unknown quantities of the model, and squares represent the observed data. Non-informative priors are not included in this diagram.

In the Section Conceptual framework:

Motivated by the need to enhance the power for identifying imaging biomarkers, we propose *LatentSNA* as a generative SNA model to identify significant links between brain networks and behavioral traits (Figure 2). Existing SNA models often analyze brain-behavior links as one-sided regression models. These models either analyze the (reduced dimension) brain connectivity as predictors in a regression with behavior as the response (Wang et al., 2021; Zhao et al., 2022), or they analyze connectivity as the response in a matrix-response regression to quantify behavioral covariate effects (Chen et al., 2023; Shi & Guo, 2016). However, both types of models lack the ability to capture the mutual variations between behavioral profiles and brain variations, i.e., brain development influences children’s behavior, and abnormal behaviors potentially reinforce brain abnormalities due to brain plasticity (Johnston, 2004; Kolb, 2013; Kolb & Whishaw, 1998). In contrast, *LatentSNA* allows connectivity differences to inform behavior/outcome variations, and vice versa—both brain connectivity (Figure 2B) and individual outcomes (Figure 2A) are the targeted modeling interests. *LatentSNA* is ideal for detecting complicated and potentially noisy and weak signals hidden in high-dimensional functional connectivity data, e.g., high heterogeneity and strong motion artifacts in children’s fMRI data (Calhoun & Sui, 2016; Chahal et al., 2020). *LatentSNA* reinforces potentially weak signals in connectivity with a two-way brain-behavior linking module (Figure 2D) that allows true connectivity signals and true internalizing signals (Figure 2A) to mutually inform each other, thus strengthening connectivity signals. Additionally, *LatentSNA* partials out random noise variations from true signal variations to further reinforce potentially weak connectivity signals.

Second, focused on inferring the relationships between brain networks and behaviors, *LatentSNA* is, philosophically, an inference model (also called explanatory models), not a prediction model (Breiman, 2001; Shmueli, 2010; Shmueli & Koppius, 2011). *LatentSNA* provides uncertainty quantification for biomarker detection and robust statistical inference under the Bayesian framework (Figure 2E). Inference models are built to describe how potential predictors and explanatory variables explain individual differences in the responses, while prediction models ignore this process and focus on accurately predicting future responses. Inference models rely on statistical theories such as the central limit theorem and the large sample properties to derive unbiased estimates of the significant effect coefficients with controlled Type I error, while prediction models often introduce biases to improve prediction. Inference models are more optimal for detecting imaging biomarkers as they allow us to quantify the uncertainty associated with the identification of imaging biomarkers, which is not possible with prediction models. With a large enough sample size, our model can, in an unbiased way, identify true mutual relationships between each region’s connectivity and individual outcomes with high enough power and controlled Type I error. Meanwhile, machine learning methods such as LASSO (Tibshirani, 1996) do not offer unbiased quantification of the relationships, suffer from low power, and inflated Type II errors.

Third, *LatentSNA* builds on the statistical network modeling literature and preserves the topological structure of the brain network (Figure 2C). In Wasserman and Faust (1994), higher-order dependencies in real-world networks are defined as dependencies among three (triad) or more nodes. Common examples of higher-order dependencies in real-world networks include homophily, balance, and clusterability (Heider, 1944; McPherson et al., 2001; Norman et al., 1965; Wasserman & Faust, 1994). Homophily is often associated with the transitive property of a network, explaining how new connections are established based on existing connections, also known as transitivity. Balance suggests a state of harmony, where positive connections are

found among nodes with similar attributes, and negative connections are found among nodes with divergent attributes. Clusterability represents a more relaxed criteria for harmony than balance (Hoff, 2005a). With balanced cycles among triads, the entire network can be divided into cohesive groups with $x_{u,v} > 0$ if nodes u and v are in the same group, and $x_{u,v} < 0$ if they are in opposite groups (Harary, 1953; Hoff, 2005a). Therefore, the presence of higher-order dependencies such as balance contributes to relational patterns and topology across the whole network, including higher-order dependencies. By modeling higher-order dependencies, the proposed LatentSNA captures relational patterns across the entire network.

Bilinear effects account for transitive, balanced, and clusterable network structures (Hoff, 2005a, 2008a). In Hoff (2008a), the author provides theoretical evidence that vector-product-based latent space models, which include bilinear effects models, capture higher-order dependencies such as homophily, balance, and clusterability. Furthermore, in Sosa and Buitrago (2021), the authors demonstrate that such models show satisfactory model fit for networks with varying degrees of transitivity and clusterability. Given that brain functional networks are known to possess small-world properties (Bassett & Bullmore, 2006), likely exhibiting both transitivity and clusterability, it is optimal for us to use bilinear effects to model higher-order dependence structures. Consequently, LatentSNA uniquely captures how network topology is implicated in brain dysfunction (Figure 2C).

6. On page 9 it is unclear what is meant by “the Averaging method simply takes the sample average connectivity as a prediction for a new subject’s connectivity.”

Response: Thank you for your comment. Following your suggestion, we would like to detail the procedure for predicting connectivity using the Average method in independent samples. The Average method and its extensions represent one of the most common approaches to capturing group-level connectivity and conducting subsequent analyses (Achard et al., 2006; Sinke et al., 2016; Song et al., 2009), often yielding satisfactory prediction accuracy (e.g., Paquette et al., 2016). First, we randomly divide the connectivity data into ten equal parts, using one set of data points as the test data and the remaining sets as the training data. We then compute the group-level connectivity by averaging the entries of individual connectivity matrices within the training data. Subsequently, we predict connectivity in the test set using the estimated connectivity from the training data. Figure 3A in the supplementary materials illustrates the average out-of-sample correlations between observed and predicted connectivity values across 100 random samples.

Our findings indicate that LatentSNA achieves satisfactory prediction accuracy for brain connectivity using individual-level estimates, outperforming the Average method particularly when the signal proportion is substantial. When predicting connectivity using group-level estimates, both LatentSNA and the Average method demonstrate satisfactory performance for the entire network, with LatentSNA showing superior performance for regions strongly associated with behavioral outcomes.

To better clarify the focus on the innovation of the method, we have relevant descriptions in the **Section Comparison method** that read,

In the **Section Comparison method**:

The Average method and its extensions represent one of the most common methods to capture group-level connectivity and to perform subsequent analysis (Achard et al., 2006; Sinke et al., 2016; Song et al., 2009) often with satisfactory prediction accuracy (e.g., Paquette et al., 2016). We first randomly divide the connectivity data into ten equal sizes, using one set of data

points as the test data and other sets of data points as the training data. We then capture the group-level connectivity using the entry-wise sample mean of individual connectivity matrices in the training data. We perform predictions for connectivity in the test set using estimated connectivity from the training data. Figure 3A in the supplementary materials shows the average out-of-sample correlations between the observed and predicted connectivity values across 100 random samples. Our results suggest that LatentSNA shows satisfactory prediction accuracy for the brain connectivity using individual-level estimates, and it outperforms the Average method when the signal proportion is large. When predicting connectivity using the group-level estimates, LatentSNA and the Average method both show satisfactory performance for the whole graph; and LatentSNA outperforms the Average method for regions with strong relational ties with behavior.

Reviewer 2

A. The LatentSNA framework proposed in this paper combines brain functional connectivity with internalizing psychopathology. In particular, the proposed model was applied to the ABCD study, which achieved sufficiently accurate prediction of both children's internalizing traits and functional connectivity, and substantially improved the ability to explain the individual internalizing differences compared with current approaches. B. This original work is of great significance, especially in detecting the neurologically meaningful network topology in the adolescents and children population. C. Both simulation studies and real applications to the ABCD study were conducted to demonstrate the outperformance of the proposed method. D. The Bayesian framework and the proposed MCMC algorithm are solid and convincing. E. The findings provided by latentSNA are consistent with existing literature. G. Previous related works are mostly cited in this paper. H. The whole paper is well written and organized.

We really appreciate Reviewer 2 for the positive comments and for taking the time to review our work. We have made changes according to your suggestions and our response to each comment is as follows.

1. Figure 1 shows the big picture of the LatentSNA. However, it is not informative and is poorly illustrated and explained in the caption contents.

Response: Following your comment, we have revised Figure 1 of the manuscript (Figure 2 in the letter), added labels for key components of the schematic diagram and provided explanations for each component in the caption.

To better clarify, we have added relevant **figure caption** and descriptions in the **Section Conceptual framework** that read,

In the **figure caption**:

The schematic diagram of LatentSNA. A. LatentSNA allows multivariate modeling of a latent outcome variable (e.g., internalizing psychopathology) with multiple components (e.g., anxious-depressed, withdrawn-depressed, and somatic complaints) to improve precision. The observed psychopathology is generated following a modified version of a psychometric Rasch model (Fischer & Molenaar, 2012), where outcomes are decomposed into item and person components. B.

LatentSNA uses the symmetric bilinear interaction effect to capture network topology (transitivity, balance, and clusterability) (Hoff, 2008b; Hoff, 2005b). C. LatentSNA-estimated internalizing network shows a star-like structure (Guimerà et al., 2002; Sawai, 2012, 2014). In contrast, this topological structure is lost when the internalizing network is estimated via marginal association tests. D. LatentSNA makes inference of the relationships between brain and behavior, e.g., internalizing psychopathology and functional connectivity. We propose a joint latent variable model, where we allow the latent connectivity variables, \mathbf{Z} , and latent behavior variables, $\boldsymbol{\theta}$, to co-vary with a shared covariance matrix, $\boldsymbol{\Sigma}$. F. The LatentSNA Bayesian diagram demonstrates a holistic model for multivariate outcomes \mathbf{Y} and brain networks \mathbf{X} . Intercepts a and error term E are part of the data generation process for connectivity, and intercepts b and error term Ψ are part of the data generation process for behavior. Circles represent unknown quantities of the model, and squares represent the observed data. Non-informative priors are not included in this diagram.

In the Section Conceptual framework:

Motivated by the need to enhance the power for identifying imaging biomarkers, we propose LatentSNA as a generative SNA model to identify significant links between brain networks and behavioral traits (Figure 2). Existing SNA models often analyze brain-behavior links as one-sided regression models. These models either analyze the (reduced dimension) brain connectivity as predictors in a regression with behavior as the response (Wang et al., 2021; Zhao et al., 2022), or they analyze connectivity as the response in a matrix-response regression to quantify behavioral covariate effects (Chen et al., 2023; Shi & Guo, 2016). However, both types of models lack the ability to capture the mutual variations between behavioral profiles and brain variations, i.e., brain development influences children’s behavior, and abnormal behaviors potentially reinforce brain abnormalities due to brain plasticity (Johnston, 2004; Kolb, 2013; Kolb & Whishaw, 1998). In contrast, LatentSNA allows connectivity differences to inform behavior/outcome variations, and vice versa—both brain connectivity (Figure 2B) and individual outcomes (Figure 2A) are the targeted modeling interests. LatentSNA is ideal for detecting complicated and potentially noisy and weak signals hidden in high-dimensional functional connectivity data, e.g., high heterogeneity and strong motion artifacts in children’s fMRI data (Calhoun & Sui, 2016; Chahal et al., 2020). LatentSNA reinforces potentially weak signals in connectivity with a two-way brain-behavior linking module (Figure 2D) that allows true connectivity signals and true internalizing signals (Figure 2A) to mutually inform each other, thus strengthening connectivity signals. Additionally, LatentSNA partials out random noise variations from true signal variations to further reinforce potentially weak connectivity signals.

Second, focused on inferring the relationships between brain networks and behaviors, LatentSNA is, philosophically, an inference model (also called explanatory models), not a prediction model (Breiman, 2001; Shmueli, 2010; Shmueli & Koppius, 2011). LatentSNA provides uncertainty quantification for biomarker detection and robust statistical inference under the Bayesian framework (Figure 2E). Inference models are built to describe how potential predictors and explanatory variables explain individual differences in the responses, while prediction models ignore this process and focus on accurately predicting future responses. Inference models rely on statistical theories such as the central limit theorem and the large sample properties to derive unbiased estimates of the significant effect coefficients with controlled Type I error, while prediction models often introduce biases to improve prediction. Inference models are more optimal for detecting imaging biomarkers as they allow us to quantify the uncertainty

associated with the identification of imaging biomarkers, which is not possible with prediction models. With a large enough sample size, our model can, in an unbiased way, identify true mutual relationships between each region's connectivity and individual outcomes with high enough power and controlled Type I error. Meanwhile, machine learning methods such as LASSO (Tibshirani, 1996) do not offer unbiased quantification of the relationships, suffer from low power, and inflated Type II errors.

Third, LatentSNA builds on the statistical network modeling literature and preserves the topological structure of the brain network (Figure 2C). In Wasserman and Faust (1994), higher-order dependencies in real-world networks are defined as dependencies among three (triad) or more nodes. Common examples of higher-order dependencies in real-world networks include homophily, balance, and clusterability (Heider, 1944; McPherson et al., 2001; Norman et al., 1965; Wasserman & Faust, 1994). Homophily is often associated with the transitive property of a network, explaining how new connections are established based on existing connections, also known as transitivity. Balance suggests a state of harmony, where positive connections are found among nodes with similar attributes, and negative connections are found among nodes with divergent attributes. Clusterability represents a more relaxed criteria for harmony than balance (Hoff, 2005a). With balanced cycles among triads, the entire network can be divided into cohesive groups with $x_{u,v} > 0$ if nodes u and v are in the same group, and $x_{u,v} < 0$ if they are in opposite groups (Harary, 1953; Hoff, 2005a). Therefore, the presence of higher-order dependencies such as balance contributes to relational patterns and topology across the whole network, including higher-order dependencies. By modeling higher-order dependencies, the proposed LatentSNA captures relational patterns across the entire network.

Bilinear effects account for transitive, balanced, and clusterable network structures (Hoff, 2005a, 2008a). In Hoff (2008a), the author provides theoretical evidence that vector-product-based latent space models, which include bilinear effects models, capture higher-order dependencies such as homophily, balance, and clusterability. Furthermore, in Sosa and Buitrago (2021), the authors demonstrate that such models show satisfactory model fit for networks with varying degrees of transitivity and clusterability. Given that brain functional networks are known to possess small-world properties (Bassett & Bullmore, 2006), likely exhibiting both transitivity and clusterability, it is optimal for us to use bilinear effects to model higher-order dependence structures. Consequently, LatentSNA uniquely captures how network topology is implicated in brain dysfunction (Figure 2C).

2. More discussions about the differences between LatentSNA (Z) and LatentSNA (THETA) are expected.

Response: Thank you for your comment. Following your suggestion, we include more discussions about the difference between LatentSNA (Z) and LatentSNA (THETA). The difference between LatentSNA (Z) and LatentSNA (THETA) lies in how predictions are performed. For LatentSNA (THETA), predicting the behavior outcome of a new subject amounts to additional draws for each new y_i from a distribution with probability determined by the model. For LatentSNA (Z), on the other hand, predicting the behavior outcome of a new subject is based on the estimated latent connectivity variable \mathbf{Z} from the training data. We evaluated the out-of-sample predictive performance for LatentSNA (Z) and LatentSNA (THETA) as follows:

- We randomly sampled 100 subjects and their behavior outcome as the test data, and other sets of data points as the training data.

- We fitted the training data to LatentSNA, obtained the posterior mean of the model parameters.
- For LatentSNA (THETA),
 - Predicting the behavior outcome of a new subject amounts to additional draws for each new y_i from a distribution with probability determined by the model.
 - The full conditional of the new observations $\mathcal{Y}^{(test)}$ is, for any $y_i \in \mathcal{Y}^{(test)}$, determined by $\pi(y_i|\theta, b_i, \Psi_i)$.
- For LatentSNA (Z),
 - Predicting the behavior outcome of a new subject is based on the estimated latent connectivity variable, $\hat{\mathbf{Z}}$ from the training data.
 - We first selected significant imaging biomarkers based on 95% posterior credible intervals of the covariance parameters and used significant imaging biomarkers' latent connectivity variables as predictors.
 - Second, we split the estimated latent connectivity variables into the test set, $\hat{\mathbf{Z}}^{(test)}$ and the training set, $\hat{\mathbf{Z}}^{(train)}$ following the split of the data.
 - Third, we fitted the training model using $\hat{\mathbf{Z}}^{(train)}$ as the predictors and the observed psychopathology outcomes for the training subjects as the response.
 - We obtained the estimated regression coefficients, $\hat{\beta}$ based on the training model.
 - Lastly, we predicted the psychopathology outcome of a new subject, for any $y_i \in \mathcal{Y}^{(test)}$, under LatentSNA (Z) following $y_i = \hat{\beta} \times \hat{\mathbf{Z}}^{(test)}$.

We repeated the process 10 times.

To better clarify the focus on the innovation of the method, we have relevant descriptions in the **Section Predicting outcomes** that read,

In the **Section Predicting outcomes**:

For LatentSNA (THETA), predicting the behavior outcome of a new subject amounts to additional draws for each new y_i from a distribution with probability determined by the model. For LatentSNA (Z), on the other hand, predicting the behavior outcome of a new subject is based on the estimated latent connectivity variable \mathbf{Z} from the training data. We evaluated the out-of-sample predictive performance for LatentSNA (Z) and LatentSNA (THETA) as follows:

- *We randomly sampled 100 subjects and their behavior outcome as the test data, and other sets of data points as the training data.*
- *We fitted the training data to LatentSNA, obtained the posterior mean of the model parameters.*
- *For LatentSNA (THETA),*
 - *Predicting the behavior outcome of a new subject amounts to additional draws for each new y_i from a distribution with probability determined by the model.*
 - *The full conditional of the new observations $\mathcal{Y}^{(test)}$ is, for any $y_i \in \mathcal{Y}^{(test)}$, determined by $\pi(y_i|\theta, b_i, \Psi_i)$.*
- *For LatentSNA (Z),*
 - *Predicting the behavior outcome of a new subject is based on the estimated latent connectivity variable, $\hat{\mathbf{Z}}$ from the training data.*

- We first selected significant imaging biomarkers based on 95% posterior credible intervals of the covariance parameters and used significant imaging biomarkers’ latent connectivity variables as predictors.
- Second, we split the estimated latent connectivity variables into the test set, $\hat{\mathbf{Z}}^{(test)}$ and the training set, $\hat{\mathbf{Z}}^{(train)}$ following the split of the data.
- Third, we fitted the training model using $\hat{\mathbf{Z}}^{(train)}$ as the predictors and the observed psychopathology outcomes for the training subjects as the response.
- We obtained the estimated regression coefficients, $\hat{\beta}$ based on the training model.
- Lastly, we predicted the psychopathology outcome of a new subject, for any $y_i \in \mathcal{Y}^{(test)}$, under LatentSNA (Z) following $y_i = \hat{\beta} \times \hat{\mathbf{Z}}^{(test)}$.

We repeated the process 10 times. Figure 3B in the manuscript shows the out-of-sample correlations between the observed and predicted internalizing values on the test data using LatentSNA (Z) and LatentSNA (THETA). Between LatentSNA (Z) and LatentSNA (THETA), the former does not directly—but indirectly—incorporate training internalizing information for prediction while the latter does. This shows that by constructing joint learning mechanisms using LatentSNA, we can effectively predict internalizing profiles for new subjects based on the available data.

3. What kind of cross-validation approach is adopted for the prediction tasks?

Response: Thank you for your comment. Following your suggestion, we provide a detailed discussion about the out-of-sample prediction procedure for predicting $\mathcal{Y}^{(test)}$ in the previous response. In addition, we would like to provide a detailed discussion about the out-of-sample prediction procedure for predicting connectivity. We evaluate the out-of-sample performance for predicting connectivity of new subjects as follows:

- We randomly sample 100 subjects and their connectivity values as the test data, and other part of data points as the training data.
- We fit the training data to LatentSNA, obtain the posterior mean of the model parameters.
- Predicting the connectivity of a new subject amounts to additional draws for each missing $x_i \in \mathcal{X}^{(test)}$ from a distribution with probability determined by the model.
- The full conditional of the new observations $\mathcal{X}^{(test)}$ is, for any $x_i \in \mathcal{X}^{(test)}$, determined by $\pi(x_i | z_i, a_i, \mathbf{E}_i)$.

We repeated the process 10 times. Figure 3C shows the average out-of-sample correlations between the observed and predicted connectivity values on the test data for predicting the whole graph, top ten internalizing regions and top 5 internalizing regions. The results show that LatentSNA provides sufficiently accurate prediction of the connectivity measurements, posing a unique opportunity to uncover brain connectivity for new subjects incorporating their internalizing measures.

To better clarify the focus on the innovation of the method, we have relevant descriptions in the **Section Predicting connectivity** that read,

In the **Section Predicting connectivity**:

We evaluate the out-of-sample performance for predicting connectivity of new subjects as follows:

- We randomly sampled 100 subjects and their connectivity values as the test data, and other part of data points as the training data.
- We fitted the training data to LatentSNA, obtained the posterior mean of the model parameters.
- Predicting the connectivity of a new subject amounts to additional draws for each missing $x_i \in \mathcal{X}^{(test)}$ from a distribution with probability determined by the model.
- The full conditional of the new observations $\mathcal{X}^{(test)}$ is, for any $x_i \in \mathcal{X}^{(test)}$, determined by $\pi(x_i|z_i, a_i, \mathbf{E}_i)$.

We repeated the process 10 times. Figure 3C shows the average out-of-sample correlations between the observed and predicted connectivity values on the test data for predicting the whole graph, top ten internalizing regions and top 5 internalizing regions. The results show that LatentSNA provides sufficiently accurate prediction of the connectivity measurements, posing a unique opportunity to uncover brain connectivity for new subjects incorporating their internalizing measures.

4. I would like to see the sensitivity of the prior settings in the MCMC algorithm.

Response: Thank you for your comment. We would like to provide a detailed discussion about the sensitivity of the prior for the covariance matrix Σ . The identification of the imaging biomarkers is based on whether the estimated covariances between connectivity and behavior are significantly different from zero. Therefore, it is of interest to expand on the sensitivity of the prior specification of the covariance parameters.

In LatentSNA, the approximation of the posterior distribution of Σ is facilitated by setting a prior distribution of inverse Wishart($\mathbf{I}_{V+1}, V+1+2$) with identify scale matrix, $\mathbf{S}_0 = \mathbf{I}$ and degree of freedom equaling to $m_0 = V + 1 + 2$. The use of an inverse Wishart (IW) distribution as a prior for the variance-covariance parameter matrix is fairly common in Bayesian analysis; see discussions in Barnard et al. (2000), Leonard and Hsu (1992), and Zhang (2021). The IW prior is a conjugate prior for the covariance matrix of the normal data. In LatentSNA, we are interested in estimating the covariance matrix Σ of the joint distribution of the latent connectivity and behavior variables, $\mathbf{D} = (z_{1,i}, z_{2,i}, \dots, z_{V,i}, \theta_i)$. With the IW prior, the posterior distribution of Σ can be obtained through the Bayes' Theorem:

$$p(\Sigma|\mathbf{D}) = \frac{p(\mathbf{D}|\Sigma)p(\Sigma)}{p(\mathbf{D})}. \quad (2)$$

From it, we can obtain the posterior distribution of Σ with the specified prior distribution as

$$\Sigma|\mathbf{D} \sim IW(\mathbf{S}_0 + \mathbf{F}'^T \mathbf{F}', m_0 + 2), \quad (3)$$

where \mathbf{F}' as a $N \times (V + 1)$ matrix with i th row as (z_i^T, θ_i^T) . Therefore, the posterior mean of Σ is a weighted average of the sample covariance matrix $\mathbf{F}'^T \mathbf{F}'$ and the prior mean \mathbf{S}_0 . When the sample size $N \rightarrow \infty$, the posterior mean approaches the sample mean.

In a sensitivity analysis conducted by Zhang (2021), the author sets the scale matrix as identity and vary the degrees of freedom by increasing m_0 . With the increase of m_0 , the posterior means become smaller and the posterior variances also become smaller. Thus, given the large sample size in the data, we expect the posterior mean of Σ to approach the sample mean.

To better clarify the focus on the innovation of the method, we have relevant descriptions in the **Section Estimation** that read,

In the **Section Estimation**:

The identification of the imaging biomarkers is based on whether the estimated covariances between connectivity and behavior are significantly different from zero. Therefore, it is of interest to expand on the sensitivity of the prior specification of the covariance parameters.

In LatentSNA, the approximation of the posterior distribution of Σ is facilitated by setting a prior distribution of inverse Wishart($\mathbf{I}_{V+1}, V + 1 + 2$) with identify scale matrix, $\mathbf{S}_0 = \mathbf{I}$ and degree of freedom equaling to $m_0 = V + 1 + 2$. The use of an inverse Wishart (IW) distribution as a prior for the variance-covariance parameter matrix is fairly common in Bayesian analysis; see discussions in Barnard et al. (2000), Leonard and Hsu (1992), and Zhang (2021). The IW prior is a conjugate prior for the covariance matrix of the normal data. In LatentSNA, we are interested in estimating the covariance matrix Σ of the joint distribution of the latent connectivity and behavior variables, $\mathbf{D} = (z_{1,i}, z_{2,i}, \dots, z_{V,i}, \theta_i)$. With the IW prior, the posterior distribution of Σ can be obtained through the Bayes' Theorem:

$$p(\Sigma|\mathbf{D}) = \frac{p(\mathbf{D}|\Sigma)p(\Sigma)}{p(\mathbf{D})}. \quad (4)$$

From it, we can obtain the posterior distribution of Σ with the specified prior distribution as

$$\Sigma|\mathbf{D} \sim IW(\mathbf{S}_0 + \mathbf{F}'^T \mathbf{F}', m_0 + 2), \quad (5)$$

where \mathbf{F}' as a $N \times (V + 1)$ matrix with i th row as (z_i^T, θ_i^T) . Therefore, the posterior mean of Σ is a weighted average of the sample covariance matrix $\mathbf{F}'^T \mathbf{F}'$ and the prior mean \mathbf{S}_0 . When the sample size $N \rightarrow \infty$, the posterior mean approaches the sample mean.

In a sensitivity analysis conducted by Zhang (2021), the author sets the scale matrix as identity and vary the degrees of freedom by increasing m_0 . With the increase of m_0 , the posterior means become smaller and the posterior variances also become smaller. Thus, given the large sample size in the data, we expect the posterior mean of Σ to approach the sample mean.

5. The mechanism for generating simulation data is not very clear.

Response: Thank you for your comment. Following your suggestion, we would like to provide a detailed discussion about the data generation process for the simulation. The data generation process for the simulation was as follows. For simplicity and consistency, the number of behavior variables was assigned as one in all generated data. We first generated the connectivity latent variables as well as the latent behavior variables from the multivariate normal distribution with the mean zero and pre-defined covariance matrix with unit variances. To conduct a comprehensive assessment of the model performance, we created a range of data situations with varying sample sizes, connectivity scale, signal to noise ratio and signal proportions. To assess the model's ability to accurately identify true imaging biomarkers for outcomes that have both strong and weak biological signals, we varied the amount of true signals in the data by assigning signal proportion to 0.1 and 0.3. When the signal proportion equals to 0.1(0.3), we randomly assigned 10% (30%) of the covariance parameters between connectivity and behavior to be non-zero. To ensure the positive definiteness of Σ , we assigned both the covariances between connectivity and behavior and the

corresponding dimensions in the latent connectivity covariance matrix as 0.9. We randomly sampled the errors for the connectivity from a normal distribution with mean 0 and variance defined by the signal-to-noise (S/N) ratio. The errors for the behavior were sampled from the normal distribution with the mean 0 and variance 0.5.

We considered three sample sizes, $N = 500$, $N = 1,000$ and $N = 2,000$ and two conditions for the number of nodes $V = 20$ and $V = 70$, and we specified two levels of the S/N ratio, 0.5 and 1 controlled by the error variance while keeping the variance of the latent variables constant. The individual-specific intercepts for connectivity and behavior were set as 0. In total, we considered 24 different scenarios combining from different signal proportions, sample sizes, node numbers and S/Ns. Under each scenario, we simulated the 100 data.

To better clarify, we have relevant descriptions in the **Section Simulation** that read,

In the **Section Simulation** :

The data generation process for the simulation was as follows. For simplicity and consistency, the number of behavior variables was assigned as one in all generated data. We first generated the connectivity latent variables as well as the latent behavior variables from the multivariate normal distribution with the mean zero and pre-defined covariance matrix with unit variances. To conduct a comprehensive assessment of the model performance, we created a range of data situations with varying sample sizes, connectivity scale, signal to noise ratio and signal proportions. To assess the model's ability to accurately identify true imaging biomarkers for outcomes that have both strong and weak biological signals, we varied the amount of true signals in the data by assigning signal proportion to 0.1 and 0.3. When the signal proportion equals to 0.1(0.3), we randomly assigned 10% (30%) of the covariance parameters between connectivity and behavior to be non-zero. To ensure the positive definiteness of Σ , we assigned both the covariances between connectivity and behavior and the corresponding dimensions in the latent connectivity covariance matrix as 0.9. We randomly sampled the errors for the connectivity from a normal distribution with mean 0 and variance defined by the signal-to-noise (S/N) ratio. The errors for the behavior were sampled from the normal distribution with the mean 0 and variance 0.5.

We considered three sample sizes, $N = 500$, $N = 1,000$ and $N = 2,000$ and two conditions for the number of nodes $V = 20$ and $V = 70$, and we specified two levels of the S/N ratio, 0.5 and 1 controlled by the error variance while keeping the variance of the latent variables constant. The individual-specific intercepts for connectivity and behavior were set as 0. In total, we considered 24 different scenarios combining from different signal proportions, sample sizes, node numbers and S/Ns. Under each scenario, we simulated the 100 data.

References

- Achard, S., Salvador, R., Whitcher, B., Suckling, J., & Bullmore, E. (2006). A resilient, low-frequency, small-world human brain functional network with highly connected association cortical hubs. *Journal of Neuroscience*, 26(1), 63–72.
- Aine, C., Bockholt, H. J., Bustillo, J. R., Cañive, J. M., Caprihan, A., Gasparovic, C., Hanlon, F. M., Houck, J. M., Jung, R. E., Lauriello, J., et al. (2017). Multimodal neuroimaging in schizophrenia: Description and dissemination. *Neuroinformatics*, 15, 343–364.
- Airoldi, E. M., Blei, D. M., Fienberg, S. E., & Xing, E. P. (2008). Mixed membership stochastic blockmodels. *Journal of Machine Learning Research*, 9(Sep), 1981–2014.
- Albawi, S., Mohammed, T. A., & Al-Zawi, S. (2017). Understanding of a convolutional neural network. *2017 international conference on engineering and technology (ICET)*, 1–6.
- Amaral, L. A. N., Scala, A., Barthélemy, M., & Stanley, H. E. (2000). Classes of small-world networks. *Proceedings of the national academy of sciences*, 97(21), 11149–11152.
- Barabási, A.-L. (2013). Network science. *Philosophical Transactions of the Royal Society A: Mathematical, Physical and Engineering Sciences*, 371(1987), 20120375.
- Barnard, J., McCulloch, R., & Meng, X.-L. (2000). Modeling covariance matrices in terms of standard deviations and correlations, with application to shrinkage. *Statistica Sinica*, 1281–1311.
- Bassett, D. S., & Sporns, O. (2017). Network neuroscience. *Nature neuroscience*, 20(3), 353–364.
- Bassett, D. S., & Bullmore, E. (2006). Small-world brain networks. *The neuroscientist*, 12(6), 512–523.
- Belgiu, M., & Drăguț, L. (2016). Random forest in remote sensing: A review of applications and future directions. *ISPRS journal of photogrammetry and remote sensing*, 114, 24–31.
- Benes, F. M. (1989). Myelination of cortical-hippocampal relays during late adolescence. *Schizophrenia bulletin*, 15(4), 585–593.
- Bookheimer, S. Y., Salat, D. H., Terpstra, M., Ances, B. M., Barch, D. M., Buckner, R. L., Burgess, G. C., Curtiss, S. W., Diaz-Santos, M., Elam, J. S., et al. (2019). The lifespan human connectome project in aging: An overview. *Neuroimage*, 185, 335–348.
- Breiman, L. (2001). Statistical modeling: The two cultures (with comments and a rejoinder by the author). *Statistical science*, 16(3), 199–231.
- Bullmore, E., & Sporns, O. (2009). Complex brain networks: Graph theoretical analysis of structural and functional systems. *Nature reviews neuroscience*, 10(3), 186–198.
- Calhoun, V. D., & Sui, J. (2016). Multimodal fusion of brain imaging data: A key to finding the missing link (s) in complex mental illness. *Biological psychiatry: cognitive neuroscience and neuroimaging*, 1(3), 230–244.
- Cammoun, L., Gigandet, X., Meskaldji, D., Thiran, J. P., Sporns, O., Do, K. Q., Maeder, P., Meuli, R., & Hagmann, P. (2012). Mapping the human connectome at multiple scales with diffusion spectrum mri. *Journal of Neuroscience Methods*, 203(2), 386–397. <https://doi.org/10.1016/j.jneumeth.2011.09.031>
- Casey, B. J., Cannonier, T., Conley, M. I., Cohen, A. O., Barch, D. M., Heitzeg, M. M., Soules, M. E., Teslovich, T., Dellarco, D. V., Garavan, H., et al. (2018). The adolescent brain cognitive development (ab cd) study: Imaging acquisition across 21 sites. *Developmental cognitive neuroscience*, 32, 43–54.
- Chahal, R., Gotlib, I. H., & Guyer, A. E. (2020). Research review: Brain network connectivity and the heterogeneity of depression in adolescence—a precision mental health perspective. *Journal of Child Psychology and Psychiatry*, 61(12), 1282–1298.
- Chen, S., Zhang, Y., Wu, Q., Bi, C., Kochunov, P., & Hong, L. E. (2023). Identifying covariate-related subnetworks for whole-brain connectome analysis. *Biostatistics*, kxad007.
- Cox, R. W. (1996). Afni: Software for analysis and visualization of functional magnetic resonance neuroimages. *Computers and Biomedical research*, 29(3), 162–173.

- D'Angelo, S., Murphy, T. B., & Alfò, M. (2018). Latent space modelling of multidimensional networks with application to the exchange of votes in eurovision song contest. *The Annals of Applied Statistics*. <https://api.semanticscholar.org/CorpusID:88523060>
- Derogatis, L. R. (1975). Brief symptom inventory. *European Journal of Psychological Assessment*.
- Desikan, R. S., Ségonne, F., Fischl, B., Quinn, B. T., Dickerson, B. C., Blacker, D., Buckner, R. L., Dale, A. M., Maguire, R. P., Hyman, B. T., et al. (2006). An automated labeling system for subdividing the human cerebral cortex on mri scans into gyral based regions of interest. *Neuroimage*, *31*(3), 968–980.
- Desmarais, B. A., & Cranmer, S. J. (2017). *Statistical inference in political networks research*. Oxford University Press Oxford.
- Dobra, A., Hans, C., Jones, B., Nevins, J. R., Yao, G., & West, M. (2004). Sparse graphical models for exploring gene expression data. *Journal of Multivariate Analysis*, *90*(1), 196–212.
- Dobra, A., Lenkoski, A., & Rodriguez, A. (2011). Bayesian inference for general gaussian graphical models with application to multivariate lattice data. *Journal of the American Statistical Association*, *106*(496), 1418–1433.
- Donohue, M. C., Sperling, R. A., Salmon, D. P., Rentz, D. M., Raman, R., Thomas, R. G., Weiner, M., Aisen, P. S., et al. (2014). The preclinical alzheimer cognitive composite: Measuring amyloid-related decline. *JAMA neurology*, *71*(8), 961–970.
- Dubitzky, W., Wolkenhauer, O., Cho, K.-H., & Yokota, H. (2013). *Encyclopedia of systems biology* (Vol. 402). Springer New York, NY, USA:
- Duffy, F. H., & Als, H. (2012). A stable pattern of eeg spectral coherence distinguishes children with autism from neuro-typical controls-a large case control study. *BMC medicine*, *10*, 1–19.
- Edwards, D. (2000). *Introduction to graphical modelling*. Springer Science & Business Media.
- Falbel, D., & Luraschi, J. (2023). *Torch: Tensors and neural networks with 'gpu' acceleration* [R package version 0.12.0]. <https://CRAN.R-project.org/package=torch>
- Farias, S. T., Mungas, D., Reed, B. R., Cahn-Weiner, D., Jagust, W., Baynes, K., & DeCarli, C. (2008). The measurement of everyday cognition (ecog): Scale development and psychometric properties. *Neuropsychology*, *22*(4), 531.
- Fatta, D. D., Caputo, F., Evangelista, F., & Dominici, G. (2016). Small world theory and the world wide web: Linking small world properties and website centrality. *International Journal of Markets and Business Systems*, *2*(2), 126–140.
- Finn, E. S., Shen, X., Scheinost, D., Rosenberg, M. D., Huang, J., Chun, M. M., Papademetris, X., & Constable, R. T. (2015). Functional connectome fingerprinting: Identifying individuals using patterns of brain connectivity. *Nature neuroscience*, *18*(11), 1664–1671.
- Fischer, G. H., & Molenaar, I. W. (2012). Rasch models: Foundations, recent developments, and applications.
- Friedman, J., Hastie, T., & Tibshirani, R. (2008). Sparse inverse covariance estimation with the graphical lasso. *Biostatistics*, *9*(3), 432–441.
- Friedman, N. (2004). Inferring cellular networks using probabilistic graphical models. *Science*, *303*(5659), 799–805.
- Friedman, N., Linial, M., Nachman, I., & Pe'er, D. (2000). Using bayesian networks to analyze expression data. *Proceedings of the fourth annual international conference on Computational molecular biology*, 127–135.
- Friel, N., Rastelli, R., Wyse, J., & Raftery, A. E. (2016). Interlocking directorates in irish companies using a latent space model for bipartite networks. *Proceedings of the National Academy of Sciences*, *113*(24), 6629–6634.
- Gao, S., Greene, A. S., Constable, R. T., & Scheinost, D. (2019). Combining multiple connectomes improves predictive modeling of phenotypic measures. *Neuroimage*, *201*, 116038.

- Geiger, D., & Heckerman, D. (2002). Parameter priors for directed acyclic graphical models and the characterization of several probability distributions. *The Annals of Statistics*, 30(5), 1412–1440.
- Goldenberg, A., Zheng, A. X., Fienberg, S. E., & Airoldi, E. M. (2010). A survey of statistical network models.
- Gollini, I., & Murphy, T. B. (2016). Joint modeling of multiple network views. *Journal of Computational and Graphical Statistics*, 25(1), 246–265.
- Gorgolewski, K., Burns, C. D., Madison, C., Clark, D., Halchenko, Y. O., Waskom, M. L., & Ghosh, S. S. (2011). Nipype: A flexible, lightweight and extensible neuroimaging data processing framework in python. *Frontiers in neuroinformatics*, 5, 13.
- Greene, A. S., Gao, S., Scheinost, D., & Constable, R. T. (2018). Task-induced brain state manipulation improves prediction of individual traits. *Nature communications*, 9(1), 2807.
- Greene, A. S., Shen, X., Noble, S., Horien, C., Hahn, C. A., Arora, J., Tokoglu, F., Spann, M. N., Carrión, C. I., Barron, D. S., et al. (2022). Brain–phenotype models fail for individuals who defy sample stereotypes. *Nature*, 609(7925), 109–118.
- Greve, D. N., Salat, D. H., Bowen, S. L., Izquierdo-Garcia, D., Schultz, A. P., Catana, C., Becker, J. A., Svarer, C., Knudsen, G. M., Sperling, R. A., et al. (2016). Different partial volume correction methods lead to different conclusions: An 18f-fdg-pet study of aging. *Neuroimage*, 132, 334–343.
- Greve, D. N., Svarer, C., Fisher, P. M., Feng, L., Hansen, A. E., Baare, W., Rosen, B., Fischl, B., & Knudsen, G. M. (2014). Cortical surface-based analysis reduces bias and variance in kinetic modeling of brain pet data. *Neuroimage*, 92, 225–236.
- Guha, S., & Rodriguez, A. (2021). Bayesian regression with undirected network predictors with an application to brain connectome data. *Journal of the American Statistical Association*, 116(534), 581–593.
- Guimerà, R., Diaz-Guilera, A., Vega-Redondo, F., Cabrales, A., & Arenas, A. (2002). Optimal network topologies for local search with congestion. *Physical review letters*, 89(24), 248701.
- Harary, F. (1953). On the notion of balance of a signed graph. *Michigan Mathematical Journal*, 2(2), 143–146.
- Hearst, M. A., Dumais, S. T., Osuna, E., Platt, J., & Scholkopf, B. (1998). Support vector machines. *IEEE Intelligent Systems and their applications*, 13(4), 18–28.
- Heider, F. (1944). Social perception and phenomenal causality. *Psychological review*, 51(6), 358.
- Hoff, P. (2005a). Bilinear mixed-effects models for dyadic data. *J. Amer. Statist. Assoc.*, 100(469), 286–295.
- Hoff, P. (2008a). Modeling homophily and stochastic equivalence in symmetric relational data. In J. Platt, D. Koller, Y. Singer, & S. Roweis (Eds.), *Advances in neural information processing systems 20* (pp. 657–664). MIT Press.
- Hoff, P. (2008b). Modeling homophily and stochastic equivalence in symmetric relational data. *Advances in neural information processing systems*, 657–664.
- Hoff, P. D. (2005b). Bilinear mixed-effects models for dyadic data. *Journal of the American Statistical Association*, 100(469), 286–295.
- Hoff, P. D., Raftery, A. E., & Handcock, M. S. (2002). Latent space approaches to social network analysis. *Journal of the American Statistical Association*, 97(460), 1090–1098.
- Holland, P. W., Laskey, K. B., & Leinhardt, S. (1983). Stochastic blockmodels: First steps. *Social networks*, 5(2), 109–137.
- Horien, C., Shen, X., Scheinost, D., & Constable, R. T. (2019). The individual functional connectome is unique and stable over months to years. *Neuroimage*, 189, 676–687.
- Jack Jr, C. R., Bernstein, M. A., Fox, N. C., Thompson, P., Alexander, G., Harvey, D., Borowski, B., Britson, P. J., L. Whitwell, J., Ward, C., et al. (2008). The alzheimer’s disease neuroimaging initiative (adni): Mri methods. *Journal of Magnetic Resonance Imaging: An Official Journal of the International Society for Magnetic Resonance in Medicine*, 27(4), 685–691.

- Jenkinson, M., Bannister, P., Brady, M., & Smith, S. (2002). Improved optimization for the robust and accurate linear registration and motion correction of brain images. *Neuroimage*, *17*(2), 825–841.
- Jenkinson, M., Beckmann, C. F., Behrens, T. E., Woolrich, M. W., & Smith, S. M. (2012). Fsl. *NeuroImage*, *62*(2), 782–790. <https://doi.org/10.1016/j.neuroimage.2011.09.015>
- Jenkinson, M., & Smith, S. (2001). A global optimisation method for robust affine registration of brain images. *Medical image analysis*, *5*(2), 143–156.
- Johnston, M. V. (2004). Clinical disorders of brain plasticity. *Brain and Development*, *26*(2), 73–80.
- Joshi, A., Scheinost, D., Okuda, H., Belhachemi, D., Murphy, I., Staib, L. H., & Papademetris, X. (2011). Unified framework for development, deployment and robust testing of neuroimaging algorithms. *Neuroinformatics*, *9*, 69–84.
- Kim, B., Lee, K. H., Xue, L., & Niu, X. (2018). A review of dynamic network models with latent variables. *Statistics surveys*, *12*, 105.
- Kolb, B. (2013). *Brain plasticity and behavior*. Psychology Press.
- Kolb, B., & Wishaw, I. Q. (1998). Brain plasticity and behavior. *Annual review of psychology*, *49*(1), 43–64.
- Krasnegor, N. A., Lyon, G., & Goldman-Rakic, P. S. (1997). *Development of the prefrontal cortex: Evolution, neurobiology, and behavior*. Paul H Brookes Publishing.
- Krueger, R. F., & Markon, K. E. (2006). Understanding psychopathology: Melding behavior genetics, personality, and quantitative psychology to develop an empirically based model. *Current directions in psychological science*, *15*(3), 113–117.
- Kueper, J. K., Speechley, M., & Montero-Odasso, M. (2018). The alzheimer’s disease assessment scale–cognitive subscale (adas-cog): Modifications and responsiveness in pre-dementia populations. a narrative review. *Journal of Alzheimer’s Disease*, *63*(2), 423–444.
- Lauritzen, S. L. (1996). *Graphical models* (Vol. 17). Clarendon Press.
- Leonard, T., & Hsu, J. S. (1992). Bayesian inference for a covariance matrix. *The Annals of Statistics*, *20*(4), 1669–1696.
- Lutkenhoff, E. S., Rosenberg, M., Chiang, J., Zhang, K., Pickard, J. D., Owen, A. M., & Monti, M. M. (2014). Optimized brain extraction for pathological brains (optibet). *PloS one*, *9*(12), e115551.
- MacDonald, P. W., Levina, E., & Zhu, J. (2022). Latent space models for multiplex networks with shared structure. *Biometrika*, *109*(3), 683–706.
- Manjón, J. V., Coupé, P., Concha, L., Buades, A., Collins, D. L., & Robles, M. (2013). Diffusion weighted image denoising using overcomplete local pca. *PLoS ONE*, *8*(9). <https://doi.org/10.1371/journal.pone.0073021>
- Marek, S., Tervo-Clemmens, B., Nielsen, A. N., Wheelock, M. D., Miller, R. L., Laumann, T. O., Earl, E., Foran, W. W., Cordova, M., Doyle, O., et al. (2019). Identifying reproducible individual differences in childhood functional brain networks: An abcd study. *Developmental cognitive neuroscience*, *40*, 100706.
- Matias, C., & Robin, S. (2014). Modeling heterogeneity in random graphs through latent space models: A selective review. *ESAIM: Proceedings and Surveys*, *47*, 55–74.
- McPherson, M., Smith-Lovin, L., & Cook, J. M. (2001). Birds of a feather: Homophily in social networks. *Annual review of sociology*, *27*(1), 415–444.
- Meyer, D., Dimitriadou, E., Hornik, K., Weingessel, A., & Leisch, F. (2023). *E1071: Misc functions of the department of statistics, probability theory group (formerly: E1071), tu wien* [R package version 1.7-14]. <https://CRAN.R-project.org/package=e1071>
- Mihalik, A., Brudfors, M., Robu, M., Ferreira, F. S., Lin, H., Rau, A., Wu, T., Blumberg, S. B., Kanber, B., Tariq, M., et al. (2019). Abcd neurocognitive prediction challenge 2019: Predicting individual fluid intelligence scores from structural mri using probabilistic segmentation and kernel ridge regression. In *Adolescent brain cognitive development neurocognitive prediction: First challenge, abcd-np 2019, held in conjunction with miccai 2019, shenzhen, china, october 13, 2019, proceedings* (pp. 133–142). Springer.

- Minhas, S., Hoff, P. D., & Ward, M. D. (2019). Inferential approaches for network analysis: Amen for latent factor models. *Political Analysis*, 27(2), 208–222.
- Moore, C., & Sciacca, F. (2019). Fiber assignment by continuous tracking algorithm (fact). *Radiopaedia.org*. <https://doi.org/10.53347/rid-72014>
- Morris, E. L., He, K., & Kang, J. (2022). Scalar on network regression via boosting. *The annals of applied statistics*, 16(4), 2755.
- Murphy, K. (2001). An introduction to graphical models. *Rap. tech*, 96, 1–19.
- Newman, M. E. (2003). The structure and function of complex networks. *SIAM review*, 45(2), 167–256.
- Ni, Y., Baladandayuthapani, V., Vannucci, M., & Stingo, F. C. (2022). Bayesian graphical models for modern biological applications. *Statistical Methods & Applications*, 31(2), 197–225.
- Ni, Y., Stingo, F. C., & Baladandayuthapani, V. (2019). Bayesian graphical regression. *Journal of the American Statistical Association*, 114(525), 184–197.
- NIH. (2022). Nih toolbox scoring and interpretation guide [Accessed on February 16, 2024]. https://www.nihtoolbox.org/app/uploads/2022/05/Toolbox_Scoring_and_Interpretation_Guide_for_iPad.v1.7-5.25.21.pdf
- Norman, R. Z., et al. (1965). Structural models: An introduction to the theory of directed graphs.
- Papp, K. V., Rentz, D. M., Maruff, P., Sun, C.-K., Raman, R., Donohue, M. C., Schembri, A., Stark, C., Yassa, M. A., Wessels, A., et al. (2021). The computerized cognitive composite (c3) in a4, an alzheimer’s disease secondary prevention trial. *The journal of prevention of Alzheimer’s disease*, 8, 59–67.
- Paquette, M., Girard, G., Chamberland, M., & Descoteaux, M. (2016). ” noise” in diffusion tractography connectomes is not additive.
- Paus, T., Zijdenbos, A., Worsley, K., Collins, D. L., Blumenthal, J., Giedd, J. N., Rapoport, J. L., & Evans, A. C. (1999). Structural maturation of neural pathways in children and adolescents: In vivo study. *Science*, 283(5409), 1908–1911.
- Petersen, R. C., Aisen, P. S., Beckett, L. A., Donohue, M. C., Gamst, A. C., Harvey, D. J., Jack Jr, C., Jagust, W. J., Shaw, L. M., Toga, A. W., et al. (2010). Alzheimer’s disease neuroimaging initiative (adni) clinical characterization. *Neurology*, 74(3), 201–209.
- Pornpattananangkul, N., Grogans, S., Yu, R., & Nusslock, R. (2019). Single-trial eeg dissociates motivation and conflict processes during decision-making under risk. *Neuroimage*, 188, 483–501.
- Reli3n, J. D. A., Kessler, D., Levina, E., & Taylor, S. F. (2019). Network classification with applications to brain connectomics. *The annals of applied statistics*, 13(3), 1648.
- Robins, G., Pattison, P., Kalish, Y., & Lusher, D. (2007). An introduction to exponential random graph (p*) models for social networks. *Social networks*, 29(2), 173–191.
- Rosenblatt, M., Rodriguez, R. X., Westwater, M. L., Dai, W., Horien, C., Greene, A. S., Constable, R. T., Noble, S., & Scheinost, D. (2023). Connectome-based machine learning models are vulnerable to subtle data manipulations. *Patterns*, 4(7).
- Salter-Townshend, M., & McCormick, T. H. (2017). Latent space models for multiview network data. *The annals of applied statistics*, 11(3), 1217.
- Satterthwaite, T. D., Kable, J. W., Vandekar, L., Katchmar, N., Bassett, D. S., Baldassano, C. F., Ruparel, K., Elliott, M. A., Sheline, Y. I., Gur, R. C., et al. (2015). Common and dissociable dysfunction of the reward system in bipolar and unipolar depression. *Neuropsychopharmacology*, 40(9), 2258–2268.
- Sawai, H. (2012). Exploring a new small-world network for real-world applications. *Networked Digital Technologies: 4th International Conference, NDT 2012, Dubai, UAE, April 24-26, 2012. Proceedings, Part I* 4, 90–101.
- Sawai, H. (2014). Hierarchical construction of multi-star small-world networks for real-world applications. *Artif. Intell. Res.*, 3(4), 1–14.

- Shen, X., Finn, E. S., Scheinost, D., Rosenberg, M. D., Chun, M. M., Papademetris, X., & Constable, R. T. (2017). Using connectome-based predictive modeling to predict individual behavior from brain connectivity. *nature protocols*, *12*(3), 506–518.
- Shen, X., Tokoglu, F., Papademetris, X., & Constable, R. T. (2013). Groupwise whole-brain parcellation from resting-state fmri data for network node identification. *Neuroimage*, *82*, 403–415.
- Shi, R., & Guo, Y. (2016). Investigating differences in brain functional networks using hierarchical covariate-adjusted independent component analysis. *The annals of applied statistics*, *10*(4), 1930.
- Shmueli, G. (2010). To Explain or to Predict? *Statistical Science*, *25*(3), 289–310. <https://doi.org/10.1214/10-STS330>
- Shmueli, G., & Koppius, O. R. (2011). Predictive analytics in information systems research. *MIS quarterly*, *553–572*.
- Simpson, S. L., Shappell, H. M., & Bahrami, M. (2024). Statistical brain network analysis. *Annual Review of Statistics and Its Application*, *11*.
- Sinke, M. R., Dijkhuizen, R. M., Caimo, A., Stam, C. J., & Otte, W. M. (2016). Bayesian exponential random graph modeling of whole-brain structural networks across lifespan. *NeuroImage*, *135*, 79–91.
- Smith, A. L., Asta, D. M., & Calder, C. A. (2019). The geometry of continuous latent space models for network data. *Statistical science: a review journal of the Institute of Mathematical Statistics*, *34*(3), 428.
- Song, M., Liu, Y., Zhou, Y., Wang, K., Yu, C., & Jiang, T. (2009). Default network and intelligence difference. *IEEE Transactions on autonomous mental development*, *1*(2), 101–109.
- Sosa, J., & Buitrago, L. (2021). A review of latent space models for social networks. *Revista Colombiana de Estadística*, *44*(1), 171–200.
- Sperling, R. A., Rentz, D. M., Johnson, K. A., Karlawish, J., Donohue, M., Salmon, D. P., & Aisen, P. (2014). The a4 study: Stopping ad before symptoms begin? *Science translational medicine*, *6*(228), 228fs13–228fs13.
- Stavropoulos, I., Pervanidou, P., Gnardellis, C., Loli, N., Theodorou, V., Mantzou, A., Soukou, F., Sinani, O., & Chrousos, G. P. (2017). Increased hair cortisol and antecedent somatic complaints in children with a first epileptic seizure. *Epilepsy & Behavior*, *68*, 146–152.
- Steinley, D. (2021). Recent advances in (graphical) network models [PMID: 34029161]. *Multivariate Behavioral Research*, *56*(2), 171–174. <https://doi.org/10.1080/00273171.2021.1911777>
- Sweet, T. (2016). Social network methods for the educational and psychological sciences. *Educational Psychologist*, *51*(3-4), 381–394.
- Tibshirani, R. (1996). Regression shrinkage and selection via the lasso. *Journal of the Royal Statistical Society Series B: Statistical Methodology*, *58*(1), 267–288.
- Tourbier, S., Rue Queralt, J., Glomb, K., Aleman-Gomez, Y., Mullier, E., Griffa, A., Schöttner, M., Wirsich, J., Tuncel, A., Jancovic, J., Bach Cuadra, M., & Hagmann, P. (2022). Connectome mapper 3: A flexible and open-source pipeline software for multiscale multimodal human connectome mapping. *Journal of Open Source Software*, *7*(74), 4248. <https://doi.org/10.21105/joss.04248>
- Vitale, M. P., Giordano, G., & Ragozini, G. (2022). Discussion to: Bayesian graphical models for modern biological applications by Y. Ni, V. Baladandayuthapani, M. Vannucci and F.C. Stingo. *Statistical Methods & Applications*, *31*(2), 269–278. <https://doi.org/10.1007/s10260-021-00603->
- Wang, L., Durante, D., Jung, R. E., & Dunson, D. B. (2017). Bayesian network–response regression. *Bioinformatics*, *33*(12), 1859–1866.
- Wang, L., Lin, F. V., Cole, M., & Zhang, Z. (2021). Learning clique subgraphs in structural brain network classification with application to crystallized cognition. *NeuroImage*, *225*, 117493.
- Wang, S. (2021). Recent integrations of latent variable network modeling with psychometric models. *Frontiers in psychology*, *12*, 773289.

- Wang, S., & Edgerton, J. (2022). Resilience to stress in bipartite networks: Application to the islamic state recruitment network. *Journal of Complex Networks*, *10*(4), cnac017.
- Wasserman, S., & Faust, K. (1994). Social network analysis: Methods and applications.
- Wasserman, S., & Pattison, P. (1996). Logit models and logistic regressions for social networks: I. an introduction to markov graphs andp. *Psychometrika*, *61*(3), 401–425.
- Weintraub, S., Dikmen, S. S., Heaton, R. K., Tulsky, D. S., Zelazo, P. D., Bauer, P. J., Carlozzi, N. E., Slotkin, J., Blitz, D., Wallner-Allen, K., et al. (2013). Cognition assessment using the nih toolbox. *Neurology*, *80*(11 Supplement 3), S54–S64.
- Wohlgemuth, J. (2012). *Small world properties of facebook group networks*. University of Nebraska at Omaha.
- Wright, M. N., & Ziegler, A. (2017). ranger: A fast implementation of random forests for high dimensional data in C++ and R. *Journal of Statistical Software*, *77*(1), 1–17. <https://doi.org/10.18637/jss.v077.i01>
- Xia, C. H., Ma, Z., Cui, Z., Bzdok, D., Thirion, B., Bassett, D. S., Satterthwaite, T. D., Shinohara, R. T., & Witten, D. M. (2020). *Multi-scale network regression for brain-phenotype associations* (tech. rep.). Wiley Online Library.
- Xu, F., Garai, S., Duong-Tran, D., Saykin, A. J., Zhao, Y., & Shen, L. (2022). Consistency of graph theoretical measurements of alzheimer’s disease fiber density connectomes across multiple parcellation scales. *(BIBM) 2022 IEEE International Conference on Bioinformatics and Biomedicine IEEE, Regular Paper*.
- Yan, J., Raja, V. V., Huang, Z., Amico, E., Nho, K., Fang, S., Sporns, O., Wu, Y. C., Saykin, A., Goñi, J., & Shen, L. (2020). Brain-wide structural connectivity alterations under the control of alzheimer risk genes. *International Journal of Computational Biology and Drug Design*, *13*(1), 58. <https://doi.org/10.1504/ijcbdd.2020.105098>
- Zhang, Z., Allen, G. I., Zhu, H., & Dunson, D. (2019). Tensor network factorizations: Relationships between brain structural connectomes and traits. *Neuroimage*, *197*, 330–343.
- Zhang, Z. (2021). A note on wishart and inverse wishart priors for covariance matrix. *Journal of Behavioral Data Science*, *1*(2), 119–126.
- Zhao, Y., Li, T., & Zhu, H. (2022). Bayesian sparse heritability analysis with high-dimensional neuroimaging phenotypes. *Biostatistics*, *23*(2), 467–484.
- Zhou, Y., & Müller, H.-G. (2022). Network regression with graph laplacians. *The Journal of Machine Learning Research*, *23*(1), 14383–14423.
- Zuo, X.-N., Ehmke, R., Mennes, M., Imperati, D., Castellanos, F. X., Sporns, O., & Milham, M. P. (2012). Network centrality in the human functional connectome. *Cerebral cortex*, *22*(8), 1862–1875.

Authors' Response to Reviewers' Comments: “Neuroimaging connectivity analysis needs network science for brain-behavior linking” (NMETH-A54306)

Dear Editor,

Thank you for giving us the opportunity to further revise our manuscript. We have responded to all points from the new round of reviews below. In the sections below, we present the verbatim text of the reviews in the order of the review. Following each review comment, we discuss how we improved the manuscript to address the point raised by the reviewer in blue. We thank the reviewers for their thoughtful and productive suggestions and believe that the manuscript is substantially improved as a result. In addition to this letter, we include a changed version of the edited manuscript with changes highlighted in blue italics for ease of spotting them.

Best wishes,

Authors

Editor

Thank you for your patience. Your Article, “Neuroimaging connectivity analysis needs network science for brain-behavior linking”, has now been seen by three reviewers (the two original ones and a new one). As you will see from their comments below, although the reviewers find your work of considerable potential interest, they continue to raise a number of concerns. We are interested in the possibility of publishing your paper in Nature Methods, but would like to consider your response to these concerns before we reach a final decision on publication.

We therefore invite you to revise your manuscript to address these concerns. Specifically, please do work on the clarity of the presentation and highlight the added datasets. Please also discuss the overfitting issue brought up by reviewer #1. Finally, I would like to mention that clinical relevance is not our focus at Nature Methods, and we therefore ask not to focus on that.

We really appreciate the Editor for the comments and for taking the time to review our work. We have made changes according to your suggestions and our response to each comment is as follows.

1. In response to the comment “*please do work on the clarity of the presentation and highlight the added datasets.*”, we have improved the clarity of the presentation and highlight the added datasets by clarifying the focus of the paper and highlighting the added datasets in the abstract, introduction, discussion and throughout the paper (see red highlighted texts). In addition, we have provided additional investigations about the robustness of our model across the added datasets in the results section.

To better highlight the added datasets, we have added relevant descriptions in the **Abstract, Introduction Section, Results Section** and **Discussion Section** that read,

In the Abstract:

LatentSNA is broadly applicable to multiple neuroimaging landmark studies, imaging modalities, outcome measures with developing, aging and transdiagnostic populations, totaling 8,003 to 11,861 participants. In these applications, LatentSNA achieves substantial accuracy gains (averaging 110% - 150%) and replicability improvements (averaging 200%) in moderate-to-large datasets. As a result, LatentSNA provides an unprecedented view of how network topology is implicated in brain dysfunction.

In the Introduction Section:

LatentSNA, a novel inference-focused generative Bayesian framework capturing universal network topologies and leveraging state-of-the-art latent space estimation techniques, is particularly designed to analyze human connectomes and identify meaningful neuroimaging biomarkers of individual outcomes (Figure 3). It comprises an integrated workflow containing three modules: networked connectome modeling (preserving transitivity and modularity), psychometric behavior profiling, and two-way brain-behavior linking. We achieve robust neuroimaging biomarker detection with markedly improved statistical power, demonstrating the method’s generalizability across seven neuroimaging landmark studies: Alzheimer’s Disease Neuroimaging Initiative Grand Opportunities and ADNI Phase 2 (ADNI-GO/2) and ADNI Phase 3 (ADNI-3, Jack Jr et al., 2008; Petersen et al., 2010), Anti-Amyloid Treatment in Asymptomatic Alzheimer’s Disease (A4) (Sperling et al., 2014), Human Connectome Project Aging (HCP-A) (Bookheimer et al., 2019), Adolescent Brain Cognitive Development Study Baseline (ABCD-B) and second release (ABCD-2) (Casey et al., 2018), and transdiagnostic data collected at Yale (Greene et al., 2022). These studies involve eight different imaging modalities and 20 outcome measures with a total of 8,003 to 11,861 participants. LatentSNA consistently improves model fit performance over nine existing methods, including three state-of-the-art deep learning techniques (SVM, RF, and CNN), two network-based brain analysis methods (GC¹ and TNFA²), and four popular brain-behavior linking approaches (CPM, rCPM, LASSSO, and CCA³). It enhances the predictability (an average of 110% improvement over TNFA and an average of 150% improvement over CPM) and replicability (averaging 200% improvement over CPM) of various imaging techniques, including fMRI, T1-weighted structural MRI, DTI, and PET. Moreover, it is generalizable to different outcome measures, including but not limited to cognition, emotion, assessment of mental disorders, focal tau PET (¹⁸F]flortaucipir) SUVR metrics, and different participant demographics across developing, aging, and transdiagnostic populations.

In the Results Section:

The lack of robustness and replicability of current fMRI studies is a well-known challenge (see Wu et al., 2023; Zuo et al., 2019). We investigate the robustness and replicability performance of our proposed method by comparing covariance effect estimation across random samplings of the test data, i.e., replicability with the same data. We calculated the absolute correlation of the estimated effects between replications and report the results in Figure 1. The results show that our model estimated effects (covariance/correlations between brain and behaviors) are consistent across independent replications—when we randomly split the data into 90% training and 10% test samples. The CPM, on the other hand, shows lower replicability and robustness. LatentSNA shows consistently higher replicability (with above 0.75 correlations) across datasets,

¹Penalized Graph Classification (Reli3n et al., 2019)

²Tensor Network Factorization Analysis (Zhang et al., 2019)

³Canonical Correlation Analysis

while CPM shows substantially lower and more variable reproducibility (correlations ranging from 0.25 to 0.75).

In the Discussion Section:

In this study, we developed a network-science driven analytic method that addresses the lack of power and inflated Type II errors in neuroimaging biomarker detection. Collectively, we have fitted the method to 7 different datasets (ADNI-GO/2, ADNI-3, A4, HCP-A, ABCD-B, ABCD-2 and Transdiagnostic) involving 8 different imaging modalities and 20 outcome measures with a total of 8,003 - 11,861 subjects' information included. The results demonstrate that the proposed method is broadly applicable to a variety of imaging techniques including functional MRI (fMRI), T1-weighted structural MRI, Diffusion Tensor Imaging (DTI) and positron emission tomography (PET); and it is generalizable to different types of outcome measures, which include but are not limited to cognition, emotion, assessment of mental disorders and focal tau PET SUVR metrics. The proposed method shows consistent improvement in the model fit and model replicability across data situations against available alternatives for a variety of outcomes using functional and structural imaging modalities in developing, aging and transdiagnostic populations.

2. In response to the comment “Please also discuss the overfitting issue brought up by reviewer #1.”, we would like to confirm that our method does not overfit. All of our model fit and prediction accuracy assessments are based on randomly sampled test data with no overlap with the training data. By definition, overfitting refers to the model being overly close to the training data with high prediction accuracy on the training data, and low prediction accuracy on the test data. Our model fit indices are calculated based on the test data; we show high prediction accuracy on the test data, which suggests that our model is not overfitted.

In addition, we assessed the replicability and robustness of our model across 10 random replications. We report the robustness assessment by comparing the consistency of the model-estimated effects across the 10 replication in Figure 1. The results show that our model is robust and replicable across replications, a strong evidence against overfitting. Overfitted models are not replicable nor robust because overfitted models only capture information in the training data with poor generalizability to the test data. By showing that our model results are robust and replicable across random replications, we provide strong evidence against overfitting.

3. In response to the comment “Finally, I would like to mention that clinical relevance is not our focus at Nature Methods, and we therefore ask not to focus on that.”, we have followed the editor’s suggestion.

Reviewer 1

We really appreciate Reviewer 1 for the comments and for taking the time to review our work. We have made changes according to your suggestions and our response to each comment is as follows.

1. Unfortunately, despite the authors’ efforts, the paper remains unfocused. Although the authors have shown some applications of LatentSNA to multiple datasets in predicting outcome measures, a large portion of the paper is similar to the previous version. About half of the paper is dedicated to validating LatentSNA and the other half is dedicated to predicting internalizing psychopathology and connectivity based on the ABCD data.

Response: In response to the comment, “*Unfortunately, despite the authors’ efforts, the paper remains unfocused.*”, we would like to detail our efforts to clarify the focus of the paper, which is to propose a novel method for identifying significant neuroimaging biomarkers of individual outcomes. To clarify the focus of the paper, our rewritten manuscript emphasizes our method’s contribution to current brain-behavior modeling literature and details its advantageous prediction accuracy and replicability capabilities, generalizable across datasets. We have improved the clarity of the presentation and highlight the added datasets by clarifying the focus of the paper and highlighting the added datasets in the abstract, introduction, discussion and throughout the paper (see red highlighted texts).

Specifically, in the introduction, we have emphasized the contribution of the current method to the broad landscape of neuroimaging connectivity models. Compared with univariate and marginal association analyses (e.g., Marek et al., 2019; Satterthwaite et al., 2015) and CPMs⁴ (Finn et al., 2015; Shen et al., 2017), LatentSNA integrates the dependent organization of brain connectivity edges in the data generation process following the latent space modeling framework (e.g., Hoff, 2005a; Kim et al., 2018; Smith et al., 2019) thus preserving informative neurobiological indicators (Bassett & Sporns, 2017; Bullmore & Sporns, 2009; Zuo et al., 2012). Compared with graphical models (e.g., Dobra et al., 2004; Dobra et al., 2011; Friedman et al., 2008; Friedman, 2004), LatentSNA aims to understand the structure and property of the observed brain networks that ultimately facilitate understandings about brain’s relationship with behavior, instead of estimating the network structure. LatentSNA shows superior model performances with substantially improved out-of-sample prediction accuracy and effect estimation robustness. Both predicability and robustness of the method is generalizable across seven neuroimaging landmark studies: Alzheimer’s Disease Neuroimaging Initiative Grand Opportunities and ADNI Phase 2 (ADNI-GO/2) and ADNI Phase 3 (ADNI-3, Jack Jr et al., 2008; Petersen et al., 2010), Anti-Amyloid Treatment in Asymptomatic Alzheimer’s Disease (A4) (Sperling et al., 2014), Human Connectome Project Aging (HCP-A) (Bookheimer et al., 2019), Adolescent Brain Cognitive Development Study Baseline (ABCD-B) and second release (ABCD-2) (Casey et al., 2018), and transdiagnostic data collected at Yale (Greene et al., 2022).

In response to the comment, “*Although the authors have shown some applications of LatentSNA to multiple datasets in predicting outcome measures, a large portion of the paper is similar to the previous version. About half of the paper is dedicated to validating LatentSNA and the other half is dedicated to predicting internalizing psychopathology and connectivity based on the ABCD data.*”, we agree that our manuscript provide a number of applications using multiple datasets. The portion of the paper that was kept—that we suspect the reviewer is referring to—was the real-data application with the ABCD study. The reason behind the ABCD study application being kept in the paper instead of being rewritten into something else is to show that our method can provide interpretable brain-behavior linking results. We argue that this part of the paper is important to demonstrate how our method result can be interpreted.

Upon communication with the editor, we have been advised not to rewrite the result section. A further investigation about whether results from the ABCD study generalize across studies is outside of the scope of *Nature Methods*. Instead, we have been informed to focus on the method itself, not the results—that the proposed method is applicable, with strong model fit, across imaging modalities, outcome measures and study populations. The generalizability of ABCD study results is outside of the scope of *Nature Methods*.

To demonstrate the generalizability of our method across studies, we have focused on two aspects. First, our method fits well to a range of datasets with strong out-of-sample prediction accuracy. Second, our method can shows robust and replicable results with consistent effect estimation across random samplings within the same study. For both predictability and replicability, we have shown the generalizability of the method by demonstrating that the method works in 7 different datasets, involving 8 different imaging

⁴Connectome-based Predictive Modeling

Figure 1: Reproducibility analysis of CPM (green) and latentSNA (red) methods across different behaviors, fMRI conditions, and datasets. Each boxplot summarizes the reproducibility performance through correlations across 10 independent replications. We use the absolute correlation between the estimated effects (regression coefficient estimates in CPM and covariance estimates in LatentSNA) across replications to represent the replicability and robustness of both methods. Higher correlation values indicate better reproducibility.

modalities and 20 outcome measures, with a total of 8,003–11,861 subjects’ information included. We have shown that the proposed LatentSNA method fits well and can be used to identify imaging biomarkers with a variety of neuroimaging data and outcome measures. The satisfactory model fit is well demonstrated via 10-fold cross validation with out-of-sample model fit shown in Figure 3A in the manuscript.

The lack of robustness and replicability of current fMRI studies is a well-known challenge (see Wu et al., 2023; Zuo et al., 2019). We investigate the robustness and replicability performance of our proposed method by comparing covariance effect estimation across random samplings of the test data, i.e., replicability with the same data. We calculated the absolute correlation of the estimated effects between replications and report the results in Figure 1. The results show that our model estimated effects (covariance/correlations between brain and behaviors) are consistent across independent replications—when we randomly split the data into 90% training and 10% test samples. The CPM, on the other hand, shows lower replicability and robustness. LatentSNA shows consistently higher replicability (with above 0.75 correlations) across datasets, while CPM shows substantially lower and more variable reproducibility (correlations ranging from 0.25 to 0.75).

To improve the focus of the paper and highlight the added datasets, we have added discussions in the **Abstract**, **Introduction Section**, **Results Section** and **Discussion Section** that read,

In the **Abstract**:

LatentSNA is broadly applicable to multiple neuroimaging landmark studies, imaging modalities, outcome measures with developing, aging and transdiagnostic populations, totaling 8,003 to 11,861 participants. In these applications, LatentSNA achieves substantial accuracy gains (averaging 110% - 150%) and replicability improvements (averaging 200%) in moderate-to-large datasets. As a result, LatentSNA provides an unprecedented view of how network topology is implicated in brain dysfunction.

In the **Introduction Section**:

LatentSNA, a novel inference-focused generative Bayesian framework capturing universal network topologies and leveraging state-of-the-art latent space estimation techniques, is particu-

larly designed to analyze human connectomes and identify meaningful neuroimaging biomarkers of individual outcomes (Figure 3). It comprises an integrated workflow containing three modules: networked connectome modeling (preserving transitivity and modularity), psychometric behavior profiling, and two-way brain-behavior linking. We achieve robust neuroimaging biomarker detection with markedly improved statistical power, demonstrating the method’s generalizability across seven neuroimaging landmark studies: Alzheimer’s Disease Neuroimaging Initiative Grand Opportunities and ADNI Phase 2 (ADNI-GO/2) and ADNI Phase 3 (ADNI-3, Jack Jr et al., 2008; Petersen et al., 2010), Anti-Amyloid Treatment in Asymptomatic Alzheimer’s Disease (A4) (Sperling et al., 2014), Human Connectome Project Aging (HCP-A) (Bookheimer et al., 2019), Adolescent Brain Cognitive Development Study Baseline (ABCD-B) and second release (ABCD-2) (Casey et al., 2018), and transdiagnostic data collected at Yale (Greene et al., 2022). These studies involve eight different imaging modalities and 20 outcome measures with a total of 8,003 to 11,861 participants. LatentSNA consistently improves model fit performance over nine existing methods, including three state-of-the-art deep learning techniques (SVM, RF, and CNN), two network-based brain analysis methods (GC⁵ and TNFA⁶), and four popular brain-behavior linking approaches (CPM, rCPM, LASSSO, and CCA⁷). It enhances the predictability (an average of 110% improvement over TNFA and an average of 150% improvement over CPM) and replicability (averaging 200% improvement over CPM) of various imaging techniques, including fMRI, T1-weighted structural MRI, DTI, and PET. Moreover, it is generalizable to different outcome measures, including but not limited to cognition, emotion, assessment of mental disorders, focal tau PET (¹⁸F]flortaucipir) SUVR metrics, and different participant demographics across developing, aging, and transdiagnostic populations.

In the **Results Section**:

The lack of robustness and replicability of current fMRI studies is a well-known challenge (see Wu et al., 2023; Zuo et al., 2019). We investigate the robustness and replicability performance of our proposed method by comparing covariance effect estimation across random samplings of the test data, i.e., replicability with the same data. We calculated the absolute correlation of the estimated effects between replications and report the results in Figure 1. The results show that our model estimated effects (covariance/correlations between brain and behaviors) are consistent across independent replications—when we randomly split the data into 90% training and 10% test samples. The CPM, on the other hand, shows lower replicability and robustness. LatentSNA shows consistently higher replicability (with above 0.75 correlations) across datasets, while CPM shows substantially lower and more variable reproducibility (correlations ranging from 0.25 to 0.75).

In the **Discussion Section**:

In this study, we developed a network-science driven analytic method that addresses the lack of power and inflated Type II errors in neuroimaging biomarker detection. Collectively, we have fitted the method to 7 different datasets (ADNI-GO/2, ADNI-3, A4, HCP-A, ABCD-B, ABCD-2 and Transdiagnostic) involving 8 different imaging modalities and 20 outcome measures with a total of 8,003 - 11,861 subjects’ information included. The results demonstrate that the proposed method is broadly applicable to a variety of imaging techniques including functional MRI

⁵Penalized Graph Classification (Reli3n et al., 2019)

⁶Tensor Network Factorization Analysis (Zhang et al., 2019)

⁷Canonical Correlation Analysis

(fMRI), T1-weighted structural MRI, Diffusion Tensor Imaging (DTI) and positron emission tomography (PET); and it is generalizable to different types of outcome measures, which include but are not limited to cognition, emotion, assessment of mental disorders and focal tau PET SUVR metrics. The proposed method shows consistent improvement in the model fit and model replicability across data situations against available alternatives for a variety of outcomes using functional and structural imaging modalities in developing, aging and transdiagnostic populations.

2. The title is awkward – “connectivity analysis” has always been closely related to “network science”. So it is unclear why now “connectivity analysis needs network science”. Similar problem with the first sentence of the abstract.

Response: We greatly appreciate the reviewer for raising this point. In response, we would like to clarify that the discipline “network science” is different from “connectivity analysis”. Network science, according to foundational research done by Drs Albert-László Barabási, Mark Newman and Stanley Wasserman, etc. (Barabási, 2013; Newman, 2003; Wasserman & Faust, 1994), is defined as a complexity-driven discipline focused on the shared architecture of networks emerging across physical, biological, and social domains. The interconnectivity of entities—entities spanning physical, biological, and social domains such as social relationships, World Wide Web, power grid (or electricity network)—defines characteristics of networked structures such as vulnerability, efficiency, resilience, etc (see detailed discussions in Barabási (2013)).

Network science is often characterized by the mathematical investigations about the universal principles of network generation—what mathematical principles define the generations of network with power law degree distributions, for example. This discipline may have overlap with neuroimaging connectivity analysis although they are not the same. For example, connectome-based predictive modeling analyzes neuroimaging connectivity data (is a neuroimaging connectivity analysis method), but it does not incorporate the networked (dependent) characteristics of the brain when modeling brain connectivity edges—it assumes one connectivity edge to be an independent observation from another (see definition of independent observations in Casella and Berger (2024))—not network science.

Therefore, our title indicates that the analysis techniques for neuroimaging connectivity data need to incorporate the networked (dependent) characteristics of the brain in model building. Neuroimaging connectivity data analysis should not treat connectivity edges as independent observations, which poses as an opportunity cost leading to low statistical power as evidenced by Figure 2 of the manuscript; instead, the analysis techniques should regard the dependence among connectivity edges (the interconnectivity of the brain) as a crucial piece of network generation and modeling, and when linking brain with behavior.

To better clarify, we have added relevant **Introduction Section** description that read,

In the **Introduction**:

What differentiates a network-science-driven analytic approach is that it draws on insights regarding the universality of the communicative structures of real-world networks (Barabási, 2013). Characteristics such as the small-world property and sparsity are universal properties found in social networks (Wohlgemuth, 2012), political networks (Wang & Edgerton, 2022), the World Wide Web (Fatta et al., 2016), and human connectomes (Amaral et al., 2000; Bassett & Bullmore, 2006; Dubitzky et al., 2013). Shared network architectures, as a result of being

governed by universal principles (Barabási, 2013; Wohlgemuth, 2012), allow us to use a common set of mathematical and statistical instruments for network modeling. Network science is characterized by the mathematical investigations about the universal principles of network generation—what mathematical principles define the generations of network with power law degree distributions, for example (Barabási, 2013; Newman, 2003; Wasserman & Faust, 1994). This discipline may have overlap with neuroimaging connectivity analysis although they are not the same. For example, connectome-based predictive modeling analyzes neuroimaging connectivity data (is a neuroimaging connectivity analysis method), but it does not incorporate the networked (dependent) characteristics of the brain when modeling brain connectivity edges—it assumes one connectivity edge to be an independent observation from another. See reviews on statistical network-based analysis (SNA) methods (Desmarais & Cranmer, 2017; Goldenberg et al., 2010; Kim et al., 2018; Matias & Robin, 2014; Newman, 2003; Smith et al., 2019; Wasserman & Faust, 1994).

3. The authors' definition of "network science" is unhelpful – "network science, a complexity-driven discipline focused on the shared architecture of networks emerging across physical, biological, and social domains." Staring a few minutes at this sentence, I'm still having difficulty understanding what it means. The general reader will unlikely understand it as well.

Response: Thank you for your comment. In response, we will provide a brief discussion on the definition of a complexity-driven discipline; but for a more comprehensive discussion, we refer the reader to network science textbooks and review papers (Barabási, 2013; Desmarais & Cranmer, 2017; Goldenberg et al., 2010; Kim et al., 2018; Matias & Robin, 2014; Newman, 2003; Smith et al., 2019; Wasserman & Faust, 1994).

A complex system is composed of interconnected parts with an involved arrangement of parts such that the mechanism of the system is intricate and difficult to understand resulting in a complex problem (Barabási, 2013). For example, our society is built upon cooperation between billions of people and connected infrastructures integrating billions of cell phones, computers and satellites. Crucially, our brain is a complex system that functions via coherent connective activities among billions of neurons. In complex systems, each individual component works in coordinations with other components; changes in one component may have cascading effects on the other components; and the properties of the system cannot be derived following knowledge of its individual components. Given the nature of complex systems and the numerous roles they play in our daily lives and in science, understanding these complex systems is major challenge of the 21st century resulting in network science, a complexity-driven discipline.

4. In general, the entire paper needs to be rewritten for clarity and understandability. This is important for broader impact and readership.

Response: Thank you for your comment. In response, we have rewritten the discussion section (see details in our response to comment 11) and made changes throughout the paper. In addition, we detail our efforts to clarify the focus of the paper; and we would like to add that we have been advised not to rewrite the results section upon communication with the editor (see response to comment 1). To clarify the focus of the paper, our rewritten manuscript emphasizes our method's contribution to current brain-behavior modeling literature and details its advantageous prediction accuracy and replicability capabilities, generalizable across datasets.

Specifically, in the introduction, we have emphasized the contribution of the current method to the broad

landscape of neuroimaging connectivity models. Compared with univariate and marginal association analyses (e.g., Marek et al., 2019; Satterthwaite et al., 2015) and CPMs⁸ (Finn et al., 2015; Shen et al., 2017), LatentSNA integrates the dependent organization of brain connectivity edges in the data generation process following the latent space modeling framework (e.g., Hoff, 2005a; Kim et al., 2018; Smith et al., 2019) thus preserving informative neurobiological indicators (Bassett & Sporns, 2017; Bullmore & Sporns, 2009; Zuo et al., 2012). Compared with graphical models (e.g., Dobra et al., 2004; Dobra et al., 2011; Friedman et al., 2008; Friedman, 2004), LatentSNA aims to understand the structure and property of the observed brain networks that ultimately facilitate understandings about brain’s relationship with behavior, instead of estimating the network structure. LatentSNA shows superior model performances with substantially improved out-of-sample prediction accuracy and effect estimation robustness. Both predicability and robustness of the method is generalizable across seven neuroimaging landmark studies: Alzheimer’s Disease Neuroimaging Initiative Grand Opportunities and ADNI Phase 2 (ADNI-GO/2) and ADNI Phase 3 (ADNI-3, Jack Jr et al., 2008; Petersen et al., 2010), Anti-Amyloid Treatment in Asymptomatic Alzheimer’s Disease (A4) (Sperling et al., 2014), Human Connectome Project Aging (HCP-A) (Bookheimer et al., 2019), Adolescent Brain Cognitive Development Study Baseline (ABCD-B) and second release (ABCD-2) (Casey et al., 2018), and transdiagnostic data collected at Yale (Greene et al., 2022).

To better clarify, we have modified our descriptions throughout the paper (see red highlighted texts in the revised manuscript). For considerations about space, we will not reiterate those edits here.

5. Page 3: the mentioned limitations of graphical models are not necessarily true. Point 1: Graphical models, particularly when higher-order connections are considered, do consider network topologies. Point 2: Graphical models have certainly been used for biomarker detection. Point 3: Graphical models have been certainly used for outcome prediction.

Response: Thank you for your comment. We have removed these points and focused on the difference between GMs and LatentSNA. A key task of graphical models, when applied to neuroimaging data, is to estimate and create brain connectivity networks based on whether signals from two brain regions are conditionally independent of each other (Ni et al., 2022). Although individual behaviors and outcomes can be incorporated in GMs, they are often used to influence the estimation of brain connectivity networks (Ni et al., 2022; Ni et al., 2019). In contrast, our proposed LatentSNA aims to understand the structure and property of brain networks (not their estimation) and how its structure is related to individual behaviors and outcomes. For further exploration of the differences between these two methodologies, we refer to discussions in Steinley (2021), Sweet and Wang (2024), Vitale et al. (2022), and Wang (2021).

To better clarify, we have added relevant **Introduction Section** description that read,

In the **Introduction**:

A critical challenge remains: connectivity edges are treated as independent observations, whereas evidence supports the dependent organization of brain networks as informative neurobiological indicators (Bassett & Sporns, 2017; Bullmore & Sporns, 2009; Zuo et al., 2012). Graphical models, consisting of both undirected Gaussian graphical models (e.g., Dobra et al., 2004; Dobra et al., 2011; Friedman et al., 2008; Friedman, 2004) and directed acyclic graphs (e.g., Friedman et al., 2000; Geiger & Heckerman, 2002), describe the conditional dependence among random variables and directly address the violation of the independence assumption. A key task of graphical models, when applied to neuroimaging data, is to estimate and create

⁸Connectome-based Predictive Modeling

brain connectivity networks based on whether signals from two brain regions are conditionally independent of each other (Ni et al., 2022). Although individual behaviors and outcomes can be incorporated in GMs, they are often used to influence the estimation of brain connectivity networks (Ni et al., 2022; Ni et al., 2019). In contrast, our proposed LatentSNA aims to understand the structure and property of brain networks (not their estimation) and how its structure is related to individual behaviors and outcomes. For further exploration of the differences between these two methodologies, we refer to discussions in Steinley (2021), Sweet and Wang (2024), Vitale et al. (2022), and Wang (2021).

6. Page 3, Para 2: “What differentiates a network-science-driven analytic approach is that it draws on insights regarding the universality of the communicative structures of real-world networks [barabasi2013network].” No idea what this means, particularly in the context of the current paper.

Response: Thank you for your comment. In response to the comment, we will provide a brief discussion on the universality of the communicative structures of real-world networks, which brain connectivity is an example of; but for a more comprehensive discussion, we refer the reader to network science textbooks and review papers (Barabási, 2013; Desmarais & Cranmer, 2017; Goldenberg et al., 2010; Kim et al., 2018; Matias & Robin, 2014; Newman, 2003; Smith et al., 2019; Wasserman & Faust, 1994), where universality of the communicative structures of real-world networks is commonly researched and discussed.

A key discovery of network science is that the properties and characteristics of networks spanning physical, biological, and social domains are similar to each other, as a result of being governed by universal organizing principles (Barabási, 2013)—this is in fact a foundation for statistical physics on network science (e.g., Cimini et al., 2019; Ghavasieh & De Domenico, 2022). A common set of mathematical tools are often used to explore these systems. Given the universality of networks, network researchers not only uncover generating principles of specific networks, but also investigate how widely these principles apply. A major scientific investigation within this field is to uncover the laws that shape network evolution and their consequences on network behavior (see network science review papers).

7. Page 3, Para 2: This paragraph is intended to motivate the use of statistical network-based analysis (SNA). But judging based on the way it is written, the motivation remains entirely unclear.

Response: Thank you for your comment. Page 3, Para 2 is the paragraph describing network science—What differentiates a network-science-driven analytic approach is that it draws on insights regarding the universality of the communicative structures of real-world networks. In response, we would like to refer our responses to comment 2, comment 3 and comment 6 that provide definitions on network science, complexity-driven discipline, the key characteristics of network science and how network science does not equate neuroimaging connectivity analysis. In particular, when neuroimaging connectivity analysis does not incorporate the networked (dependent) characteristics of the brain in model building, the research suffer a opportunity cost leading to low statistical power as evidenced by Figure 2 of the manuscript; instead, the analysis techniques should regard the dependence among connectivity edges (the interconnectivity of the brain) as a crucial piece of network generation and modeling, and when linking brain with behavior.

To better clarify, we have added relevant **Introduction Section** description that read,

In the **Introduction**:

What differentiates a network-science-driven analytic approach is that it draws on insights regarding the universality of the communicative structures of real-world networks (Barabási,

2013). *Characteristics such as the small-world property and sparsity are universal properties found in social networks (Wohlgemuth, 2012), political networks (Wang & Edgerton, 2022), the World Wide Web (Fatta et al., 2016), and human connectomes (Amaral et al., 2000; Bassett & Bullmore, 2006; Dubitzky et al., 2013). Shared network architectures, as a result of being governed by universal principles (Barabási, 2013; Wohlgemuth, 2012), allow us to use a common set of mathematical and statistical instruments for network modeling. Network science is characterized by the mathematical investigations about the universal principles of network generation—what mathematical principles define the generations of network with power law degree distributions, for example (Barabási, 2013; Newman, 2003; Wasserman & Faust, 1994). This discipline may have overlap with neuroimaging connectivity analysis although they are not the same. For example, connectome-based predictive modeling analyzes neuroimaging connectivity data (is a neuroimaging connectivity analysis method), but it does not incorporate the networked (dependent) characteristics of the brain when modeling brain connectivity edges—it assumes one connectivity edge to be an independent observation from another. See reviews on statistical network-based analysis (SNA) methods (Desmarais & Cranmer, 2017; Goldenberg et al., 2010; Kim et al., 2018; Matias & Robin, 2014; Newman, 2003; Smith et al., 2019; Wasserman & Faust, 1994).*

8. Page 4, Para 1: Why is “preserving transitivity and modularity” important? Why not consider other network properties?

Response: Thank you for your comment. We would like to clarify that preserving transitivity and modularity is an example of the characteristics of the networked connectivity modeling of the LatentSNA model, not the only characteristic. To make this clear, we have revised the sentence as “It comprises an integrated workflow containing three modules: networked connectivity modeling (e.g., preserving transitivity and modularity), psychometric behavior profiling, and two-way brain-behavior linking.” In addition, we would like to emphasize that higher-order dependencies in real-world networks include not only transitivity and modularity. Common examples of higher-order dependencies in real-world networks include homophily, balance, and clusterability (Heider, 1944; McPherson et al., 2001; Norman et al., 1965; Wasserman & Faust, 1994). Homophily is often associated with the transitive property of a network, explaining how new connections are established based on existing connections, also known as transitivity. Balance suggests a state of harmony, where positive connections are found among nodes with similar attributes, and negative connections are found among nodes with divergent attributes. Clusterability represents a more relaxed criteria for harmony than balance (Hoff, 2005a). With balanced cycles among triads, the entire network can be divided into cohesive groups with $x_{u,v} > 0$ if nodes u and v are in the same group, and $x_{u,v} < 0$ if they are in opposite groups (Harary, 1953; Hoff, 2005a). Therefore, the presence of higher-order dependencies such as balance contributes to relational patterns and topology across the whole network, including higher-order dependencies. By modeling higher-order dependencies, the proposed LatentSNA captures relational patterns across the entire network.

9. Figure 1a remains unclear and unhelpful.

Response: Thank you for your comment. Following your comment, we have revised Figure 1 of the manuscript for a clearer, more sequential depiction of how LatentSNA functions (Figure 2). In Figure 2, we show that neuroimaging and multivariate behavior data are inputted in the LatentSNA model, which subsequently goes through an iterative MCMC algorithm that estimates the model parameters theorizing the data generation process of three interconnected components. These three interconnected components

consist of (A) psychometric behavior profiling, (B) latent space network modeling and (C) brain-behavior linking. In Figure 2, the core components of LatentSNA are demonstrated to be mutually reinforcing each other in the iterative MCMC estimation process (highlighted in red). In this way, we highlight the connection between these components to help understand how LatentSNA works.

To better clarify, we have added relevant **figure caption** description that read,

In the **figure caption**:

The schematic diagram of LatentSNA. The LatentSNA Bayesian diagram demonstrates a holistic model for multivariate outcomes \mathbf{Y} and brain networks \mathbf{X} . The neuroimaging and multivariate behavior data are inputted in the LatentSNA model, which subsequently goes through an

Figure 2: The schematic diagram of LatentSNA. The LatentSNA Bayesian diagram demonstrates a holistic model for multivariate outcomes \mathbf{Y} and brain networks \mathbf{X} . The neuroimaging and multivariate behavior data are inputted in the LatentSNA model, which subsequently goes through an iterative MCMC algorithm that estimates the model parameters theorizing the data generation process of three interconnected components. These three interconnected components consist of (A) psychometric behavior profiling, (B) latent space network modeling and (C) brain-behavior linking. A. LatentSNA allows multivariate modeling of a latent behavior variable (e.g., internalizing psychopathology) with multiple components (e.g., anxious-depressed, withdrawn-depressed, and somatic complaints) to improve precision. The observed psychopathology is generated following a modified version of a psychometric Rasch model (Fischer & Molenaar, 2012), where outcomes are decomposed into item and person components. B. LatentSNA uses the symmetric bilinear interaction effect to capture network topology (transitivity, balance, and clusterability) (Hoff, 2008; Hoff, 2005b). C. LatentSNA makes inference of the relationships between brain and behavior, e.g., internalizing psychopathology and functional connectivity. We propose a joint latent variable model, where we allow the latent connectivity variables, \mathbf{Z} , and latent behavior variables, θ , to co-vary with a shared covariance matrix, Σ .

iterative MCMC algorithm that estimates the model parameters theorizing the data generation process of three interconnected components. These three interconnected components consist of (A) psychometric behavior profiling, (B) latent space network modeling and (C) brain-behavior linking. A. LatentSNA allows multivariate modeling of a latent behavior variable (e.g., internalizing psychopathology) with multiple components (e.g., anxious-depressed, withdrawn-depressed, and somatic complaints) to improve precision. The observed psychopathology is generated following a modified version of a psychometric Rasch model (Fischer & Molenaar, 2012), where outcomes are decomposed into item and person components. B. LatentSNA uses the symmetric bilinear interaction effect to capture network topology (transitivity, balance, and clusterability) (Hoff, 2008; Hoff, 2005b). C. LatentSNA makes inference of the relationships between brain and behavior, e.g., internalizing psychopathology and functional connectivity. We propose a joint latent variable model, where we allow the latent connectivity variables, \mathbf{Z} , and latent behavior variables, θ , to co-vary with a shared covariance matrix, Σ .

10. In Section “Markedly improved model performance is observed across imaging modalities, outcome measures and population demographics“, is “model fit” assessed for overfitting? If not, improvements might not be surprising and may be simply due to overfitting.

Response: Thank you for your comment. We would like to confirm that our method does not overfit. All of our model fit and prediction accuracy assessments are based on randomly sampled test data with no overlap with the training data. By definition, overfitting refers to the model being overly close to the training data with high prediction accuracy on the training data, and low prediction accuracy on the test data. Our model fit indices are calculated based on the test data; we show high prediction accuracy on the test data, which suggests that our model is not overfitted.

In addition, we assessed the replicability and robustness of our model across 10 random replications; we randomly split the data into 90% training and 10% test, fitted the model to the training component, and performed the random splitting of the data 10 times. We report the robustness assessment by comparing the consistency of the model-estimated effects across the 10 replication in Figure 1. The results show that our model is robust and replicable across replications, a strong evidence against overfitting. Overfitted models are not replicable nor robust because overfitted models only capture information in the training data with poor generalizability to the test data. By showing that our model results are robust and replicable across random replications, we provide strong evidence against overfitting.

11. The discussion is short and does not provide meaningful insights.

Response: Thank you for your comment. To directly address the comment, we have provided a comprehensive discussion about the strength, limitations and potential future extensions about the proposed method. We have added discussions regarding the specific methodological innovations of the proposed method in the field of (1) statistical network analysis, (2) neuroimaging connectivity analysis, (3) neuroimaging regression models, distinct from (4) graphical models. We have also added discussions about the limitations and future extension the method to (a) longitudinal settings, (b) clinical relevance, (c) subtype identification, and (d) other networks.

The proposed method represents a pioneering effort to extend statistical network analysis (SNA, Barabási, 2013; Desmarais & Cranmer, 2017; Goldenberg et al., 2010; Kim et al., 2018; Matias & Robin, 2014; Newman, 2003; Smith et al., 2019; Wasserman & Faust, 1994) to jointly model functional connectomes and continuous behavioral outcomes, significantly enhancing the ability to detect region-specific imaging

biomarkers. While the current SNA methods mentioned above primarily focus on modeling single networks, brain connectivity networks can be viewed as multiplex networks with multiple layers of brain connections observed across a shared set of brain regions. To model these multiplex structures, Gollini and Murphy (2016) and D’Angelo et al. (2018) propose utilizing a shared set of latent variables across layers, assuming a joint relational structure across sets of connectivity. Salter-Townshend and McCormick (2017) and MacDonald et al. (2022) alternatively distinguish between shared and individual components across layers. In contrast to these approaches, LatentSNA captures individual differences in brain connectivity networks across layers and identifies specific brain regions where the covariation between layers of brain networks and outcome variables is significant.

LatentSNA contributes to the current neuroimaging connectivity methods by offering a high power whole-brain approach for identifying brain-behavior links. A critical challenge of the current neuroimaging connectivity methods is that the connectivity edges are treated as independent observations, resulting in low statistical power and inflated Type-II errors. Univariate and marginal association analyses independently calculate associations between each connectivity edge and outcomes to identify significant links (e.g., Marek et al., 2019; Satterthwaite et al., 2015). CPM⁹ (Finn et al., 2015; Shen et al., 2017) identifies imaging biomarker detection by vectorizing unique pairwise edges from symmetric functional connectomes for behavior prediction.

LatentSNA makes a unique contribution to the existing neuroimaging regression methods such as network response regression (e.g., Simpson et al., 2024; Wang et al., 2017; Xia et al., 2020; Zhou & Müller, 2022) and scalar-on-network regression (Guha & Rodriguez, 2021; Morris et al., 2022; Wang et al., 2021; Zhao et al., 2022). In contrast to neuroimaging regression methods, LatentSNA posits a shared data generation process that links brain connectivity and outcomes. LatentSNA offers several advantages over network response and scalar-on-network regressions by positing a shared data generation process for connectivity and outcomes. First, unlike regression models that typically assume one-directional relationships between brain and behaviors or outcomes—estimating either the impact of brain on behavior or vice versa—LatentSNA acknowledges the mutual relationship between them. Changes in the brain often correlate with changes in behavior, but neuroplasticity suggests that disordered behaviors and dysfunctional environments can also influence brain function over time. Second, in scalar-on-network and network response regressions, using brain connectivity (or behavior outcomes) as predictors assumes that these variables are fully observed. This assumption becomes problematic when data includes partially missing observations for brain connectivity and individual outcomes. Regression methods struggle to handle situations where data is incomplete for both brain connectivity and outcomes. Lastly, traditional regression methods lack robustness in estimating parameters related to brain connectivity or behavior when they do not simultaneously model the reciprocal influence between them. In contrast, LatentSNA integrates both brain and behavior within a unified modeling framework, allowing mutual information exchange during model estimation. In summary, LatentSNA addresses these limitations by proposing a shared data generation process for multiplex brain networks and individual outcomes, facilitating a more comprehensive understanding of their interrelationships.

In terms of modeling objectives and approaches, LatentSNA methodologically differs from graphical models (GMs). GMs are utilized to represent joint probability distributions and establish conditional dependencies between random variables (Edwards, 2000; Lauritzen, 1996; Murphy, 2001). In GMs, researchers construct a graph (network) based on whether or not two variables are conditionally independent of each other. For LatentSNA, the edges in a brain functional network indicate the strength and nature of co-activation between two brain regions over time. In contrast, edges in covariance networks for GMs represent the con-

⁹Connectome-based Predictive Modeling

ditional dependence strength between variables. Crucially, covariance networks in GMs are derived from positive definite covariance matrices, specifying a particular type of relationship that is more constrained than real-world observed networks, including functional brain networks. Unlike GMs, SNAs accommodate a broader array of real-world network structures that are not feasible for covariance networks in GMs. Therefore, GMs and SNAs serve distinct purposes and are not directly comparable; latent variables in SNAs serve different functions than those in GMs. For further exploration of the differences between these methodologies, refer to discussions in Steinley (2021), Vitale et al. (2022), and Wang (2021).

In summary, compared to existing available approaches, LatentSNA is a novel application of network science in a generative statistical process. LatentSNA provides an unprecedented view of how network topology is implicated in brain dysfunction. Our study is among the very first to comprehensively uncover the brain's functional network architectures and their organizational principles related to childhood internalization. Our results show that internalizing psychopathology occurs in children via star-like topological architectures across functional systems. These star-like architectures suggest that the functional connectivity of a few star regions and cliques can explain individual differences in internalizing psychopathology and serve as the fingerprints to inform the development of internalizing psychopathology and its deterioration.

The LatentSNA model has limitations which prompt important future extensions. First, with LatentSNA, researchers can obtain satisfactorily accurate predictions of both connectivity and behavioral variants in cross-section settings. Accurate prediction is achieved by incorporating latent variables to separate signals from noise, using joint modeling frameworks, and allowing information communication between behavior and connectivity during model estimation. With the increased availability of longitudinal datasets such as ADNI and ABCD, it is of importance to extend current LatentSNA to longitudinal data. Longitudinal extensions would allow us to explore the temporal dynamics of fMRI across developmental or aging stages.

Second, LatentSNA offers substantially improved interpretability of neuroimaging study as it provides inferences about specific neuroimaging connectivity features that contribute to behavior outcomes. Future research are needed to investigate the clinical relevance of LatentSNA by exploring the specific contributions of different neuroimaging modalities in behavior predictions and investigating how these features can translate to clinical applications that ultimately improves the practical value of LatentSNA. In particular, a more clinical heterogeneous cohort is needed to understand functional substrates of psychopathologies. The ABCD study offers an opportunity to explore brain-behavior relationships in an unprecedentedly large children population. Yet, at these ages (9-10 at baseline and 11-12 at ABCD-2), relatively few children exhibit depression-related symptoms. Minimal participants in the ABCD study are diagnosed with depression through the KSADS, which limits the psychopathology findings. A more clinical heterogeneous children cohort is thus needed to explore psychopathologies in children.

Future work should also consider the group structure among the regions and how regions collectively contribute to internalizing psychopathology—past work has documented the importance of the group structures of the functional brain via functional systems in cognition and disease. Beyond neuroscience, the LatentSNA allows the detection of dependence between complex networks and nodal attributes, with applications in many other domains of science. Many complex systems such as social relationships, worldwide webs and transportation grids are impacted by higher-level attributes, and LatentSNA is a statistical technique that can open up many fields with rigorous and powerful analysis.

To better clarify, we have added relevant description in **Discussion Section** that read,

In the **Discussion Section**:

In this study, we developed a network-science driven analytic method that addresses the lack of power and inflated Type II errors in neuroimaging biomarker detection. Collectively, we have

fitted the method to 7 different datasets (ADNI-GO/2, ADNI-3, A4, HCP-A, ABCD-B, ABCD-2 and Transdiagnostic) involving 8 different imaging modalities and 20 outcome measures with a total of 8,003 - 11,861 subjects' information included. The results demonstrate that the proposed method is broadly applicable to a variety of imaging techniques including functional MRI (fMRI), T1-weighted structural MRI, Diffusion Tensor Imaging (DTI) and positron emission tomography (PET); and it is generalizable to different types of outcome measures, which include but are not limited to cognition, emotion, assessment of mental disorders and focal tau PET SUVR metrics. The proposed method shows consistent improvement in the model fit and model replicability across data situations against available alternatives for a variety of outcomes using functional and structural imaging modalities in developing, aging and transdiagnostic populations.

The proposed method represents a pioneering effort to extend statistical network analysis (SNA, Barabási, 2013; Desmarais & Cranmer, 2017; Goldenberg et al., 2010; Kim et al., 2018; Matias & Robin, 2014; Newman, 2003; Smith et al., 2019; Wasserman & Faust, 1994) to jointly model functional connectomes and continuous behavioral outcomes, significantly enhancing the ability to detect region-specific imaging biomarkers. While the current SNA methods mentioned above primarily focus on modeling single networks, brain connectivity networks can be viewed as multiplex networks with multiple layers of brain connections observed across a shared set of brain regions. To model these multiplex structures, Gollini and Murphy (2016) and D'Angelo et al. (2018) propose utilizing a shared set of latent variables across layers, assuming a joint relational structure across sets of connectivity. Salter-Townshend and McCormick (2017) and MacDonald et al. (2022) alternatively distinguish between shared and individual components across layers. In contrast to these approaches, LatentSNA captures individual differences in brain connectivity networks across layers and identifies specific brain regions where the covariation between layers of brain networks and outcome variables is significant.

LatentSNA contributes to the current neuroimaging connectivity methods by offering a high power whole-brain approach for identifying brain-behavior links. A critical challenge of the current neuroimaging connectivity methods is that the connectivity edges are treated as independent observations, resulting in low statistical power and inflated Type-II errors. Univariate and marginal association analyses independently calculate associations between each connectivity edge and outcomes to identify significant links (e.g., Marek et al., 2019; Satterthwaite et al., 2015). CPM¹⁰ (Finn et al., 2015; Shen et al., 2017) identifies imaging biomarker detection by vectorizing unique pairwise edges from symmetric functional connectomes for behavior prediction.

LatentSNA makes a unique contribution to the existing neuroimaging regression methods such as network response regression (e.g., Simpson et al., 2024; Wang et al., 2017; Xia et al., 2020; Zhou & Müller, 2022) and scalar-on-network regression (Guha & Rodriguez, 2021; Morris et al., 2022; Wang et al., 2021; Zhao et al., 2022). In contrast to neuroimaging regression methods, LatentSNA posits a shared data generation process that links brain connectivity and outcomes. LatentSNA offers several advantages over network response and scalar-on-network regressions by positing a shared data generation process for connectivity and outcomes. First, unlike regression models that typically assume one-directional relationships between brain and behaviors or outcomes—estimating either the impact of brain on behavior or vice versa—LatentSNA acknowledges the mutual relationship between them. Changes in the brain often correlate with changes in behavior, but neuroplasticity suggests that disordered behaviors and dysfunctional

¹⁰Connectome-based Predictive Modeling

environments can also influence brain function over time. Second, in scalar-on-network and network response regressions, using brain connectivity (or behavior outcomes) as predictors assumes that these variables are fully observed. This assumption becomes problematic when data includes partially missing observations for brain connectivity and individual outcomes. Regression methods struggle to handle situations where data is incomplete for both brain connectivity and outcomes. Lastly, traditional regression methods lack robustness in estimating parameters related to brain connectivity or behavior when they do not simultaneously model the reciprocal influence between them. In contrast, LatentSNA integrates both brain and behavior within a unified modeling framework, allowing mutual information exchange during model estimation. In summary, LatentSNA addresses these limitations by proposing a shared data generation process for multiplex brain networks and individual outcomes, facilitating a more comprehensive understanding of their interrelationships.

In terms of modeling objectives and approaches, LatentSNA methodologically differs from graphical models (GMs). GMs are utilized to represent joint probability distributions and establish conditional dependencies between random variables (Edwards, 2000; Lauritzen, 1996; Murphy, 2001). In GMs, researchers construct a graph (network) based on whether or not two variables are conditionally independent of each other. For LatentSNA, the edges in a brain functional network indicate the strength and nature of co-activation between two brain regions over time. In contrast, edges in covariance networks for GMs represent the conditional dependence strength between variables. Crucially, covariance networks in GMs are derived from positive definite covariance matrices, specifying a particular type of relationship that is more constrained than real-world observed networks, including functional brain networks. Unlike GMs, SNAs accommodate a broader array of real-world network structures that are not feasible for covariance networks in GMs. Therefore, GMs and SNAs serve distinct purposes and are not directly comparable; latent variables in SNAs serve different functions than those in GMs. For further exploration of the differences between these methodologies, refer to discussions in Steinley (2021), Vitale et al. (2022), and Wang (2021).

In summary, compared to existing available approaches, LatentSNA is a novel application of network science in a generative statistical process. LatentSNA provides an unprecedented view of how network topology is implicated in brain dysfunction. Our study is among the very first to comprehensively uncover the brain's functional network architectures and their organizational principles related to childhood internalization. Our results show that internalizing psychopathology occurs in children via star-like topological architectures across functional systems. These star-like architectures suggest that the functional connectivity of a few star regions and cliques can explain individual differences in internalizing psychopathology and serve as the fingerprints to inform the development of internalizing psychopathology and its deterioration.

The LatentSNA model has limitations which prompt important future extensions. First, with LatentSNA, researchers can obtain satisfactorily accurate predictions of both connectivity and behavioral variants in cross-section settings. Accurate prediction is achieved by incorporating latent variables to separate signals from noise, using joint modeling frameworks, and allowing information communication between behavior and connectivity during model estimation. With the increased availability of longitudinal datasets such as ADNI and ABCD, it is of importance to extend current LatentSNA to longitudinal data. Longitudinal extensions would allow us to explore the temporal dynamics of fMRI across developmental or aging stages.

Second, LatentSNA offers substantially improved interpretability of neuroimaging study as it provides inferences about specific neuroimaging connectivity features that contribute to behav-

ior outcomes. Future research are needed to investigate the clinical relevance of LatentSNA by exploring the specific contributions of different neuroimaging modalities in behavior predictions and investigating how these features can translate to clinical applications that ultimately improves the practical value of LatentSNA. In particular, a more clinical heterogeneous cohort is needed to understand functional substrates of psychopathologies. The ABCD study offers an opportunity to explore brain-behavior relationships in an unprecedentedly large children population. Yet, at these ages (9-10 at baseline and 11-12 at ABCD-2), relatively few children exhibit depression-related symptoms. Minimal participants in the ABCD study are diagnosed with depression through the KSADS, which limits the psychopathology findings. A more clinical heterogeneous children cohort is thus needed to explore psychopathologies in children.

Future work should also consider the group structure among the regions and how regions collectively contribute to internalizing psychopathology—past work has documented the importance of the group structures of the functional brain via functional systems in cognition and disease. Beyond neuroscience, the LatentSNA allows the detection of dependence between complex networks and nodal attributes, with applications in many other domains of science. Many complex systems such as social relationships, worldwide webs and transportation grids are impacted by higher-level attributes, and LatentSNA is a statistical technique that can open up many fields with rigorous and powerful analysis.

Reviewer 2

All my comments on the Figures, statistical method, validation strategies, and simulation settings are well addressed.

We really appreciate Reviewer 2 for the positive comments and for taking the time to review our work.

Reviewer 3

In this paper, Wang et al. introduce LatentSNA, a novel latent variable-based statistical network analysis designed to enhance predictive accuracy of behavior outcomes and interpretability as providing inferences about specific neuroimaging connectivity features that contribute to behavior outcomes. I appreciate the novelty of the propose methodology and its proof of generalizability using different image modalities for different type of outcome prediction through various datasets. However, further clarifications and rescoping are needed for the manuscript to be ready for publication.

We really appreciate Reviewer 3 for the positive comments and for taking the time to review our work. We have made changes according to your suggestions and our response to each comment is as follows.

1. Significance: The methodological significance of LatentSNA is well explained, particularly its increased statistical power, lower Type II error rates in prediction, and feature selection capabilities. Biologically, the authors emphasize that a strength of LatentSNA lies in its ability to identify neuroimaging biomarkers; however, it is not clearly stated what type of biomarkers are targeted. Additionally, the manuscript lacks detail on how these biomarkers might enhance clinical utility. Given that the ADNI and ABCD datasets are longitudinal, it is especially important to clarify what is meant by "prediction" in this context—whether LatentSNA aims to predict future behavioral outcomes based on baseline connectivity or merely capture associations in a cross-sectional manner.

Response: We greatly appreciate the reviewer for these comments. In response to the comment, “*The methodological significance of LatentSNA is well explained, particularly its increased statistical power, lower Type II error rates in prediction, and feature selection capabilities.*”, we would like to thank the reviewer for this positive feedback.

In response to the comment, “*Biologically, the authors emphasize that a strength of LatentSNA lies in its ability to identify neuroimaging biomarkers; however, it is not clearly stated what type of biomarkers are targeted.*”, we would like to clarify that our proposed method LatentSNA can identify susceptibility/risk neuroimaging biomarkers that are associated with an increased or decreased chance of developing a disease or medical condition. When a brain region is identified to be a significant neuroimaging biomarker for an individual outcome, we know that the functional connectivity information associated with this brain region is significantly correlated with the targeted individual outcome. For example, suppose we found that frontal-parietal lobe regions had significant negative correlations with individuals’ cognitive capabilities; this finding suggests that higher latent functional connectivity estimates for these brain regions are associated with lower cognitive capabilities. In this way, the identified frontal-parietal lobe regions are potential susceptibility/risk neuroimaging biomarker for medical conditions that are characterized by low cognition function, e.g., intellectual disabilities.

In response to the comment, “*Additionally, the manuscript lacks detail on how these biomarkers might enhance clinical utility.*”, we defer to the editor who has advised not to pursue the clinical utility of the identified biomarkers as it is outside of the scope of *Nature Methods*.

Lastly, in response to the comment, “*Given that the ADNI and ABCD datasets are longitudinal, it is especially important to clarify what is meant by “prediction” in this context—whether LatentSNA aims to predict future behavioral outcomes based on baseline connectivity or merely capture associations in a cross-sectional manner.*”, we would like to clarify that prediction accuracy is calculated based on independently sampled test data in the cross-sectional setting. To make this point clear, we have made changes throughout the manuscript to emphasize the cross-sectional nature of our study.

In addition, we have provided details about the cross-sectional nature of the current method and provide possibility for longitudinal extension in the discussion section. With LatentSNA, researchers can obtain satisfactorily accurate predictions of both connectivity and behavioral variants in cross-section settings. Accurate prediction is achieved by incorporating latent variables to separate signals from noise, using joint modeling frameworks, and allowing information communication between behavior and connectivity during model estimation. With the increased availability of longitudinal datasets such as ADNI and ABCD, it is of importance to extend current LatentSNA to longitudinal data. Longitudinal extensions would allow us to explore the temporal dynamics of fMRI across developmental or aging stages.

To better clarify the focus on the innovation of the method, we have added relevant descriptions in the **Discussion Section** that read,

In the **Discussion Section**:

The LatentSNA model has limitations which prompt important future extensions. First, with LatentSNA, researchers can obtain satisfactorily accurate predictions of both connectivity and behavioral variants in cross-section settings. Accurate prediction is achieved by incorporating latent variables to separate signals from noise, using joint modeling frameworks, and allowing information communication between behavior and connectivity during model estimation. With the increased availability of longitudinal datasets such as ADNI and ABCD, it is of importance to extend current LatentSNA to longitudinal data. Longitudinal extensions would allow us to explore the temporal dynamics of fMRI across developmental or aging stages.

2. It is not well illustrated the process of the proposed LatentSNA in Figure 1. Figure 1 provides different components in the LatentSNA, however, it is hard to make the connection between these components and understand how it works. A clearer, more sequential depiction of how LatentSNA functions would greatly improve comprehension.

Response: Following your comment, we have revised Figure 1 of the manuscript for a clearer, more sequential depiction of how LatentSNA functions (Figure 3). In Figure 3, we show that neuroimaging and multivariate behavior data are inputted in the LatentSNA model, which subsequently goes through an iterative MCMC algorithm that estimates the model parameters theorizing the data generation process of three interconnected components. These three interconnected components consist of (A) psychometric behavior profiling, (B) latent space network modeling and (C) brain-behavior linking. In Figure 3, the core components of LatentSNA are demonstrated to be mutually reinforcing each other in the iterative MCMC estimation process (highlighted in red). In this way, we highlight the connection between these components to help understand how LatentSNA works.

To better clarify, we have added relevant **figure caption** that read,

In the **figure caption**:

The schematic diagram of LatentSNA. The LatentSNA Bayesian diagram demonstrates a holistic model for multivariate outcomes Y and brain networks X . The neuroimaging and multivariate behavior data are inputted in the LatentSNA model, which subsequently goes through an iterative MCMC algorithm that estimates the model parameters theorizing the data generation process of three interconnected components. These three interconnected components consist of (A) psychometric behavior profiling, (B) latent space network modeling and (C) brain-behavior linking. A. LatentSNA allows multivariate modeling of a latent behavior variable (e.g., internalizing psychopathology) with multiple components (e.g., anxious-depressed, withdrawn-depressed, and somatic complaints) to improve precision. The observed psychopathology is generated following a modified version of a psychometric Rasch model (Fischer & Molenaar, 2012), where outcomes are decomposed into item and person components. B. LatentSNA uses the symmetric bilinear interaction effect to capture network topology (transitivity, balance, and clusterability) (Hoff, 2008; Hoff, 2005b). C. LatentSNA makes inference of the relationships between brain and behavior, e.g., internalizing psychopathology and functional connectivity. We propose a joint latent variable model, where we allow the latent connectivity variables, Z , and latent behavior variables, θ , to co-vary with a shared covariance matrix, Σ .

3. Experiment results: It would be nice to see more results on proving the generalizability of the method rather than just heavily focusing on the ABCD dataset for the following reasons: 1) Interpretability and Clinical Relevance: The interpretability of LatentSNA is presented as a major benefit, yet it would be helpful to more explicitly discuss the specific contributions of different neuroimaging modalities in behavior predictions. Moreover, clarifying how these features can translate to clinical applications would make the method's practical value more tangible. 2) Limitations of the ABCD Dataset for Psychopathology Findings: The authors focus on understanding different internalizing psychopathologies using the ABCD baseline and ABCD-2 datasets, particularly examining depression-related items from the Child Behavior Checklist (CBCL), such as anxious-depressed (13 items) and withdrawn-depressed (8 items). However, at these ages (9-10 at baseline and 11-12 at ABCD-2), relatively few children exhibit depression-related

Figure 3: The schematic diagram of LatentSNA. The LatentSNA Bayesian diagram demonstrates a holistic model for multivariate outcomes Y and brain networks X . The neuroimaging and multivariate behavior data are inputted in the LatentSNA model, which subsequently goes through an iterative MCMC algorithm that estimates the model parameters theorizing the data generation process of three interconnected components. These three interconnected components consist of (A) psychometric behavior profiling, (B) latent space network modeling and (C) brain-behavior linking. A. LatentSNA allows multivariate modeling of a latent behavior variable (e.g., internalizing psychopathology) with multiple components (e.g., anxious-depressed, withdrawn-depressed, and somatic complaints) to improve precision. The observed psychopathology is generated following a modified version of a psychometric Rasch model (Fischer & Molenaar, 2012), where outcomes are decomposed into item and person components. B. LatentSNA uses the symmetric bilinear interaction effect to capture network topology (transitivity, balance, and clusterability) (Hoff, 2008; Hoff, 2005b). C. LatentSNA makes inference of the relationships between brain and behavior, e.g., internalizing psychopathology and functional connectivity. We propose a joint latent variable model, where we allow the latent connectivity variables, Z , and latent behavior variables, θ , to co-vary with a shared covariance matrix, Σ .

symptoms. Moreover, there are minimal participants diagnosed with depression through the KSADS in the ABCD dataset, which limits the robustness of the psychopathology findings.

Response: We thank the reviewer for the comment. In response to the suggestion to proving the generalizability of the real-data application results outside of the ABCD study, we defer to the editor who has advised not to pursue proving the generalizability of the ABCD data results across datasets as it is outside of the scope of *Nature Methods*.

We agree with the reviewer that the ABCD Dataset is limited for investigating Psychopathology Findings. In response, we have provided discussions about the limitation of the ABCD study on investigating Psychopathology findings. A more clinical heterogeneous cohort is needed to understand functional substrates of psychopathologies. The ABCD study offers an opportunity to explore brain-behavior relationships in

Figure 4: Reproducibility analysis of CPM (green) and latentSNA (red) methods across different behaviors, fMRI conditions, and datasets. Each boxplot summarizes the reproducibility performance through correlations across 10 independent replications. We use the absolute correlation between the estimated effects (regression coefficient estimates in CPM and covariance estimates in LatentSNA) across replications to represent the replicability and robustness of both methods. Higher correlation values indicate better reproducibility.

an unprecedentedly large children population. Yet, at these ages (9-10 at baseline and 11-12 at ABCD-2), relatively few children exhibit depression-related symptoms. Minimal participants in the ABCD study are diagnosed with depression through the KSADS, which limits the psychopathology findings. A more clinical heterogeneous children cohort is thus needed to explore psychopathologies in children.

To better clarify, we have added relevant **Discussion Section** that read,

In the **Discussion Section**:

Second, LatentSNA offers substantially improved interpretability of neuroimaging study as it provides inferences about specific neuroimaging connectivity features that contribute to behavior outcomes. Future research are needed to investigate the clinical relevance of LatentSNA by exploring the specific contributions of different neuroimaging modalities in behavior predictions and investigating how these features can translate to clinical applications that ultimately improves the practical value of LatentSNA. In particular, a more clinical heterogeneous cohort is needed to understand functional substrates of psychopathologies. The ABCD study offers an opportunity to explore brain-behavior relationships in an unprecedentedly large children population. Yet, at these ages (9-10 at baseline and 11-12 at ABCD-2), relatively few children exhibit depression-related symptoms. Minimal participants in the ABCD study are diagnosed with depression through the KSADS, which limits the psychopathology findings. A more clinical heterogeneous children cohort is thus needed to explore psychopathologies in children.

Lastly, we have provided additional discussions on the generalizability of model performance across studies—that the proposed method can be fitted to a range of imaging modalities, outcome measures and study populations with strong model fit and robust/replicable effect estimation. The lack of robustness and replicability of current fMRI studies is a well-known challenge (see Wu et al., 2023; Zuo et al., 2019). We investigate the robustness and replicability performance of our proposed method by comparing covariance effect estimation across random samplings of the test data, i.e., replicability with the same data. We calculated the absolute correlation of the estimated effects between replications and report the results in Figure 4. The results show that our model estimated effects (covariance/correlations between brain and behaviors) are consistent across independent replications—when we randomly split the data into 90%

training and 10% test samples. The CPM, on the other hand, shows lower replicability and robustness. LatentSNA shows consistently higher replicability (with above 0.75 correlations) across datasets, while CPM shows substantially lower and more variable reproducibility (correlations ranging from 0.25 to 0.75). To better clarify, we have added relevant **Results Section** that read,

In the **Results Section**:

The lack of robustness and replicability of current fMRI studies is a well-known challenge (see Wu et al., 2023; Zuo et al., 2019). We investigate the robustness and replicability performance of our proposed method by comparing covariance effect estimation across random samplings of the test data, i.e., replicability with the same data. We calculated the absolute correlation of the estimated effects between replications and report the results in Figure 4. The results show that our model estimated effects (covariance/correlations between brain and behaviors) are consistent across independent replications—when we randomly split the data into 90% training and 10% test samples. The CPM, on the other hand, shows lower replicability and robustness. LatentSNA shows consistently higher replicability (with above 0.75 correlations) across datasets, while CPM shows substantially lower and more variable reproducibility (correlations ranging from 0.25 to 0.75).

4. Minor: Figure 6, there is no color legend for (B,C).

Response: Thank you for your comment. We have added color legend for (B,C). In B and C, red line indicates positive connectivity edges, and blue line indicates negative connectivity edges.

To better clarify, we have added relevant **figure caption** that read,

In the **figure caption**:

In B and C, red line indicates positive connectivity edges, and blue line indicates negative connectivity edges.

5. A more detailed documentation, including clear instructions for installation (specifying prerequisite packages), explanations of outputs, and a sample toy example with evaluations, would be greatly appreciated.

Response: Thank you for your comment. We have included clear instructions for installation (specifying prerequisite packages), explanations of outputs, and a sample toy example with evaluations in our github repository. More specifically, we have added a comprehensive prerequisites section listing all required R packages (MASS, magic, psych, coda). We have provided clear explanations for each step of the analysis workflow, from data generation to model fitting. We have included a simulation example, demonstrating both model estimation accuracy and prediction performance. We have added thorough evaluations comparing our method with CPM, showing LatentSNA's superior performance in both identifying significant brain regions and predicting behavioral outcomes. The code is available via the user-friendly github at <https://github.com/selenashuowang/latentSNA> with tutorial.

To better clarify, we have added relevant **Code Availability** that read,

In the **Code Availability**:

The estimation algorithm for this paper was implemented in the open sourced programming language for statistical computing and graphics, R. The code is available via the user-friendly github at <https://github.com/selenashuowang/latentSNA> with tutorial. In this github repository, we have provided instructions for installation (specifying prerequisite packages), explanations of outputs, and a sample toy example with evaluations in our github repository. More specifically, we have added a comprehensive prerequisites section listing all required R packages (MASS, magic, psych, coda). We have provided clear explanations for each step of the analysis workflow, from data generation to model fitting. We have included a simulation example, demonstrating both model estimation accuracy and prediction performance. We have added thorough evaluations comparing our method with CPM, showing LatentSNA's superior performance in both identifying significant brain regions and predicting behavioral outcomes.

References

- Amaral, L. A. N., Scala, A., Barthélemy, M., & Stanley, H. E. (2000). Classes of small-world networks. *Proceedings of the national academy of sciences*, 97(21), 11149–11152.
- Barabási, A.-L. (2013). Network science. *Philosophical Transactions of the Royal Society A: Mathematical, Physical and Engineering Sciences*, 371(1987), 20120375.
- Bassett, D. S., & Sporns, O. (2017). Network neuroscience. *Nature neuroscience*, 20(3), 353–364.
- Bassett, D. S., & Bullmore, E. (2006). Small-world brain networks. *The neuroscientist*, 12(6), 512–523.
- Bookheimer, S. Y., Salat, D. H., Terpstra, M., Ances, B. M., Barch, D. M., Buckner, R. L., Burgess, G. C., Curtiss, S. W., Diaz-Santos, M., Elam, J. S., et al. (2019). The lifespan human connectome project in aging: An overview. *Neuroimage*, 185, 335–348.
- Bullmore, E., & Sporns, O. (2009). Complex brain networks: Graph theoretical analysis of structural and functional systems. *Nature reviews neuroscience*, 10(3), 186–198.
- Casella, G., & Berger, R. (2024). *Statistical inference*. CRC Press.
- Casey, B. J., Cannonier, T., Conley, M. I., Cohen, A. O., Barch, D. M., Heitzeg, M. M., Soules, M. E., Teslovich, T., Dellarco, D. V., Garavan, H., et al. (2018). The adolescent brain cognitive development (ab cd) study: Imaging acquisition across 21 sites. *Developmental cognitive neuroscience*, 32, 43–54.
- Cimini, G., Squartini, T., Saracco, F., Garlaschelli, D., Gabrielli, A., & Caldarelli, G. (2019). The statistical physics of real-world networks. *Nature Reviews Physics*, 1(1), 58–71.
- D’Angelo, S., Murphy, T. B., & Alfò, M. (2018). Latent space modelling of multidimensional networks with application to the exchange of votes in eurovision song contest. *The Annals of Applied Statistics*. <https://api.semanticscholar.org/CorpusID:88523060>
- Desmarais, B. A., & Cranmer, S. J. (2017). *Statistical inference in political networks research*. Oxford University Press Oxford.
- Dobra, A., Hans, C., Jones, B., Nevins, J. R., Yao, G., & West, M. (2004). Sparse graphical models for exploring gene expression data. *Journal of Multivariate Analysis*, 90(1), 196–212.
- Dobra, A., Lenkoski, A., & Rodriguez, A. (2011). Bayesian inference for general gaussian graphical models with application to multivariate lattice data. *Journal of the American Statistical Association*, 106(496), 1418–1433.
- Dubitzky, W., Wolkenhauer, O., Cho, K.-H., & Yokota, H. (2013). *Encyclopedia of systems biology* (Vol. 402). Springer New York, NY, USA:
- Edwards, D. (2000). *Introduction to graphical modelling*. Springer Science & Business Media.
- Fatta, D. D., Caputo, F., Evangelista, F., & Dominici, G. (2016). Small world theory and the world wide web: Linking small world properties and website centrality. *International Journal of Markets and Business Systems*, 2(2), 126–140.
- Finn, E. S., Shen, X., Scheinost, D., Rosenberg, M. D., Huang, J., Chun, M. M., Papademetris, X., & Constable, R. T. (2015). Functional connectome fingerprinting: Identifying individuals using patterns of brain connectivity. *Nature neuroscience*, 18(11), 1664–1671.
- Fischer, G. H., & Molenaar, I. W. (2012). Rasch models: Foundations, recent developments, and applications.
- Friedman, J., Hastie, T., & Tibshirani, R. (2008). Sparse inverse covariance estimation with the graphical lasso. *Biostatistics*, 9(3), 432–441.
- Friedman, N. (2004). Inferring cellular networks using probabilistic graphical models. *Science*, 303(5659), 799–805.
- Friedman, N., Linial, M., Nachman, I., & Pe’er, D. (2000). Using bayesian networks to analyze expression data. *Proceedings of the fourth annual international conference on Computational molecular biology*, 127–135.
- Geiger, D., & Heckerman, D. (2002). Parameter priors for directed acyclic graphical models and the characterization of several probability distributions. *The Annals of Statistics*, 30(5), 1412–1440.

- Ghavasieh, A., & De Domenico, M. (2022). Statistical physics of network structure and information dynamics. *Journal of Physics: Complexity*, 3(1), 011001.
- Goldenberg, A., Zheng, A. X., Fienberg, S. E., & Airoldi, E. M. (2010). A survey of statistical network models.
- Gollini, I., & Murphy, T. B. (2016). Joint modeling of multiple network views. *Journal of Computational and Graphical Statistics*, 25(1), 246–265.
- Greene, A. S., Shen, X., Noble, S., Horien, C., Hahn, C. A., Arora, J., Tokoglu, F., Spann, M. N., Carrión, C. I., Barron, D. S., et al. (2022). Brain–phenotype models fail for individuals who defy sample stereotypes. *Nature*, 609(7925), 109–118.
- Guha, S., & Rodriguez, A. (2021). Bayesian regression with undirected network predictors with an application to brain connectome data. *Journal of the American Statistical Association*, 116(534), 581–593.
- Harary, F. (1953). On the notion of balance of a signed graph. *Michigan Mathematical Journal*, 2(2), 143–146.
- Heider, F. (1944). Social perception and phenomenal causality. *Psychological review*, 51(6), 358.
- Hoff, P. (2005a). Bilinear mixed-effects models for dyadic data. *J. Amer. Statist. Assoc.*, 100(469), 286–295.
- Hoff, P. (2008). Modeling homophily and stochastic equivalence in symmetric relational data. *Advances in neural information processing systems*, 657–664.
- Hoff, P. D. (2005b). Bilinear mixed-effects models for dyadic data. *Journal of the American Statistical Association*, 100(469), 286–295.
- Jack Jr, C. R., Bernstein, M. A., Fox, N. C., Thompson, P., Alexander, G., Harvey, D., Borowski, B., Britson, P. J., L. Whitwell, J., Ward, C., et al. (2008). The alzheimer’s disease neuroimaging initiative (adni): Mri methods. *Journal of Magnetic Resonance Imaging: An Official Journal of the International Society for Magnetic Resonance in Medicine*, 27(4), 685–691.
- Kim, B., Lee, K. H., Xue, L., & Niu, X. (2018). A review of dynamic network models with latent variables. *Statistics surveys*, 12, 105.
- Lauritzen, S. L. (1996). *Graphical models* (Vol. 17). Clarendon Press.
- MacDonald, P. W., Levina, E., & Zhu, J. (2022). Latent space models for multiplex networks with shared structure. *Biometrika*, 109(3), 683–706.
- Marek, S., Tervo-Clemmens, B., Nielsen, A. N., Wheelock, M. D., Miller, R. L., Laumann, T. O., Earl, E., Foran, W. W., Cordova, M., Doyle, O., et al. (2019). Identifying reproducible individual differences in childhood functional brain networks: An abcd study. *Developmental cognitive neuroscience*, 40, 100706.
- Matias, C., & Robin, S. (2014). Modeling heterogeneity in random graphs through latent space models: A selective review. *ESAIM: Proceedings and Surveys*, 47, 55–74.
- McPherson, M., Smith-Lovin, L., & Cook, J. M. (2001). Birds of a feather: Homophily in social networks. *Annual review of sociology*, 27(1), 415–444.
- Morris, E. L., He, K., & Kang, J. (2022). Scalar on network regression via boosting. *The annals of applied statistics*, 16(4), 2755.
- Murphy, K. (2001). An introduction to graphical models. *Rap. tech*, 96, 1–19.
- Newman, M. E. (2003). The structure and function of complex networks. *SIAM review*, 45(2), 167–256.
- Ni, Y., Baladandayuthapani, V., Vannucci, M., & Stingo, F. C. (2022). Bayesian graphical models for modern biological applications. *Statistical Methods & Applications*, 31(2), 197–225.
- Ni, Y., Stingo, F. C., & Baladandayuthapani, V. (2019). Bayesian graphical regression. *Journal of the American Statistical Association*, 114(525), 184–197.
- Norman, R. Z., et al. (1965). Structural models: An introduction to the theory of directed graphs.
- Petersen, R. C., Aisen, P. S., Beckett, L. A., Donohue, M. C., Gamst, A. C., Harvey, D. J., Jack Jr, C., Jagust, W. J., Shaw, L. M., Toga, A. W., et al. (2010). Alzheimer’s disease neuroimaging initiative (adni) clinical characterization. *Neurology*, 74(3), 201–209.

- Reli3n, J. D. A., Kessler, D., Levina, E., & Taylor, S. F. (2019). Network classification with applications to brain connectomics. *The annals of applied statistics*, *13*(3), 1648.
- Salter-Townshend, M., & McCormick, T. H. (2017). Latent space models for multiview network data. *The annals of applied statistics*, *11*(3), 1217.
- Satterthwaite, T. D., Kable, J. W., Vandekar, L., Katchmar, N., Bassett, D. S., Baldassano, C. F., Ruparel, K., Elliott, M. A., Sheline, Y. I., Gur, R. C., et al. (2015). Common and dissociable dysfunction of the reward system in bipolar and unipolar depression. *Neuropsychopharmacology*, *40*(9), 2258–2268.
- Shen, X., Finn, E. S., Scheinost, D., Rosenberg, M. D., Chun, M. M., Papademetris, X., & Constable, R. T. (2017). Using connectome-based predictive modeling to predict individual behavior from brain connectivity. *nature protocols*, *12*(3), 506–518.
- Simpson, S. L., Shappell, H. M., & Bahrami, M. (2024). Statistical brain network analysis. *Annual Review of Statistics and Its Application*, *11*.
- Smith, A. L., Asta, D. M., & Calder, C. A. (2019). The geometry of continuous latent space models for network data. *Statistical science: a review journal of the Institute of Mathematical Statistics*, *34*(3), 428.
- Sperling, R. A., Rentz, D. M., Johnson, K. A., Karlawish, J., Donohue, M., Salmon, D. P., & Aisen, P. (2014). The a4 study: Stopping ad before symptoms begin? *Science translational medicine*, *6*(228), 228fs13–228fs13.
- Steinley, D. (2021). Recent advances in (graphical) network models [PMID: 34029161]. *Multivariate Behavioral Research*, *56*(2), 171–174. <https://doi.org/10.1080/00273171.2021.1911777>
- Sweet, T., & Wang, S. (2024). Network science in psychology. *arXiv preprint arXiv:2410.00301*.
- Vitale, M. P., Giordano, G., & Ragozini, G. (2022). Discussion to: Bayesian graphical models for modern biological applications by Y. Ni, V. Baladandayuthapani, M. Vannucci and F.C. Stingo. *Statistical Methods & Applications*, *31*(2), 269–278. <https://doi.org/10.1007/s10260-021-00603->
- Wang, L., Durante, D., Jung, R. E., & Dunson, D. B. (2017). Bayesian network–response regression. *Bioinformatics*, *33*(12), 1859–1866.
- Wang, L., Lin, F. V., Cole, M., & Zhang, Z. (2021). Learning clique subgraphs in structural brain network classification with application to crystallized cognition. *NeuroImage*, *225*, 117493.
- Wang, S. (2021). Recent integrations of latent variable network modeling with psychometric models. *Frontiers in psychology*, *12*, 773289.
- Wang, S., & Edgerton, J. (2022). Resilience to stress in bipartite networks: Application to the islamic state recruitment network. *Journal of Complex Networks*, *10*(4), cnac017.
- Wasserman, S., & Faust, K. (1994). *Social network analysis: Methods and applications*.
- Wohlgemuth, J. (2012). *Small world properties of facebook group networks*. University of Nebraska at Omaha.
- Wu, J., Li, J., Eickhoff, S. B., Scheinost, D., & Genon, S. (2023). The challenges and prospects of brain-based prediction of behaviour. *Nature human behaviour*, *7*(8), 1255–1264.
- Xia, C. H., Ma, Z., Cui, Z., Bzdok, D., Thirion, B., Bassett, D. S., Satterthwaite, T. D., Shinohara, R. T., & Witten, D. M. (2020). *Multi-scale network regression for brain-phenotype associations* (tech. rep.). Wiley Online Library.
- Zhang, Z., Allen, G. I., Zhu, H., & Dunson, D. (2019). Tensor network factorizations: Relationships between brain structural connectomes and traits. *Neuroimage*, *197*, 330–343.
- Zhao, Y., Li, T., & Zhu, H. (2022). Bayesian sparse heritability analysis with high-dimensional neuroimaging phenotypes. *Biostatistics*, *23*(2), 467–484.
- Zhou, Y., & M3ller, H.-G. (2022). Network regression with graph laplacians. *The Journal of Machine Learning Research*, *23*(1), 14383–14423.
- Zuo, X.-N., Biswal, B. B., & Poldrack, R. A. (2019). Reliability and reproducibility in functional connectomics.

Zuo, X.-N., Ehmke, R., Mennes, M., Imperati, D., Castellanos, F. X., Sporns, O., & Milham, M. P. (2012). Network centrality in the human functional connectome. *Cerebral cortex*, 22(8), 1862–1875.